# An antisense oligonucleotide-based strategy to ameliorate cognitive dysfunction in the 22q11.2 Deletion Syndrome

Pratibha Thakur[1†], Martin Lackinger[1,2†], Anastasia Diamantopoulou[1‡], Sneha Rao[1], Yijing Chen[1,3], Khakima Khalizova[1,2], Annie Ferng[4], Curt Mazur[4], Holly Kordasiewicz[4], Robert J Shprintzen[5], Sander Markx[2,6], Bin Xu[2,6], Joseph A Gogos[1,2,6,7,8]*

[1]Mortimer B. Zuckerman Mind Brain and Behavior Institute, Columbia University, New York, United States; [2]Stavros Niarchos Foundation Center for Precision Psychiatry and Mental Health, Columbia University, New York, United States; [3]Department of Genetics and Development, Columbia University Irving Medical Center, New York, United States; [4]Ionis Pharmaceuticals, Inc, Carlsbad, United States; [5]The Virtual Center for Velo-Cardio-Facial-Syndrome, Inc, Manlius, United States; [6]Department of Psychiatry, Vagelos College of Physicians & Surgeons, Columbia University, New York, United States; [7]Department of Physiology and Cellular Biophysics, College of Physicians and Surgeons, Columbia University, New York, United States; [8]Department of Neuroscience, Columbia University, New York, United States

*For correspondence:
jag90@columbia.edu

[†]These authors contributed equally to this work

Present address: [‡]Institute of Neurophysiology, Goethe University, Theodor-Stern-Kai 7, Frankfurt am Main, Germany

## eLife Assessment

This is an **important** study that establishes how anti-sense oligonucleotides (ASOs) degrading a specific target protein called EMC10 can rescue neuronal function in models of chromosome 22.11.2 deletions. The authors use human iPSC-derived neurons and a mouse model to provide **compelling** data for the rescue of cellular and cognitive features of 22.11.2 deletion phenotypes upon ASO regulation of EMC10. These pre-clinical data are of interest because they support reduction of ECM10 as a promising therapeutic strategy.

**Abstract** Adults and children with the 22q11.2 Deletion Syndrome demonstrate cognitive, social, and emotional impairments and high risk for schizophrenia. Work in mouse model of the 22q11.2 deletion provided compelling evidence for abnormal expression and processing of microRNAs. A major transcriptional effect of the microRNA dysregulation is upregulation of *Emc10*, a component of the ER membrane complex, which promotes membrane insertion of a subset of polytopic and tail-anchored membrane proteins. We previously uncovered a key contribution of EMC10 in mediating the behavioral phenotypes observed in 22q11.2 deletion mouse models. Here, we show that expression and processing of miRNAs is abnormal and *EMC10* expression is elevated in neurons derived from 22q11.2 deletion carriers. Reduction of *EMC10 levels* restores defects in neurite outgrowth and calcium signaling in patient neurons. Furthermore, antisense oligonucleotide administration and normalization of *Emc10* in the adult mouse brain not only alleviates cognitive deficits in social and spatial memory but remarkably sustains these improvements for over 2 months post-injection, indicating its therapeutic potential. Broadly, our study integrates findings from both animal models and human neurons to elucidate the translational potential

of modulating *EMC10* levels and downstream targets as a specific venue to ameliorate disease progression in 22q11.2 Deletion Syndrome.

## Introduction

Adults and children with the 22q11.2 Deletion Syndrome (22q11.2DS) demonstrate cognitive, social, and emotional impairments (*Morrison et al., 2020*; *Woodin et al., 2001*; *McCabe et al., 2013*). 22q11.2 deletions are also one of the strongest genetic risk factors for schizophrenia (SCZ) (*Xu et al., 2008*). There are currently no targeted therapies that address the underlying molecular mechanisms of 22q11.2DS. Previous work in a model of the 22q11.2 deletion, carrying a hemizygous 1.3 Mb deficiency on mouse chromosome 16 [*Df(16)A*], which is syntenic to the 1.5 Mb 22q11.2 deletion [*Df(16)A⁺/⁻*mice] revealed a distinct behavioral and cognitive profile (*Stark et al., 2008*; *Piskorowski et al., 2016*). Molecular analysis of the *Df(16)A⁺/⁻*strain provided compelling evidence for abnormal processing of brain-enriched microRNAs (miRNAs) (*Stark et al., 2008*; *Xu et al., 2013*). The *Df(16) A⁺/⁻*related miRNA dysregulation is due to (i) hemizygosity of *Dgcr8*, a component of the "microprocessor" complex that is essential for miRNA production (*Gregory et al., 2004*) and (ii) hemizygosity of miRNA genes residing within the deletion, including *Mir185*. Reduction of *Mir185* levels and to a lesser degree of miRNAs residing outside the deletion {such as *Mir485* *Xu et al., 2013*} result in a de-repression of *Emc10* gene (alias Mirta22), whose expression is under the repressive control of *miRNAs Xu et al., 2013*. Indeed, comprehensive RNA profiling of *Df(16)A⁺/⁻*mice found that postnatal elevation in the expression of the *Emc10* gene represents a key transcriptional effect of the 22q11.2 deletion (*Xu et al., 2013*). Increased brain expression of *Emc10* is recapitulated in *Df(16)A⁺/⁻*primary neurons (*Sun et al., 2018*) as well as in mouse models of the more common 3 Mb 22q11.2 deletion (*Saito et al., 2020*). Other miRNA targets are dysregulated, but their levels of change are subtler and more variable. *Emc10* encodes for a component of the ER membrane complex (EMC), which promotes membrane insertion and maturation of a subset of polytopic and tail-anchored membrane proteins including neurotransmitter receptors, channels, and transporters (*Guna et al., 2018*; *Chitwood et al., 2018*; *Richard et al., 2013*; *Satoh et al., 2015*; *Bircham et al., 2011*; *Louie et al., 2012*; *Shurtleff et al., 2018*; *Tian et al., 2019*). *Emc10* is a prenatally biased gene with high expression in embryonic life that gradually subsides after birth (*Xu et al., 2013*), a developmental pattern of expression conserved between mice, humans and nonhuman primates (*Diamantopoulou et al., 2017*). *Emc10* Loss-of-Function (LoF) mutation that leads to reduction of *Emc10* levels rescues key cellular, cognitive and behavioral alterations in the *Df(16)A⁺/⁻*mice (*Diamantopoulou et al., 2017*). However, whether similar beneficial effects could be achieved in human neurons and whether Emc10 normalization in the adult brain could reverse established cognitive deficits remained unknown.

Here, we show that 22q11.2 deletion results in abnormal processing of miRNAs in human neurons and in turn drives misexpression of *EMC10* as previously described in animal models (*Stark et al., 2008*). Human *EMC10* expression is elevated in neurons derived from 22q11.2 deletion carriers and reversal of *EMC10* expression leads to restoration of key morphological and functional alterations linked to 22q11.2 deletions, supporting normalization of *EMC10* expression as a disease-modifying intervention. Toward this end, we also show that antisense oligonucleotide (ASO)-mediated *Emc10* normalization in the adult mouse brain is effective at reversing cognitive alterations. Improvements in cognition are sustained for over 2 months post ASO administration, underscoring the potential of this approach for providing durable therapeutic benefits. The observations that ASO-mediated Emc10 reduction in adult mouse brain rescues cognitive deficits linked to 22q11.2 deletion strongly support a key contribution of Emc10 and Emc10-dependent membrane protein trafficking in mediating the effects of 22q11.2 deletions on cognitive function and pave the way toward translating these observations into potential disease-modifying therapeutic interventions.

## Results

To investigate whether miRNA dysregulation and upregulation of *EMC10* is also prominent in cortical neurons from patients carrying 22q11.2 deletions (*Figure 1A*), we used hiPSC lines obtained from three independent 22q11.2DS/SCZ donors carrying a 3 Mb deletion and diagnosed with SCZ, along with matched healthy controls (*Supplementary file 1*) to ensure the robustness and generalizability

**eLife digest** Our genetic material is 'packaged' into chromosomes, which are compact structures made of DNA found in every cell. Chromosomal abnormalities occur either when a person has the wrong number of chromosomes, or when parts of a chromosome are deleted or duplicated. This can cause a wide range of health problems, including psychiatric and cognitive symptoms.

Individuals with '22q11.2 deletion syndrome' are missing a small DNA segment on chromosome 22. This results in cognitive impairment and a high risk of disorders like schizophrenia. Recent research in mice has shown that the DNA deletion in 22q11.2 deletion syndrome disrupts tiny molecules called microRNAs, which help control the activity of many genes (usually by 'switching' them off).

The gene for EMC10 is affected by these changes in microRNAs. Normally, the EMC10 protein plays a role in maintaining the health of brain cells. However, further studies in mice have shown that a DNA deletion equivalent to the one in humans with 22q11.2 deletion syndrome leads to excessive production of EMC10 – suggesting that too much EMC10 can be harmful. Reducing the amount of EMC10 in these mice restored normal brain function and behaviour.

Based on these results, Thakur, Lackinger et al. investigated if the same connection between disrupted microRNAs and abnormally high EMC10 levels also occurred in humans with 22q11.2 deletion syndrome. Analysis of cultured brain cells derived from patients with the condition confirmed that problems similar to those previously observed in mice emerged: disruption of microRNAs led to the cells accumulating too much EMC10, resulting in abnormal cell behaviour such as defective growth.

Thakur, Lackinger et al. then tested if gene-targeting tools called antisense oligonucleotides (ASOs) could be used to treat 22q11.2 syndrome. Similarly to microRNAs, ASOs work by turning off specific genes. When adult mice with the deletion were given ASOs targeting the gene for EMC10, their levels of EMC10 protein decreased, and their 'cognitive function' (including performance in simple memory tests) improved. Importantly, these benefits lasted over two months after a single treatment.

These results shed new light on the molecular mechanisms behind the effects of 22q11.2 deletion in humans. They also highlight ASOs targeting the EMC10 gene as a potential treatment for the condition, even when treatment begins in adulthood. In the future, Thakur, Lackinger et al. hope that this work will help to develop therapies that improve quality of life for those affected by the syndrome.

of our findings across different genetic backgrounds. The first patient/control pair is derived from dizygotic twins discordant for the 22q11.2DS and SCZ [Q6 (22q11.2) and Q5 (Ctrl)] (*Figure 1—figure supplement 1A–D*). The second patient/control pair is derived from siblings [Q1 (22q11.2) and Q2 (Ctrl)] while the third patient/control pair is a case and age/sex-matched unrelated control pair from the NIMH Repository and Genomic Resource [QR27 (22q11.2) and QR20 (Ctrl)].

We examined whether 22q11.2 deletion results in abnormal processing of miRNAs in human neurons as we have previously described in animal models (*Stark et al., 2008*). We performed parallel small RNA/miRNA sequencing on DIV8 differentiated human cortical neurons from the sibling (Q5/Q6) pair derived using an approach that combines small-molecule inhibitors to repress SMAD and WNT signaling pathways to promote CNS fate (*Qi et al., 2017*). This protocol has been extensively validated and is known to robustly generate cortical neurons while actively suppressing glial differentiation. We confirmed the efficiency of differentiation using immunohistochemistry (IHC) and gene expression assays, which indicated the anticipated increase of TUJ1/TBR1-positive derived neurons and downregulation of embryonic stem cell marker *OCT4* (*Figure 1B* and *Figure 1—figure supplement 1E–G*). We identified a number of mature miRNAs dysregulated in response to the 22q11.2 deletion (*Figure 1C*, *Figure 1—figure supplement 2A, B Supplementary file 2*). As a validation of our approach, we observed the expected reduction of expressions of miRNA genes *MIR185*, *MIR1286*, and *MIR1306* that reside in the 22q11.2 locus (expression of three other predicted 22q11.2 miRNA genes, *MIR649*, *MIR3618*, and *MIR4761* were not detected in DIV8 neurons) (*Figure 1—figure supplement 2A*). Among miRNAs located outside the 22q11.2 region, we note downregulation of mature miRNAs such as miR-137, as well as miR-134 and several other members from the largest placental mammal-specific miRNA gene cluster miR379-410 (*Figure 1—figure supplement 2B*) that have been previously implicated in neuronal development, differentiation, and function (*Stark et al., 2008*; *Siegert et al., 2015*; *Thomas et al., 2017*; *Schratt et al., 2006*; *Fiore et al., 2009*; *Rago et al.,*

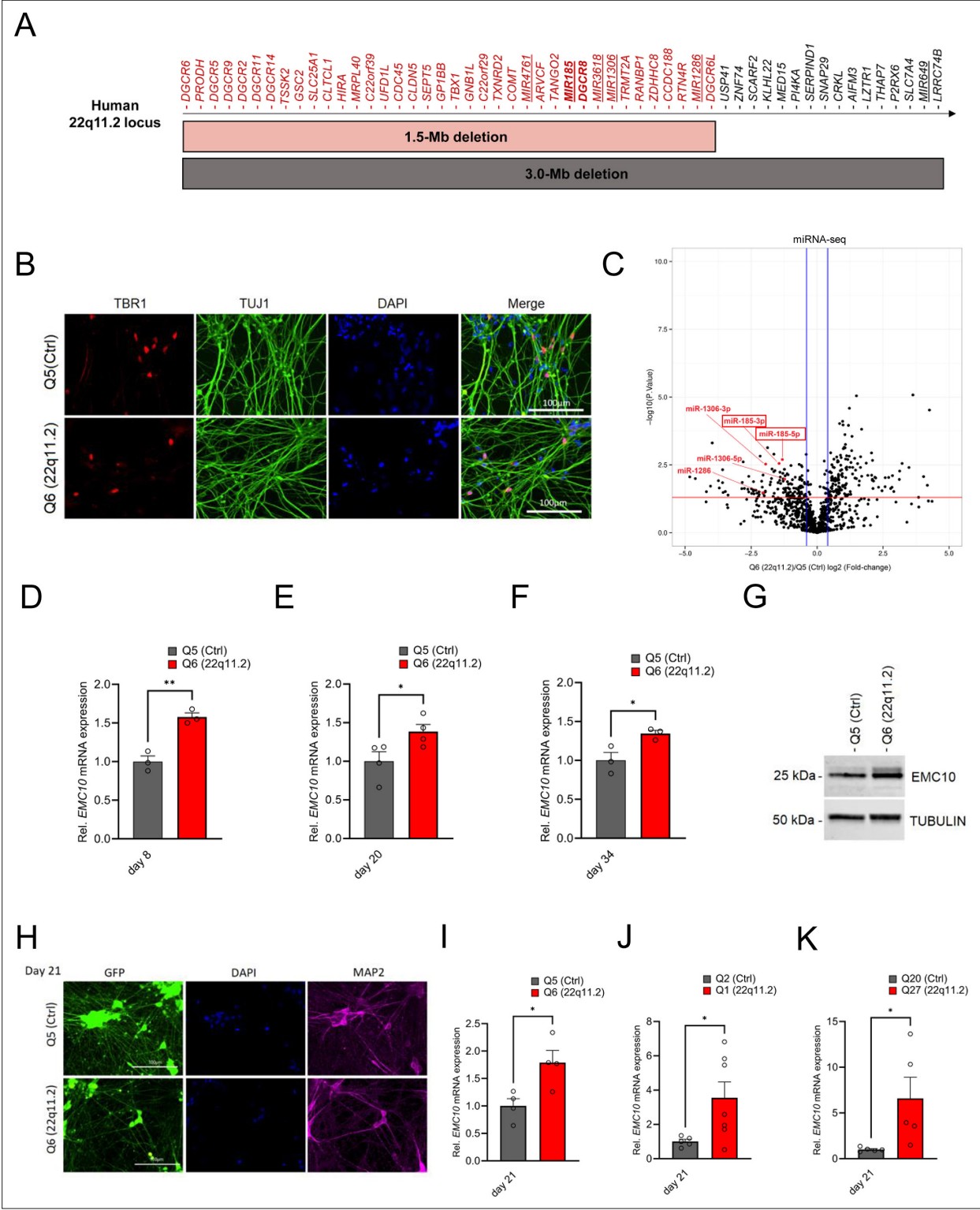

**Figure 1.** EMC10 is robustly upregulated in hiPSC-derived neurons from 22q11.2 deletion carriers. (**A**) Schematic diagram depicting the human chromosome 22q11.2 region. Bright grey and red horizontal bars indicate the two most common hemizygous genomic deletions found in the 22q11.2 Deletion Syndrome. The location of the coding genes and non-coding RNAs (miRNAs, underlined) are shown for chromosome 22q11.2. The microprocessor *DGCR8* (DiGeorge Syndrome Critical Region Gene 8) and *MIR185* are shown in bold. (**B**) Cortical marker TBR1 and pan-neuronal marker TUJ1 expression in cortical neurons as detected by immunocytochemistry at day 13 of differentiation. TBR1 (red), TUJ1 (green) and DAPI (blue) expression are shown. Scale bar: 100 μm. (**C**) Volcano plot showing differentially expressed human mature miRNAs (DEmiRs) in cortical neurons at day

*Figure 1 continued on next page*

*Figure 1 continued*

8 of differentiation. Significant DEmiRs (p-value <5%) are shown above red line; Q5 (Ctrl) n=3, Q6 (22q11.2) n=3. 153/133 miRNAs were significantly up- and downregulated in Q6 (22q11.2) hiPSC-derived cortical neurons, respectively. 22q11.2 deletion region residing miRNAs miR-185, miR-1286 and miR-1306 are highlighted. (**D–F**) Consistent upregulation of *EMC10* mRNA in Q6 (22q11.2) line derived cortical neurons as assayed by qRT-PCR at (**D**) day 8 (p=0.031; Q5: n=3, Q6: n=3), (**E**) day 20 (p=0.0478; Q5: n=4, Q6: n=4) and (**F**) day 34 (p=0.0358; Q5: n=3, Q6: n=3) of differentiation. (**G**) Western blot analysis showing upregulated EMC10 protein levels in Q6 (22q11.2) line derived cortical neurons at day 8 of differentiation. Tubulin was probed as a loading control. (**H**) Immunofluorescence images of NGN2 generated cells. Representative images of NGN2-iNs at DIV21 from Q5 (Ctrl) and Q6 (22q11.2) hiPSC lines identified via EGFP fluorescence and immunostained for neuronal dendrite marker MAP2 and the nuclear marker DAPI. Scale bar = 100 μm. (**I–K**) qRT-PCR assay of *EMC10* mRNA expression level in NGN2-iNs at DIV21. (**I**) Upregulation of *EMC10* mRNA in Q6 (22q11.2) line derived neurons compared to the healthy control line Q5 (p=0.0222). Q5 (Ctrl) n=4, Q6 (22q11.2) n=4. (**J**) Upregulation of *EMC10* mRNA in Q1 (22q11.2) patient line compared to healthy control line Q2 (p0.0441). Q2 (Ctrl) n=5 and Q1 (22q11.2) n=7. (**K**) Upregulation of *EMC10* mRNA in QR27 (22q11.2) patient line compared to healthy control line QR20 (p=0.0414). QR20 (Ctrl) n=5 and QR27 (22q11.2) n=5. Data are presented as mean ± SEM, unpaired two-tailed t-test, *p<0.05, **p<0.01.

The online version of this article includes the following source data and figure supplement(s) for figure 1:

**Source data 1.** PDF file containing original western blots for *Figure 1G*, indicating the relevant bands.

**Source data 2.** Original files for western blot analysis shown in *Figure 1G*.

**Figure supplement 1.** Validation and characterization of hiPSCs and hiPSC-derived neurons.

**Figure supplement 1—source data 1.** PDF file containing original gel for *Figure 1—figure supplement 1G*.

**Figure supplement 1—source data 2.** Original file for hiPSC validation shown in *Figure 1—figure supplement 1G*.

**Figure supplement 2.** Expression profile of selected human mature miRNAs in Q5 (Ctrl) and Q6 (22q11.2) cortical neurons at day 8 of differentiation.

**Figure supplement 3.** GO-term analysis of altered miRNA expression in hiPSC-derived cortical neurons with 22q11.2 deletion.

**Figure supplement 4.** Altered gene expression in hiPSC-derived cortical neurons from 22q11.2 deletion carriers.

**Figure supplement 4—source data 1.** PDF file containing original western blots for *Figure 1—figure supplement 4C*, indicating the relevant bands.

**Figure supplement 4—source data 2.** Original files for western blot analysis shown in *Figure 1—figure supplement 4C*.

*2014*; *Lackinger et al., 2019*; *Gardiner et al., 2012*; *Tomasello et al., 2022*; *Whipple et al., 2020*). We used the miRNA-target interaction network tool miRNet 2.0 (*Chang et al., 2020*) to perform target enrichment and network analysis for the dysregulated miRNAs and conducted GO term enrichment analysis on this target interaction network. Affected biological processes were prominently centered on cell division and intracellular protein transport (*Figure 1—figure supplement 3A*) whereas cellular components were associated with the nucleus and the perinuclear region (endoplasmic reticulum and Golgi apparatus) of the cytoplasm (*Figure 1—figure supplement 3B*).

In addition to 22q11.2 deletion region miRNAs, lower abundance of miRNAs in cases is likely due to haploinsufficiency of the *DGCR8* gene and is expected to result in upregulation of target genes. To identify candidate miRNA target genes, we performed an unbiased evaluation of the transcriptional responses using bulk RNA sequencing on RNA collected from DIV8 differentiated cortical neurons derived from the patient (Q6) and the corresponding healthy dizygotic twin (Q5) line (*Figure 1—figure supplement 4A*). We observed the expected downregulation of genes within the 22q11.2 locus in patient neurons (*Supplementary file 3*). Further, RNA and protein expression characterization confirmed the reductions in the abundance of the 22q11.2 locus residing genes *DGCR8* and *RANBP1* (*Figure 1—figure supplement 4B, C*). Among the differentially expressed genes (DEGs) 2094 were downregulated and 1937 were upregulated. As expected EMC10 expression was elevated in patient neurons while expression of other EMC subunits (*EMC1-4, EMC6-9*) detected in our DIV8 sequencing data did not show significant differences. GO term enrichment analysis on downregulated DEGs identified significantly altered biological processes centered on neurogenesis, neuronal development, and differentiation (*Figure 1—figure supplement 4D*). Among the upregulated DEGs, the GO terms enriched were related to neuronal development as well as neuronal cilia assembly and structure (*Figure 1—figure supplement 4E*).

Intersection of predicted targets of downregulated miRNAs and upregulated DEGs identified 774 predicted targets of downregulated miRNAs (*Figure 1—figure supplement 4F*, *Supplementary file 4*) including *EMC10*. Notably, functional annotation revealed that predicted targets of downregulated miRNAs include genes that modulate neuronal development and are associated with GO terms such as endoplasmic reticulum and endomembrane system of neurons (*Figure 1—figure supplement 4G, H*).

qRT-PCR assays confirmed a robust and significant upregulation of *EMC10* levels in RNA extracted from cortical neurons derived from hiPSCs of the Q5/Q6 pair through SMAD/WNT signaling inhibition, at three distinct stages of in vitro maturation (*Figure 1D–F*). Additionally, this upregulation was confirmed in protein extracts from cortical neurons at day 8 of differentiation (*Figure 1G*). To examine whether transcriptional *EMC10* upregulation is independent of the neuronal derivation method, we generated neurons via inducible expression of Neurogenin-2 (NGN2), a widely used protocol that generates a robust population of excitatory neurons (NGN2-iNs) within 3 weeks (*Zhang et al., 2013*; *Yi et al., 2016*; *Ho et al., 2016*; *Pak et al., 2018*). MAP2 staining was used to demonstrate the successful neuronal differentiation of the hiPSC lines (*Figure 1H*). qRT-PCR assay of *EMC10* mRNA expression level in NGN2-iNs at DIV21 confirmed transcriptional *EMC10* upregulation in three independent pairs of patient and sex-/age matched healthy control lines (*Figure 1I–K*). Taken together our results highlight a reproducible and robust upregulation of *EMC10* in neurons derived from patients with 22q11.2 deletions, which is independent of the derivation method. It is noteworthy that in addition to monolayer cultures, *EMC10* shows significant upregulation along the excitatory neuron lineage (radial glia, intermediate progenitors and excitatory neurons) but not in astrocytes, choroid or interneuron lineage cells, in patient forebrain organoids generated by the same hiPSCs lines used in the present study (*Rao et al., 2023*).

We have previously shown that upregulation of the murine orthologue of *Emc10* is primarily due to downregulation of miR-185 and to a lesser degree of miR-485 (*Xu et al., 2013*). Both conserved and non-conserved binding sites at the 3'UTR of human *EMC10* are predicted in silico for both miRNAs (*Figure 2—figure supplement 1A*). Consistently, the observed upregulation in the levels of *EMC10* gene is accompanied by a robust reciprocal decrease in the levels of the miRNA precursor of miR-185 at DIV8 as indicated both by our miRNA sequencing analysis (*Figure 1C*, *Supplementary file 2*) and follow-up qRT-PCR assays (*Figure 2A*). The miRNA precursor of miR-485 exhibited a modest but non-significant reduction in abundance (*Figure 2B*), consistent with our miRNA sequencing profile (*Figure 1—figure supplement 2B*). This may be attributed to the early developmental stage of the neurons, as miR-485 expression increases during neuronal maturation (*Cohen et al., 2011*; *Soutschek et al., 2023*). Collectively, these findings demonstrate a strong inverse correlation between EMC10 upregulation and miR-185 downregulation, while suggesting that miR-485 may play a less prominent role at this early stage of neuronal development. Notably, overexpression of miR-185 and miR-485 using miRNA mimics in human cortical neurons at DIV10 resulted in a reduction of *EMC10* expression levels in both the healthy control (Q5, *Figure 2C*) and patient line (Q6, *Figure 2D*). Furthermore, inhibition of endogenous miR-185 and miR-485 in the control line by using specific miRNA inhibitors increased *EMC10* expression level (*Figure 2E*) confirming the predicted conserved functionality of miR-185 and miR-485 miRNA binding sites in *EMC10*. It is worth noting that in addition to miR-185, non-conserved binding sites at the 3'UTR of human *EMC10* are predicted in silico for two additional downregulated miRNA genes residing within the 22q11.2 locus, *MIR1286* and *MIR1306* (*Supplementary file 5*). The functionality of these miRNA binding sites in *EMC10* and whether they contribute to the observed elevation of its expression in human neurons remains to be determined. Taken together, our results confirm that miRNA dysregulation emerges in human neurons as a result of the 22q11.2 deletion and in turn drives misexpression of genes primarily involved in intracellular membrane and protein trafficking-related processes required for neuronal development and maturation. Among them, *EMC10* represents a major downstream effector of the 22q11.2-linked miRNA dysregulation.

To investigate the relevance of *EMC10* de-repression in the development and function of patient neurons, we generated derivatives of the Q6 patient hiPSC line carrying either heterozygous (Q6/EMC10[HET]) or homozygous (Q6/EMC10[HOM]) *EMC10* LoF mutations using standard CRISPR/Cas9 editing approaches (*Figure 3—figure supplement 1A*). Mutations were confirmed by sequencing (*Figure 3—figure supplement 1A*, lower panel) and karyotyping confirmed normal chromosome complement (*Figure 3—figure supplement 1B*). We confirmed reduced expression levels of 22q11.2 gene *RANBP1* by western blot in both derivative hiPSC lines (*Figure 3—figure supplement 1C*) whereas stem-cell markers *NANOG* and *OCT4* were equally expressed in all lines assayed by qRT-PCR (*Figure 3—figure supplement 1D, E*). EMC10 mRNA and protein levels were reduced by ~50% in the Q6/EMC10[HET] hiPSC line and abolished in the Q6/EMC10[HOM] line (*Figure 3—figure supplement 1F, G*). It is noteworthy that we did not observe an upregulation of *EMC10* mRNA levels in the Q6 hiPSC lines (*Figure 3—figure supplement 1F*), a finding likely attributed to the general low expression

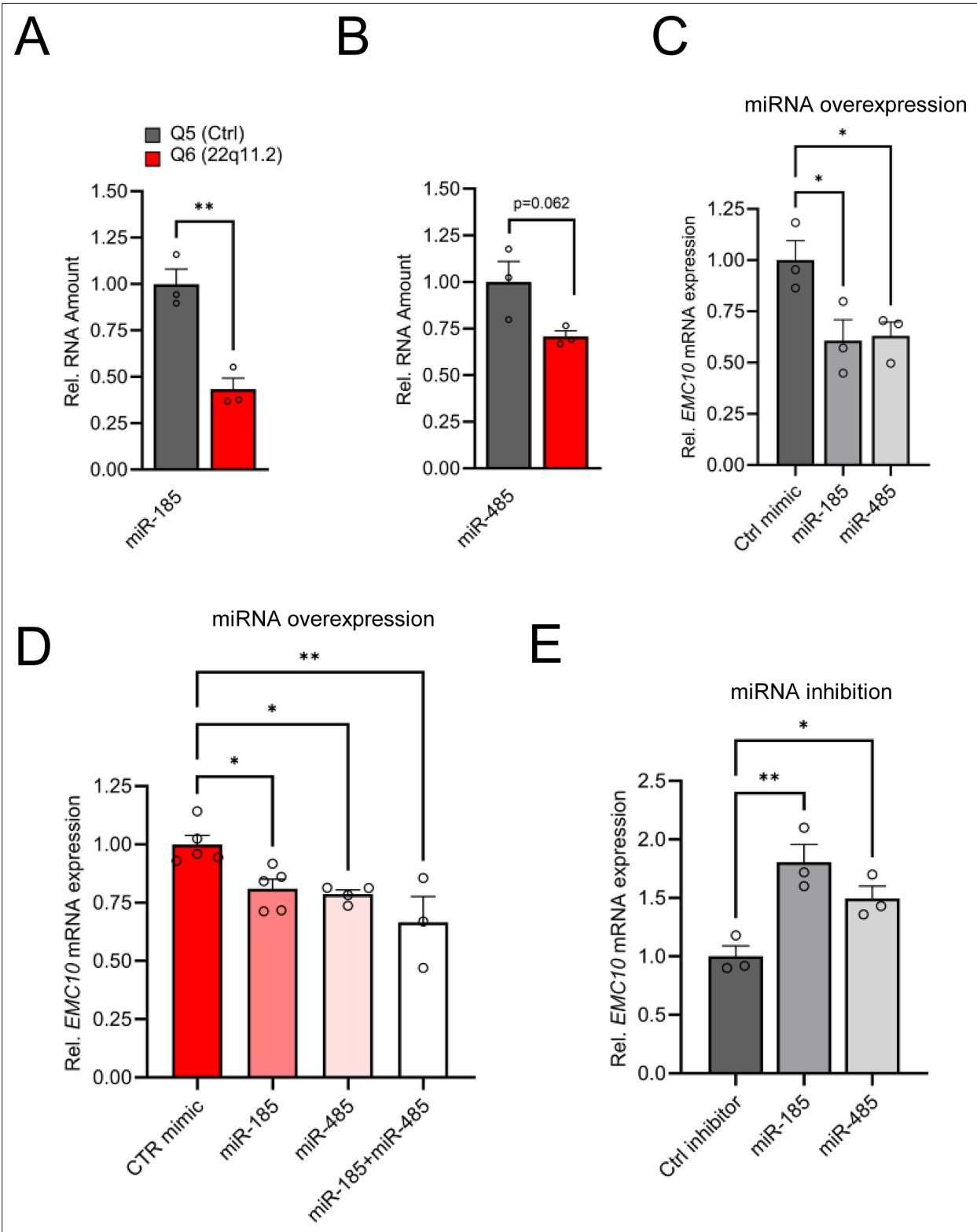

**Figure 2.** Altered miRNA expression in hiPSC-derived cortical neurons from 22q11.2 deletion carriers. (**A**) Precursor miRNA expression level of miR-185 (p=0.0038), predicted to target *EMC10*, are downregulated in Q6 (22q11.2) cortical neurons as assayed by qRT-PCR (Q5: n=3, Q6: n=3). (**B**) Precursor miRNA expression level of miR-485 (p=0.0622), predicted to target *EMC10*, are downregulated in Q6 (22q11.2) cortical neurons as assayed by qRT-PCR (Q5: n=3, Q6: n=3). (**C–E**) miR-185 and miR-485 modulate *EMC10* in human iPSC-derived cortical neurons. (**C**) qRT-PCR quantification shows reduced expression levels of *EMC10* mRNA in Q5 (Ctrl) line derived cortical neurons at day 10 of differentiation transfected with miR-185 [one-way ANOVA, F

*Figure 2 continued on next page*

*Figure 2 continued*

(2, 6)=6.079, p=0.0361; post hoc Bonferroni, p=0.0366] or miR-485 [post hoc Bonferroni, p=0.0464] mimics at day 8 of differentiation. Expression levels in miR-185 or miR-485 mimic-treated neurons were normalized to expression levels under scramble mimic controls treatment (n=3, each treatment). (**D**) qRT-PCR quantification shows reduced expression levels of *EMC10* mRNA in Q6 line-derived cortical neurons transfected with miR-185 [one-way ANOVA, F (3, 13)=7.167, p=0.0044; post hoc Tukey, p=0.0345] or miR-485 [post hoc Tukey, p=0.0251] or a combination of both miRNA mimics [post hoc Tukey, p=0.0020]. Expression levels in miR-185, miR-485 or the combination of both mimic-treated neurons were normalized to expression levels under scramble mimic controls treatment. Ctrl mimic n=5, miR-185 mimic n=5, miR-485 mimic n=4 and miR-185 +miR-485 mimics n=3. (**E**) qRT-PCR quantification shows increased expression levels of *EMC10* mRNA in Q5 line derived cortical neurons transfected with miRNA inhibitors miR-185 [one-way ANOVA, F (2, 6)=11.94, p=0.0081; post hoc Bonferroni, p=0.0057]or miR-485 [post hoc Bonferroni, p=0.0491] at day 8 of differentiation. Expression levels in miR-185 or miR-485 inhibitor-treated neurons were normalized to expression levels under scramble miRNA inhibitor controls treatment (n=3, each treatment). Data are presented as mean ± SEM, unpaired two-tailed t-test or one-way ANOVA as indicated, *p<0.05, **p<0.01.

The online version of this article includes the following figure supplement(s) for figure 2:

**Figure supplement 1.** Predicted miRNA targets.

level of miR-185 and miR-485 in hiPSCs (*Wilson et al., 2009*). Indeed both miRNAs are developmentally regulated and show increased expression levels during neuronal development (https://ethz-ins.org/igNeuronsTimeCourse/) (*Soutschek et al., 2023*). Additional characterization of hiPSC-derived NGN2-iNs (*Figure 3A*), conclusively demonstrated a reduction (Q6/EMC10[HET]) or elimination (Q6/EMC10[HOM]) of *EMC10* mRNA (*Figure 3B*). Expression assays of a panel of cell type-specific markers did not reveal significant differences between NGN2-iNs from the Q6 patient line and both derivative lines, indicating that gene editing has no adverse effect on neuronal differentiation (*Figure 3—figure supplement 1H*).

*Df(16)A[+/−]*mice show impaired formation of dendrites in deep layer cortical neurons, which are faithfully recapitulated in primary neuronal cultures and are partially reversed by reduction of *Emc10* levels (*Xu et al., 2013*). We asked whether impaired dendritic formation is also observed in human neuronal cultures from patients with 22q11.2 deletions and whether reduction of *EMC10* levels could prevent such morphological alterations during neuronal maturation. We employed monolayer neuronal cultures of NGN2-iNs. Neuronal cells were fixed at DIV21 of differentiation, immuno-stained, traced and key indices of dendritic architecture were quantified (see Materials and methods). Our analysis confirmed a reduced dendritic complexity in mutant neurons as reflected in total neuronal length, the number of branch points and the total number of dendrites per cell (*Figure 3C–F*). The number of primary dendrites per cell was unchanged (*Figure 3G*) in accordance with previous findings from the murine 22q11.2 deletion model where only subtle changes were detected in the number of primary neurons (*Xu et al., 2013*). Importantly, we found that reduction or elimination of *EMC10 expression* restored to WT levels neuronal length and branch points. Importantly, we found that reducing or eliminating EMC10 expression restored neuronal length and branch points to WT levels. Notably, the number of branch points in Q6/EMC10[HOM] neurons exceeded those in WT neurons (*Figure 3E*) likely suggesting that reduced (or abolished) *Emc10* expression can alter normal neurite growth, resulting in excessive responses, potentially triggered upon gene restoration by the mutant system's adaptation to dysfunction, leading to altered receptor sensitivity or signaling dynamics. This highlights the critical importance of precise *Emc10* expression for maintaining proper neuronal development and function.

Our previous evaluation of Ca[2+] homeostasis perturbations caused by 22q11.2 deletions using Ca[2+] imaging on primary neurons from *Df(16)A[+/−]*mice revealed a significantly lower amplitude of Ca[2+] elevation following KCl evoked depolarization (*Sun et al., 2018*). This impairment was replicated in human cortical neurons from patients with 22q11.2 deletions (*Khan et al., 2020*) and shown to be partially restored by exogenous expression of *DGCR8*, indicating a potential role of miRNA dysregulation. Using the green-fluorescent calcium indicator Fluo-4 and time-lapse microscopy, we confirmed a decrease in the amplitude of Ca[2+] rise following KCl evoked depolarization, in patient (Q6) derived NGN2-iNs at DIV37/38 compared to the healthy twin (Q5) (*Figure 3—figure supplement 2A, B*). We asked whether reduction of EMC10 levels could reverse such alterations. Notably, the observed defect in Ca[2+] signaling were reversed in both Q6/EMC10[HET] and Q6/EMC10[HOM] NGN2-iNs as demonstrated by the increased amplitude of Ca[2+] rise following depolarization (*Figure 3H, I*). Interestingly, the amplitudes of Ca[2+] rise in Q6/EMC10[HET] and Q6/EMC10[HOM] were slightly elevated compared to the WT control group, consistent with the effects on neurite outgrowth (*Figure 3H*). The observation that reduction of *EMC10* levels fully restores the Ca[2+] signaling deficits observed in patient neurons

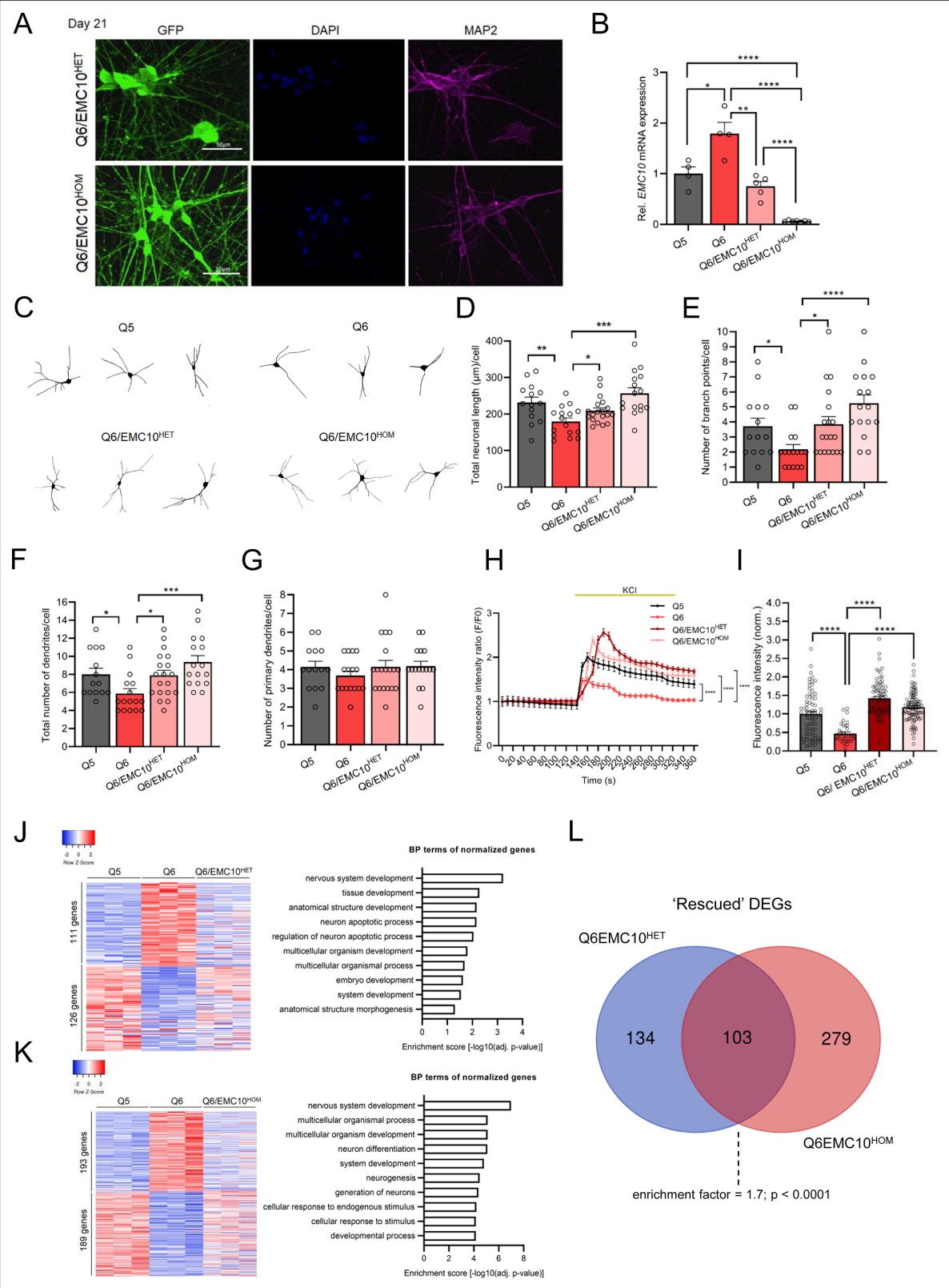

**Figure 3.** Reduction of *EMC10 levels* restores defects in neurite outgrowth and calcium signaling in neurons from 22q11.2 deletion carriers. (**A**) Representative images of NGN2-iNs at DIV21 from Q6/EMC10[HET] and Q6/EMC10[HOM] hiPSC lines identified via EGFP fluorescence and immunostained for neuronal dendrite marker MAP2 and the nuclear marker DAPI. Scale bar = 50 μm. (**B**) qRT-PCR assay of *EMC10* mRNA expression level in NGN2-iNs at DIV21. *EMC10* expression is normalized to near WT levels level in Q6/EMC10[HET] line (p=0.166) and abolished in Q6/EMC10[HOM] line (p<0.0001). Q5

*Figure 3 continued on next page*

*Figure 3 continued*

(Ctrl) n=4, Q6 (22q11.2/SCZ) n=4, Q6/EMC10^HET^ n=5 and Q6/EMC10^HOM^ n=7. (**C–G**) Neuronal morphology analysis in Q5, Q6 (22q11.2), Q6/EMC10^HET^ and Q6/EMC10^HOM^ neurons. (**C**) Representative images of traced neurons. (**D**) Total neuronal length is reduced in Q6 line (p=0.0044) and restored in Q6/ EMC10^HET^ (p=0.0253) and Q6/EMC10^HOM^ line (p=0.0001). (**E**) Reduction in number of branch points/cell in Q6 (p=0.0195) is restored in the Q6/ EMC10^HET^ (p=0.0134) and Q6/EMC10^HOM^ lines (p<0.0001). (**F**) Reduction in the total number of dendrites/cells in Q6 (p=0.0202) is reversed in the Q6/ EMC10^HET^ (p=0.0166) and Q6/EMC10^HOM^ lines (p=0.0005) (**G**) The number of primary dendrites per cell is unchanged. Q5 (Ctrl) n=14, Q6 (22q11.2) n=16, Q6/EMC10^HET^ n=19 and Q6/EMC10^HOM^ n=16 neuronal cells. (**H–I**) Defects in cytoplasmic calcium signaling in Q6 (22q11.2) neurons are reversed in Q6/ EMC10^HET^ and Q6/EMC10^HOM^ lines. (**H**) Changes in Fluo4-AM fluorescence signal intensity in response to 75 mM KCl in Q5 (Ctrl), Q6 (22q11.2). Q6/ EMC10^HET^ and Q6/EMC10^HOM^ hiPSC-derived neurons at DIV38. Q5 vs. Q6 (KS D=0.5405, p<0.0001), Q6 vs. Q6 EMC10^HET^ (KS D=0.5556, p<0.0001) and Q6 vs. Q6 EMC10^HOM^ (KS D=0.5676, p<0.0001). (**I**) Quantification of KCl-induced Fluo4 intensity peak amplitude (ΔF) demonstrates a reduction in Q6 line (p<0.0001) that is reversed in Q6/EMC10^HET^ (p<0.0001) and Q6/EMC10^HOM^ lines (p<0.0001). Q5 (Ctrl) n=70, Q6 (22q11.2/SCZ) n=37, Q6/EMC10^HET^ n=82, Q6/EMC10^HOM^ n=97 neuronal cells. (**J**) Heatmap (left) showing the expression of differentially regulated 237 genes in Q5 (Ctrl) and Q6 (22q11.2) that are normalized in the Q6/EMC10^HET^ NGN2-iNs at DIV21 (n=3 per genotype). Gene Ontology (GO) biological process (BP) terms (right) associated with the up- and downregulated genes normalized in the Q6/EMC10^HET^ NGN2-iNs. (**K**) Heatmap (left) showing the expression of differentially regulated 382 genes in Q5 (Ctrl) and Q6 (22q11.2) that are normalized in the Q6/EMC10^HOM^ NGN2-iNs at DIV21 (n=3 per genotype). Gene Ontology (GO) biological process (BP) terms (right) associated with the up- and downregulated genes normalized in the Q6/EMC10^HOM^ NGN2-iNs. (**L**) Intersection of rescued up- and downregulated DEGs in the Q6/ EMC10^HET^ and Q6/EMC10^HOM^ lines: Venn diagram highlighting the 103 rescued DEGs (enrichment factor = 1.7; p<0.0001, based on a hypergeometric test). Data are presented as mean ± SEM, unpaired two-tailed t-test or Kolmogorov–Smirnov test as indicated, *p<0.05, **p<0.01, ***p<0.001, ****p<0.0001.

The online version of this article includes the following source data and figure supplement(s) for figure 3:

**Figure supplement 1.** Characterization of hiPSC lines carrying an *EMC10* LoF mutation.

**Figure supplement 1—source data 1.** PDF file containing original western blots for *Figure 3—figure supplement 1C* and G, indicating the relevant bands.

**Figure supplement 1—source data 2.** Original files for western blot analysis shown in *Figure 3—figure supplement 1C and G*.

**Figure supplement 2.** Defects in cytoplasmic calcium signaling in Q6 (22q11.2) neurons.

**Figure supplement 3.** PPI network analysis of the 103 shared DEGs normalized in both Q6/EMC10^HET^ and Q6/EMC10^HOM^ NGN2-iNs.

suggests that miRNA-dependent elevation of *EMC10* may interfere with one or more sources of intracellular Ca^2+^ and a wide range of calcium-dependent processes.

DEGs are often organized into functional groups or pathways based on their known biological roles. We used transcriptional profiling as an indirect measure of cellular pathways affected by the reduction of EMC10 levels by identifying genes differentially expressed between the parental Q6/ Q5 lines whose expression differences are abolished or nearly abolished ('rescued') in either Q6/ EMC10^HET^ or Q6/EMC10^HOM^ NGN2-iNs (*Figure 3J and K*). In the Q6/EMC10^HET^ line, 237 DEGs (6%) were rescued (111 downregulated and 126 upregulated), while in the Q6/EMC10^HOM^ line, 382 DEGs (11%) were rescued (193 downregulated and 189 upregulated; *Supplementary file 6*). In both cases, functional annotation analysis indicated highest enrichment scores for terms related to nervous system development as well as an enrichment in GO terms relevant to neuronal generation and differentiation. Intersection of "rescued" genes in Q6/EMC10^HET^ and Q6/EMC10^HOM^ NGN2-iNs identified 103 shared DEGs (*Figure 3L* and *Supplementary file 7*). To assess the significance of the overlap between the rescued DEGs in Q6/EMC10^HET^ and Q6/EMC10^HOM^ NGN2-iNs, we performed a hypergeometric test, which calculates the probability of observing the degree of overlap between two gene groups under the null hypothesis that the overlap occurs by chance. Our analysis revealed a significant overlap (enrichment factor = 1.7; p<0.0001), indicating that the overlap is much greater than expected by chance. These 103 shared DEGs likely play a key role in pathways influenced by EMC10 levels, particularly those involved in nervous system development, as suggested by our functional annotation analysis. To further investigate the functional relationships among these shared DEGs, we conducted a protein-protein interaction (PPI) network analysis. This analysis highlighted a functional cluster including 30 of these genes, such as the SCZ-linked genes *PCDHA2* (*Shao et al., 2019*), *RBFOX1* (*O'Leary et al., 2022*) and *RGS4* (*Mirnics et al., 2001*; *Erdely et al., 2006*; *Figure 3— figure supplement 3A*), involved in nervous system development (*Figure 3—figure supplement 3B*). It should be noted that the beneficial effect of elimination of EMC10 expression is consistent with previous findings that lack of EMC10 does not compromise EMC assembly (*Volkmar et al., 2019*) and implying an auxiliary or modulatory role of EMC10 in the EMC function.

Taken together, our analysis of neurons from 22q11.2 deletion carriers indicate that elevation of *EMC10* expression disrupts their development and maturation in a way similar to observations in murine neurons, and support normalization of *EMC10* expression as a disease-modifying intervention. While our previous work has shown that constitutive genetic reduction of *Emc10* levels rescues key cognitive and behavioral alterations in the *Df(16)A*$^{+/-}$mice, translating these observations into therapeutic interventions requires demonstration that it is the sustained elevation of *EMC10* throughout the adult life that interferes with the underlying neural processes rather than an irreversible impact on brain maturation during early development. Toward this end, we first investigated whether restoration of *Emc10* levels in the brain of adult (2–4 month-old) *Df(16)A*$^{+/-}$mice is effective at reversing cognitive alterations (*Stark et al., 2008*; *Diamantopoulou et al., 2017*). Specifically, we examined the effects of *Emc10* reduction in adult brain on social memory (SM), a cognitive domain robustly and reproducibly affected in adult *Df(16)A*$^{+/-}$mice (*Piskorowski et al., 2016*; *Diamantopoulou et al., 2017*; *Donegan et al., 2020*). Notably, SM deficits are also present in juvenile *Df(16)A*$^{+/-}$mice as early as postnatal day 22 (*Figure 4—figure supplement 1A, B*), underscoring the severity of this phenotype, which emerges during early adulthood. In humans, SM, a key component of social cognition, involves encoding, storing, and retrieving information about social experiences, such as recognizing familiar individuals and recalling past interactions and emotions. Social cognition, the broader ability to perceive, interpret, and respond to social cues, is essential for forming relationships and understanding others' behavior. Disruptions in SM can impair social cognition, contributing to the functional deficits commonly observed in schizophrenia (*Green et al., 2015*). Deficits in social cognition are present in individuals with 22q11.2 deletions (*Jalal et al., 2021*) and use of rodent tasks that evaluate SM can serve as a useful proxy of the human condition. Impaired SM in *Df(16)A*$^{+/-}$mice is fully restored by constitutive genetic reduction of *Emc10* levels (*Diamantopoulou et al., 2017*).

To manipulate the expression of the *Emc10* gene in adult *Df(16)A*$^{+/-}$mice, we used an *Emc10* conditional 'knockout-first' design by conducting a Flp- and Cre-dependent genetic switch strategy (*Figure 4—figure supplement 2A*). Parental *Emc10*$^{+/- tm1a}$ mice were crossed to a germline Flp mouse line that activates global Flp function and leads to the deletion of the frt-flanked sequence(s) in the offspring. The *Emc10*$^{tm1c}$ offspring from this cross carry a *loxP* flanked WT *Emc10* allele and are essentially WT. To achieve temporal control of *Emc10* expression, we used an inducible UBC-Cre/ERT2 mouse line that activates global Cre function upon tamoxifen (TAM) treatment. This approach enables postnatal normalization of *Emc10* expression at its endogenous locus preserving *Emc10* expression within its physiological levels. We used UBC-Cre/ERT2 mice in crosses to generate compound *Emc10*$^{tm1c+/-}$; UBC-cre/ERT2; *Df(16)A*$^{+/-}$mice. These mice have two WT *Emc10* copies upregulated, as expected in the *Df(16)A* background, until TAM-induced Cre expression deletes the tagged *Emc10* allele. We used oral gavage to deliver TAM and implement Cre-mediated *Emc10* deletion during adulthood (postnatal day 56–70). Corn oil treatment served as a control. Behavioral analysis was performed on the following four groups: *Emc10*$^{tm1c+/-}$; UBC-cre/ERT2; *Df(16)A*$^{+/+}$ mice treated with TAM (WT +TAM), *Emc10*$^{tm1c+/-}$; UBC-cre/ERT2; *Df(16)A*$^{+/-}$mice treated with TAM (*Df(16)A*$^{+/-}$+TAM), *Emc10*$^{tm1c+/-}$; UBC-cre/ERT2; *Df(16)A*$^{+/+}$ mice treated with corn oil vehicle (WT +oil), and *Emc10*$^{tm1c+/-}$; UBC-cre/ERT2; *Df(16)A*$^{+/-}$mice treated with corn oil vehicle (*Df(16)A*$^{+/-}$+oil). Investigation of the efficiency of Cre-mediated deletion in brain lysate preparations from prefrontal cortex (PFC) (*Figure 4—figure supplement 2B, C*), hippocampus (HPC) (*Figure 4—figure supplement 2D, E*) and cerebellum (CB; *Figure 4—figure supplement 2F*) confirmed that upon TAM treatment, *Emc10* mRNA and protein levels were restored to near WT levels in the adult brain of *Df(16)A*$^{+/-}$mice. As expected, *Df(16)A*$^{+/-}$+oil mice showed impaired SM performance compared to WT +oil control littermates, which was fully rescued upon TAM treatment. Specifically, upon reintroduction of a familiar juvenile mouse *Df(16)A*$^{+/-}$+TAM mice showed a strong reduction in social interaction, indicative of intact SM, comparable to TAM-treated WT littermates and significantly different from *Df(16)A*$^{+/-}$mice treated with corn oil (*Figure 4—figure supplement 2G*). The intact SM of the *Df(16)A*$^{+/-}$+TAM mice was further evident in analysis of difference score (*Figure 4—figure supplement 2H*) compared to the corn oil-treated *Df(16)A*$^{+/-}$mice. Interaction times during the first trial of the SM assay, which measures general social interest, were unaffected by TAM treatment. In contrast to SM, *Df(16)A*$^{+/-}$mice hyperactivity in the open field was not affected upon TAM treatment consistent with our previous results from constitutive genetic reduction of *Emc10* levels (*Figure 4—figure supplement 2I*). Notably, TAM treatment did not alter the time spent in the center area of the open field, indicating an absence of changes in anxiety-related behavior. These findings

demonstrate that restoring Emc10 levels in adult *Df(16)A⁺ᐟ⁻* mice can significantly improve cognitive deficits, underscoring a broad therapeutic window and establishing Emc10 as a promising target for postnatal interventions.

We explored the translation potential of this finding by employing transient injection of single-stranded ASOs targeting the mouse gene as dictated by their demonstrated efficacy as a therapeutic modality in preclinical models (*Becker et al., 2017*; *DeVos et al., 2017*; *Kordasiewicz et al., 2012*; *Korobeynikov et al., 2022*; *Scoles et al., 2017*; *Sztainberg et al., 2015*) and clinical studies of neuro-developmental disorders (NDDs) or neurodegenerative disorders (*Tabrizi et al., 2019*; *Winkelsas and Fischbeck, 2020*; *Rinaldi and Wood, 2018*). Over 300 chimeric 2'-O-methoxyethyl (2'MOE)/ DNA gapmer ASOs were generated and screened for *Emc10* mRNA reduction in 4T1 cells via electroporation (*Figure 4—figure supplement 3A*). Lead ASOs were then confirmed in a dose-response assay (*Figure 4—figure supplement 3B*), including the lead ASO (1081815, herein referred to as Emc10^ASO1^), which targets intron 2 of mouse *Emc10* (*Figure 4A*). Emc10^ASO1^ was selected for subsequent studies, as it was effective in lowering *Emc10* expression both in vitro and in vivo. Specifically, following transient intracerebral ventricular (ICV) injection in the posterior ventricle of 8 weeks old WT mice, Emc10^ASO1^ effectively suppressed the levels of *Emc10* mRNA (*Figure 4—figure supplement 3C*) and protein (*Figure 4—figure supplement 3D*) in both left and right HPC compared to WT mice treated with a control ASO (Ctrl^ASO1^) without complementarity in the mouse. Analysis of *Gfap* and *Aif1* expression did not reveal any changes (*Figure 4—figure supplement 3E, F*) suggesting lack of astroglial and microglial activation upon Emc10^ASO1^ injection. Emc10^ASO1^ injected mice showed normal gait and no signs of behavioral toxicity. IHC analysis using an antibody that selectively recognizes the phosphorothioate backbone verified a robust diffusion primarily in HPC and to a lesser degree in surrounding brain areas. Colocalization with the neuronal marker NeuN and glial fibrillary acidic protein (GFAP) confirmed accumulation in hippocampal neurons as well as GFAP-expressing astrocytes (*Figure 4B*). Analysis of *Df(16)A⁺ᐟ⁻* mice treated by intraventricular injection at 8 weeks of age showed that Emc10^ASO1^ effectively lowered hippocampal *Emc10* mRNA to nearly WT levels 3 weeks post-injection resulting in normalization of *Emc10* expression (*Figure 4C*, left panel). By contrast, consistent with the pattern of ASO distribution, we did not observe a significant reduction of *Emc10* expression levels in the PFC of *Df(16)⁺ᐟ⁻* mice treated with Emc10^ASO1^ (*Figure 4C*, right panel). In addition to targeted assays, we performed bulk RNA-sequencing analysis of Ctrl^ASO1^ and Emc10^ASO1^-treated *Df(16)A⁺ᐟ⁻* mice and WT littermates to evaluate the effect of Emc10^ASO1^ treatment on the hippocampal transcriptome profile. In the Ctrl^ASO1^-treated group (*Figure 4D*, left panel), we observe the tripartite differential gene expression signature characteristic of *Df(16)A⁺ᐟ⁻* mice: upregulation of *Emc10* and non-coding RNAs (pri-forms of miRNAs and long non-coding RNAs (*Figure 4D*, left panel) and inset, Log2Fold Change = 0.5) as well as the expected downregulation of genes included within the *Df(16)A* deficiency. In the Emc10^ASO1^-treated group (*Figure 4D*, right panel and inset), *Emc10* is no longer upregulated in *Df(16)A⁺ᐟ⁻* mice while non-coding RNAs remain upregulated, and genes included in the deficiency are robustly downregulated. Apart from *Emc10*, seven other genes (*Mir9-3hg*, *Plxnd1*, *Cd68*, *Mir22hg*, *Gm28439*, *Adgre1*, and *Tnn*) are significantly upregulated in *Df(16) A⁺ᐟ⁻* mice in the Ctrl^ASO1^ but not in the Emc10^ASO1^-treated group (*Supplementary file 8*). We used the Bowtie mapping tool (*Langmead et al., 2009*) to align short sequencing reads on both genomic and transcript sequence to assess whether these expression changes represent potential off-target effects of the Emc10^ASO1^ in the mouse transcriptome. Emc10^ASO1^ exclusively aligned with full complementarity to an intronic region in the *Emc10* gene (*Figure 4A*) providing additional support for high target specificity. The observed changes might represent downstream effects of *Emc10* level reduction or reflect expression variability due to low expression levels of the upregulated genes.

Eight-week-old *Df(16)A⁺ᐟ⁻* mice and WT littermates were treated by ICV injection of Emc10^ASO1^ and Ctrl^ASO1^ and tested 3 weeks later in SM assays. *Df(16)A⁺ᐟ⁻* mice treated with Ctrl^ASO1^ showed the expected deficits in SM as reflected in the sustained high interaction time with the reintroduced familiar juvenile mouse. By contrast, *Df(16)A⁺ᐟ⁻* mice injected with Emc10^ASO1^ had significantly improved memory performance to levels indistinguishable from Emc10^ASO1^-treated WT littermates, consistent with improvement of function arising from adult restoration of *Emc10* levels (*Figure 4E and F*). Interaction times during the first trial of the SM assay, which measures general social interest, were unaffected by ASO treatment. Rescue was observed in both sexes and no significant differences were seen in treatment across sexes. In control experiments, we did not observe any effects of genotype or

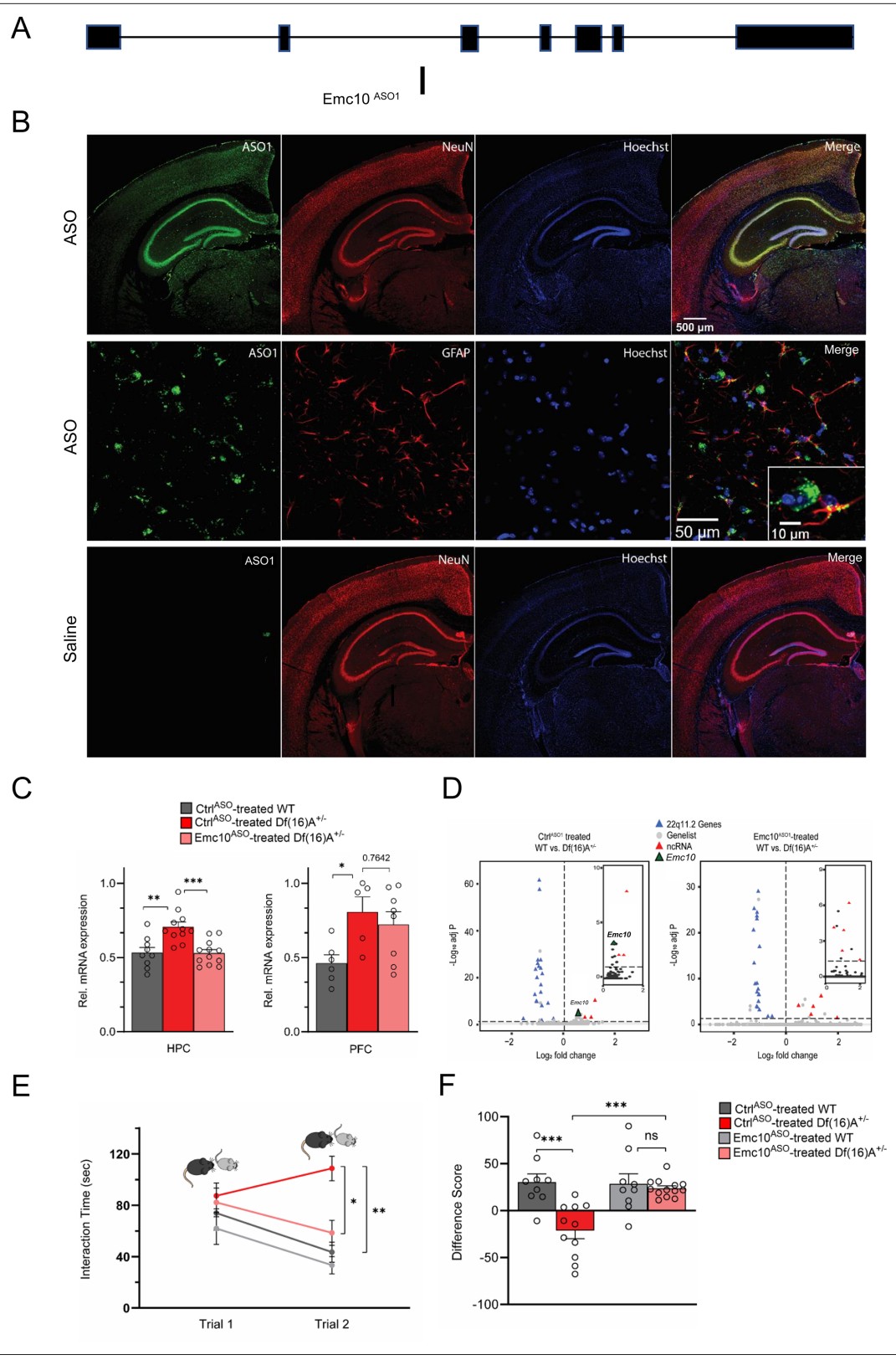

Figure 4. ASO-mediated suppression of murine *Emc10* in vivo. (**A**) Mouse *Emc10* gene map plot (ENSMUST00000118808) showing the *Emc10*[ASO1] target site. (**B**) Mouse brains collected 3 weeks post ICV injection were stained with an ASO antibody (green), counterstained with neuronal marker NeuN (red) and nuclear stain Hoechst (blue). A robust and uniform ASO diffusion (top panel) is observed in the HPC. No signal is detected in

*Figure 4 continued on next page*

*Figure 4 continued*

saline injected mice (bottom panel). Overlap with NeuN (yellow, top-right panel) confirms presence in neuronal cells. Accumulation in glial cells, specifically GFAP-labeled astrocytes is also observed (middle-right panel and inset; ASO in green, GFAP in red, and Hoechst in blue). Images are taken with 4 x, 20 x and 40 x objectives. (**C**) qRT-PCR analysis shows Emc10$^{ASO}$-mediated normalization of *Emc10* mRNA levels in the HPC of *Df(16)A$^{+/-}$*mice (left panel). Significant upregulation of *Emc10* mRNA expression levels is seen in Ctrl$^{ASO}$-treated-*Df(16)A$^{+/-}$*compared to WT mice one-way ANOVA, $F_{(2, 29)}=11.65$, $p<0.001$; post hoc Tukey, $p=0.001$. Following ASO treatment, *Emc10* expression is normalized to WT levels in Emc10$^{ASO}$-treated-*Df(16)A$^{+/-}$*compared to Ctrl$^{ASO}$-treated-*Df(16)A$^{+/-}$*mice (post hoc Tukey, $p=<0.001$). Ctrl$^{ASO}$-treated WT mice: n=9 (5 males, 4 females), Ctrl$^{ASO}$-treated *Df(16)A$^{+/}$* mice: n=11 (5 males, 6 females), and Emc10$^{ASO}$-treated *Df(16)A$^{+/}$* mice: n=12 (7 males, 5 females). qRT-PCR analysis shows a significant upregulation of *Emc10* mRNA expression levels in the PFC of Ctrl$^{ASO}$-treated-*Df(16)A$^{+/}$* (right panel) compared to WT mice one-way ANOVA, $F_{(2, 16)}=4.253$, $p=0.0330$; post hoc Tukey, $p=0.0385$. ASO injection does not normalize *Emc10* expression levels in PFC of *Df(16)A$^{+/}$* mice injected with Emc10$^{ASO}$ (post hoc Tukey vs Ctrl$^{ASO}$ treated *Df(16)A$^{+/}$* mice, $p=0.7642$). Ctrl$^{ASO}$-treated-WT male mice: n=6, Ctrl$^{ASO}$-treated-*Df(16)A$^{+/-}$*male mice: n=5, and Emc10$^{ASO}$-treated-*Df(16)A$^{+/}$* male mice: n=8. (**D**) Volcano plots showing upregulation of *Emc10* expression in the Ctrl$^{ASO}$ treated-*Df(16)A$^{+/-}$*compared to the Ctrl$^{ASO}$-treated WT mice (left panel and inset) but not in the Emc10$^{ASO}$ treated-*Df(16)A$^{+/-}$*compared to the Emc10$^{ASO}$-treated WT mice (right panel and inset). The expected down-regulation of genes included in the *Df(16)A$^{+/-}$*deletion (blue) and upregulation of non-coding RNAs (ncRNAs, red) is also evident in both panels. Downregulated genes from the 22q11.2 locus (*Dgcr8*, *Ranbp1*, and *Tango2*) as well as the upregulated ncRNA (miRNA-containing) gene *Mirg*, are highlighted. Ctrl$^{ASO}$-treated-*Df(16)A$^{+/-}$*males: n=5, Emc10$^{ASO}$-treated-*Df(16)A$^{+/-}$*males: n=4, Ctrl$^{ASO}$-treated WT males: n=4, and Emc10$^{ASO}$-treated-WT males: n=4. (**E–F**) ASO-mediated behavioral rescue of SM deficit in *Df(16)A$^{+/-}$*mice. (**E**) Ctrl$^{ASO}$-treated *Df(16)A$^{+/-}$*mice show a robust SM deficit compared to Ctrl$^{ASO}$-treated WT mice as indicated by the significant difference in trial 2 interaction time upon reintroduction of a familiar juvenile mouse [three-way ANOVA for Trial X Genotype X Treatment Interaction matching by trial: $F_{(1, 38)}=9.393$ $p=0.0040$; post hoc Tukey, $p=0.0012$]. A reduction in trial 2 interaction time indicates rescue of the SM deficit in Emc10$^{ASO}$-treated *Df(16)A$^{+/-}$*compared to Ctrl$^{ASO}$-treated *Df(16)A$^{+/-}$*mice (post hoc Tukey, $p=0.0114$). (**F**) A negative difference score (trial 1- trial 2) confirms the SM deficit in Ctrl$^{ASO}$-treated adult *Df(16)A$^{+/-}$*mice compared to WT littermates two-way ANOVA for Genotype X Treatment interaction: $F_{(1, 38)}=9.369$, $p=0.0040$; post hoc Tukey, $p=0.0002$. Increase in the difference score of *Df(16)A$^{+/-}$*mice in the Emc10$^{ASO}$- vs Ctrl$^{ASO}$-treated group demonstrates SM rescue (post hoc Tukey, $p=0.0004$). Ctrl$^{ASO}$-treated WT mice: n=9 (5 males, 4 females), Ctrl$^{ASO}$-treated *Df(16)A$^{+/}$* mice: n=11 (5 males, 6 females), Emc10$^{ASO}$-treated WT mice: n=9 (5 males, 4 females), and Emc10$^{ASO}$-treated *Df(16)A$^{+/}$* mice: n=13 (7 males, 6 females). Data are presented as mean ± SEM, $*p<0.05$; $**p<0.01$; $***p<0.001$.

The online version of this article includes the following source data and figure supplement(s) for figure 4:

**Figure supplement 1.** Social memory is impaired in juvenile *Df(16)A$^{+/-}$*mice.

**Figure supplement 2.** Genetic restoration of Emc10 levels in adulthood rescues SM performance in *Df(16)A$^{+/-}$*mice.

**Figure supplement 2—source data 1.** PDF file containing original western blots for *Figure 4—figure supplement 2C*, indicating the relevant bands.

**Figure supplement 2—source data 2.** Original files for western blot analysis used in *Figure 4—figure supplement 2C*.

**Figure supplement 3.** In vitro and in vivo screening of designed ASOs.

**Figure supplement 3—source data 1.** PDF file containing original western blots for *Figure 4—figure supplement 3D*, indicating the relevant bands.

**Figure supplement 3—source data 2.** Original files for western blot analysis used in *Figure 4—figure supplement 3D*.

**Figure supplement 4.** Social memory interaction task with novel stimulus mice.

treatment upon reintroduction of a novel juvenile mouse in trial 2 (*Figure 4—figure supplement 4A, B*), strongly indicating that SM deficits are not driven by a simple task fatigue.

To evaluate the consistency and reproducibility of the ASO-mediated SM rescue we generated additional ASOs targeting different regions within the Emc10 transcript and screened them for *Emc10* mRNA reduction in vivo (*Figure 5—figure supplement 1A–C*). One of these ASOs (1466182, herein referred to as Emc10$^{ASO2}$), which targets intron 1 of mouse *Emc10* (*Figure 5A*) was selected as the lead ASO candidate for the replication analysis based on its efficacy in reducing *Emc10* levels, distribution pattern, as well as the lack of any signs of astroglial/microglial activation or behavioral toxicity (see

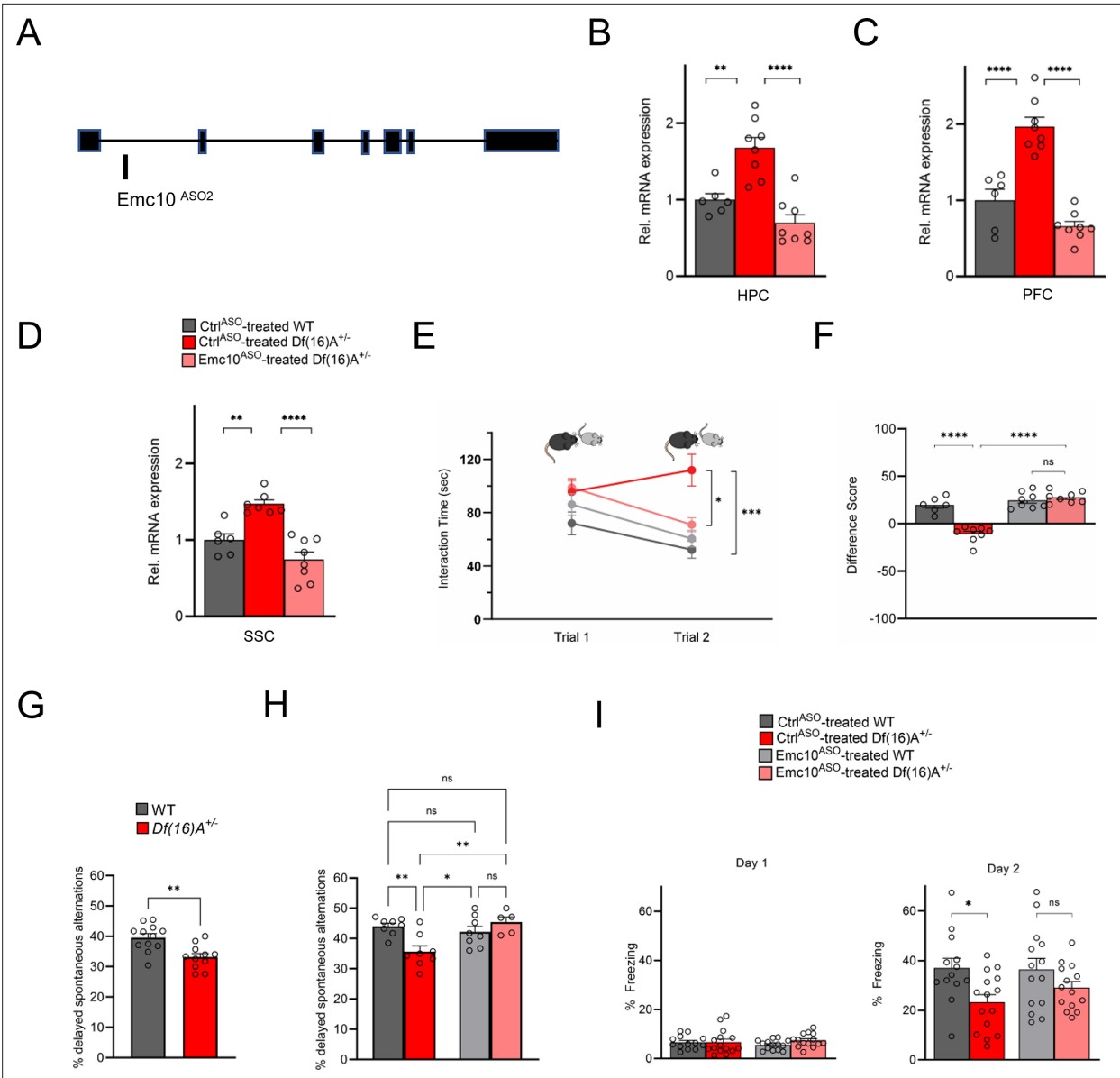

**Figure 5.** Restoration of cognitive function in *Df(16)A*⁺/⁻mice using an independent *Emc10* ASO. (**A**) Mouse *Emc10* gene map plot (ENSMUST00000118808) showing the Emc10^ASO2 target site. (**B**) qRT-PCR analysis shows significant upregulation of *Emc10* mRNA expression levels in the HPC of Ctrl^ASO-treated-*Df(16)A*⁺/⁻compared to WT mice [one-way ANOVA, F (2, 19)=20.92, p<0.0001; post hoc Tukey, p=0.0018]. Following ASO treatment, *Emc10* expression is normalized to WT levels in Emc10^ASO-treated-*Df(16)A*⁺/⁻compared to Ctrl^ASO-treated-*Df(16)A*⁺/⁻mice (post hoc Tukey, p=<0.0001). (**C**) qRT-PCR analysis shows a significant upregulation of *Emc10* mRNA expression levels in the PFC of Ctrl^ASO-treated-*Df(16)A*⁺/⁻compared to WT mice [one-way ANOVA, F (2, 19)=40.75, p=<0.0001; post hoc Tukey, p=<0.0001]. Following ASO treatment, *Emc10* expression is normalized to WT levels in the Emc10^ASO-treated-*Df(16)A*⁺/⁻compared to Ctrl^ASO-treated-*Df(16)A*⁺/⁻mice (post hoc Tukey, p=<0.0001). Ctrl^ASO-treated WT male mice: n=6, Ctrl^ASO-treated *Df(16)A*⁺/ male mice: n=8, and Emc10^ASO-treated *Df(16)A*⁺/ mice: n=8. (**D**) qRT-PCR analysis shows a significant upregulation of *Emc10* mRNA expression levels in Somatosensory Cortex (SSC) of Ctrl^ASO-treated-*Df(16)A*⁺/⁻compared to WT mice [one-way ANOVA, F (2, 18)=21.53, p=<0.0001; post hoc Tukey, p=<0.0026]. Following ASO treatment, *Emc10* expression is normalized to WT levels in Emc10^ASO-treated-*Df(16)A*⁺/⁻compared to Ctrl^ASO-treated-*Df(16)A*⁺/⁻mice (post hoc Tukey, p=<0.0001). Ctrl^ASO-treated WT mice: n=6, Ctrl^ASO-treated *Df(16)A*⁺/ mice: n=7, and Emc10^ASO-treated *Df(16)A*⁺/ mice: n=8. (**E**) Ctrl^ASO-treated *Df(16)A*⁺/⁻mice show a robust SM deficit compared to Ctrl^ASO-treated WT mice as indicated by the significant difference in trial 2 interaction time upon reintroduction of a familiar juvenile mouse [three-way ANOVA for Trial X Genotype X Treatment Interaction matching by trial: F (1, 26) = 35.74 p<0.0001; post hoc Tukey, p=0.0004]. A reduction in trial 2 interaction time indicates rescue of the SM deficit in Emc10^ASO-treated *Df(16)A*⁺/⁻compared to Ctrl^ASO-treated *Df(16)A*⁺/⁻mice (post hoc Tukey, p=0.0255). [Ctrl^ASO-treated WT male mice: n=6, Ctrl^ASO-treated *Df(16)A*⁺/ mice: n=8, Emc10^ASO-treated WT mice: n=8, and Emc10^ASO-treated *Df(16)A*⁺/ mice: n=8]. (**F**) A negative difference score (trial 1- trial 2) confirms the SM deficit in Ctrl^ASO-treated adult *Df(16)A*⁺/⁻mice compared to WT littermates two-way ANOVA for Genotype X Treatment interaction F (1, 26)=35.74, p<0.0001; post hoc Tukey, p<0.0001. Increase in the difference score of *Df(16)A*⁺/⁻mice in the Emc10^ASO- vs Ctrl^ASO- treated group demonstrates SM rescue (post hoc

*Figure 5 continued on next page*

*Figure 5 continued*

Tukey, p<0.0001). (**G**) Y-maze task displayed memory impairments in adult *Df(16)A*[+/-]mice. Impaired short-term spatial memory in *Df(16)A*[+/-]mice shown by the reduced amount of delayed alternations (%) after a delay of 1 hr (*P*=0.0015). WT mice: n=12, *Df(16)A*[+/]mice: n=11. (**H**) Deficits in short-term spatial memory in the Y-maze task in Ctrl[ASO]-treated adult *Df(16)A*[+/-]mice can be rescued in Emc10[ASO]-treated *Df(16)A*[+/-]mice (one-way ANOVA, F (3, 25)=6.727, p=0.0018, post hoc Tukey, p=0.0042) after 3 weeks of ASO-injection. Ctrl[ASO]-treated WT mice: n=8, Emc10[ASO]-treated WT mice: n=8, Ctrl[ASO]-treated *Df(16)A*[+/]mice: n=8 and Emc10[ASO]-treated *DfA*[+/]mice: n=5. (**I**) In a contextual fear memory assay, minimal freezing is observed on day 1 (left) with no significant changes across groups [two-way ANOVA for Genotype X Treatment interaction: F (1, 52)=1.003, p=0.3211]. In the Ctrl[ASO]-treated group, *Df(16)A*[+/-] mice show the expected contextual fear memory deficit compared to WT mice (right panel) one-way ANOVA, F (3, 52)=3.524 p=0.0212; post hoc Tukey, 0.0384. Emc10[ASO] treatment was not sufficient to fully rescue the learning deficit in *Df(16)A*[+/-]mice compared to Ctrl[ASO]-treated WT levels (post hoc Tukey, p=0.1045). However, there is increased freezing in *Df(16)A*[+/-]mice injected with Emc10[ASO]- versus Ctrl[ASO]-treated group, which results in a non-significant difference in freezing between Emc10[ASO]-treated *DfA*[+/]and WT mice (two-way ANOVA for genotype x treatment F (1, 52)=0.8524, p=0.3601; post hoc Tukey, p=0.4676) indicating a partial rescue of the contextual fear memory deficit. Ctrl[ASO]-treated WT: n=13 (9 males, 4 females), Emc10[ASO]-treated WT mice: n=14 (10 males, 4 females), Ctrl[ASO]-treated *Df(16)A*[+/]mice: n=15 (11 males, 4 females) and Emc10[ASO]-treated *Df(16)A*[+/]mice: n=14 (10 males, 4 females). Unpaired students t-test, one- two- or three-way ANOVA as indicated. Data are presented as mean ± SEM, *p<0.05, **p<0.01, ***p<0.01, ****p<0.0001.

The online version of this article includes the following figure supplement(s) for figure 5:

**Figure supplement 1.** In vivo post-injection screening of designed ASOs.

**Figure supplement 2.** Emc10[ASO2] reduces Emc10 expression in adult *Df(16)A*[+/-]mice.

Materials and methods). IHC analysis showed robust ASO distribution in both the hippocampus (HPC, *Figure 5—figure supplement 2A*, first and second panels) and prefrontal cortex (PFC, *Figure 5—figure supplement 2A*, third and fourth panels), as well as diffusion into both neuronal and non-neuronal cells. This was indicated by colocalization with the neuronal marker NeuN and the glial marker GFAP (*Figure 5—figure supplement 2A*, bottom panel). Higher-magnification (40 x) analysis revealed extensive overlap between the ASO signal and NeuN staining, demonstrating that the vast majority (>97%) of neurons exhibited ASO uptake. qRT-PCR analysis of *Df(16)A*[+/-]mice treated by intraventricular injection at 8 weeks of age showed that Emc10[ASO2] effectively lowered *Emc10* mRNA to nearly WT levels 3 weeks post-injection resulting in normalization of *Emc10* expression in the HPC (*Figure 5B*), PFC (*Figure 5C*) and somatosensory cortex (SSC) (*Figure 5D*). To study the effects of Emc10[ASO2]-mediated *Emc10* reduction on SM performance, 8-week-old *Df(16)A*[+/-]male mice and WT littermates were treated by ICV injection of Emc10[ASO2] and Ctrl[ASO2] and tested 3 weeks later. *Df(16)A*[+/-] mice injected with Emc10[ASO2] had significantly improved SM performance to levels indistinguishable from Emc10[ASO2]-treated WT littermates (*Figure 5E and F*).

In addition to SM deficits, mouse models of the 22q11.2 deletion show a spectrum of cognitive impairments in episodic and spatial memory as reflected, for example, in impaired performance in an Y-maze-based delayed alternations task that probes short-term spatial memory (*Tripathi et al., 2020*) and contextual fear conditioning a form of associative learning test used for studying episodic learning and spatial memory (*Stark et al., 2008*). We investigated the impact of ASO-mediated Emc10 reduction in the adult brain on both of these cognitive tasks. First, we confirmed that adult male *Df(16) A*[+/-]mice exhibit impaired short-term spatial memory during novelty exploration in a two-trial delayed alternation Y-maze task (*Figure 5G*) as previously described for another mouse model of the 22q11.2 deletion (*Benger et al., 2018*). The total number of arm entries remained unchanged, indicating no alterations in locomotor activity (*Figure 5—figure supplement 2B*). To determine whether reducing Emc10 expression in the brain via ASO treatment could rescue short-term spatial memory deficits, we tested a new cohort of *Df(16)A*[+/-]mice and WT littermates 3 weeks following ASO administration (*Figure 5H*). ASO-treated *Df(16)A*[+/-]mice exhibited a significant improvement in delayed alternations compared to *Df(16)A*[+/-]mice treated with control ASO (*Figure 5H*). Furthermore, no significant differences in total number of arm entries were observed between the groups (*Figure 5—figure supplement 2C*). We confirmed the reduction of *Emc10* levels in the ASO-treated animals through qRT-PCR assays of the HPC, PFC, and SSC brain regions (*Figure 5—figure supplement 2D–F*). In the contextual fear conditioning task, while ASO treatment was not sufficient to fully rescue the learning deficit in *Df(16)A*[+/-]mice to WT levels (*Figure 5I*, right panel), there was a modest improvement in fear memory of ASO-treated *Df(16)A*[+/-]mice, since these mice did not differ significantly from the ASO-treated WT littermates. Interestingly, we have previously shown that genetic reduction of *Emc10* levels in *Df(16) A*[+/-]mice resulted in only partial restoration of deficits in contextual fear memory (*Diamantopoulou*

*et al., 2017*). Thus, our finding faithfully recapitulates results from our previous constitutive genetic rescue assays (*Diamantopoulou et al., 2017*) and likely indicates a more limited role of *Emc10* upregulation in the 22q11.2-linked fear memory deficits rather than requirement for additional treatment time or for earlier onset of *Emc10* normalization.

The application of ASOs as a novel therapeutic strategy has seen a significant rise in recent years, in part due to their versatility in durably modifying RNA transcripts. Therefore, we investigated the longevity of ASO-mediated repression of *Emc10* as an indicator of future therapeutic relevance for the treatment of 22q11.2DS. To this end, we conducted the SM and Y-maze assays on a new cohort of *Df(16)A$^{+/-}$*mice at 3–4 weeks and 8–9 weeks post-injection with Emc10$^{ASO2}$ (*Figure 6A*). We observed behavioral rescue in Emc10$^{ASO2}$-treated *Df(16)A$^{+/-}$*mice in the SM assay at 3e weeks (*Figure 6B*, left panel) and in the Y-maze assay at 4 weeks post ASO-administration (*Figure 6C*, left panel) in accordance to our previous findings (*Figure 5F and H*). Remarkably, we replicated these results at 8–9 weeks post-injection, demonstrating sustained behavioral rescue of SM (*Figure 6B*, right panel) and spatial memory deficits (*Figure 6C*, right panel). Importantly, locomotor activity remained unchanged in the Y-maze assays at both, 4 weeks (*Figure 6—figure supplement 1A*) and 9 weeks (*Figure 6—figure supplement 1B*) post-injection. Finally, we confirmed the downregulation of *Emc10* in ASO-treated animals via qRT-PCR assays of the HPC, PFC, and SSC brain regions at 10 weeks post-treatment (*Figure 6D–F*). These results suggest that normalizing *Emc10* expression in the brain can ameliorate social and spatial memory deficits in adult *Df(16)A$^{+/-}$*mice in a time period of at least 2 months.

## Discussion

Despite an understanding of the molecular mechanisms of 22q11.2DS, especially ones related to abnormal expression and processing of miRNAs (*Stark et al., 2008*; *Xu et al., 2013*; *Diamantopoulou et al., 2017*; *Khan et al., 2020*; *Paranjape et al., 2023*), we still do not have a promising therapeutic avenue for the cognitive and neuropsychiatric symptoms associated with the 22q11.2 deletion. By leveraging our recent understanding of the molecular, cellular and behavioral consequences of 22q11.2-linked miRNA dysregulation, the present study represents an advancement towards developing a potential therapeutic strategy in two ways:

*First*, we show that 22q11.2 deletion results in abnormal processing of miRNAs in human neurons and in turn drives upregulation of *EMC10 levels* as previously described in mouse models (*Stark et al., 2008*). Effective reduction to near WT levels or even complete depletion of *EMC10* leads to restoration of key alterations in morphological and functional neuronal maturation emerging due to 22q11.2 deletions. The miRNA regulatory mechanism underlying *EMC10* upregulation and the restoration of cellular deficits are very similar to the ones we previously described in *Df(16)A$^{+/-}$*mice, highlighting the robustness of this molecular alteration as well as the translational value of using animal models to probe the link between *Emc10* upregulation and 22q11.2-linked behavioral dysfunction.

*Second*, we show that normalization of *Emc10* levels in adult *Df(16)A$^{+/-}$*mouse brain, by ASO-mediated targeted knockdown, is effective in rescuing SM deficits (which emerge during postnatal development and are present as early as postnatal day 22) as well as short-term spatial memory deficits. Significantly, these improvements in cognition were sustained for over 2 months post-ASO administration. These findings strongly suggest that at least for a subset of 22q11.2-associated cognitive deficits it is the sustained miRNA dysregulation and elevation of *Emc10* throughout the adult life that interferes with the underlying neural processes rather than an irreversible impact on brain maturation during early development and demonstrate the therapeutic potential for treating a wide range of cognitive symptoms associated with 22q11.2DS.

In vivo delivery of ASOs offers a potential venue for emerging treatments of genetically driven and postnatally reversible symptoms of neurodevelopmental disorders (NDDs) focusing in reduction of culprit gene expression via sequence-specific knockdown of mRNA transcripts (*Benger et al., 2018*). A common challenge in efforts to employ gene-knockdown therapies for dosage-sensitive genes such as *EMC10* is restricting target gene expression within optimal levels to avoid potential toxicity due to target hyper-knockdown or complete elimination (*Shao et al., 2021b*; *Kaiyrzhanov et al., 2022*; *Shao et al., 2021a*; *Umair et al., 2020*). A large number of relatively rare LoF variants or potentially damaging missense variants have been identified in the human *EMC10* gene among likely healthy individuals in gnomAD (*Shao et al., 2021a*; *Karczewski et al., 2020*) which is depleted of individuals known to be affected by severe NDDs. Taken together with

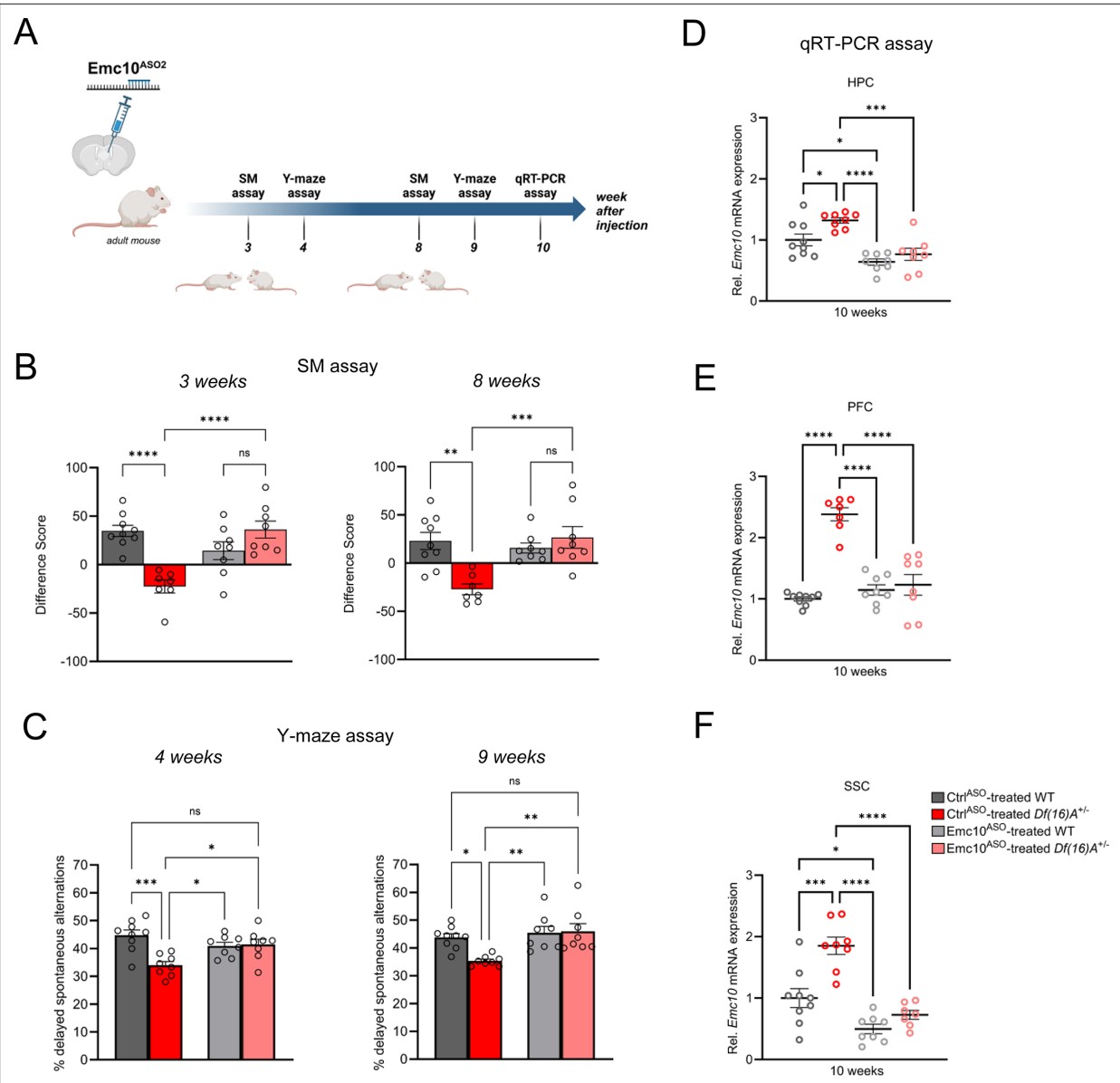

**Figure 6.** Sustained rescue of cognitive deficits in *Df(16)A⁺/⁻* mice following ASO administration. (**A**) Experimental timeline of Emc10^ASO2^-treated adult *Df(16)A⁺/⁻* mice to determine long-term rescue effect of Emc10 de-repression. This panel was created using BioRender.com. (**B–C**) Sustained ASO effect on behavioral rescue of Emc10^ASO^-treated *Df(16)A⁺/⁻* mice in HPC- and PFC-dependent tasks. (**B**) Rescue of SM deficit in *Df(16)A⁺/⁻* mice after 3- and 8 weeks of ASO injection. After 3 weeks of ASO treatment (left panel), a negative difference score (trial 1- trial 2) confirms the SM deficit in Ctrl^ASO^-treated adult *Df(16)A⁺/⁻* mice compared to WT littermates two-way ANOVA for Genotype X Treatment interaction F (1, 28)=26.20, p<0.0001; post hoc Tukey, p<0.0001. Increase in the difference score of *Df(16)A⁺/⁻* mice in the Emc10^ASO^- vs Ctrl^ASO^- treated group demonstrates SM rescue (post hoc Tukey, p<0.0001). After 8 weeks of ASO treatment (right panel), a negative difference score (trial 1- trial 2) confirms the SM deficit in Ctrl^ASO^-treated adult *Df(16) A⁺/⁻* mice compared to WT littermates two-way ANOVA for Genotype X Treatment interaction F (1, 28)=13.36, p=0.0010; post hoc Tukey, p=0.0012. Increase in the difference score of *Df(16)A⁺/⁻* mice in the Emc10^ASO^- vs Ctrl^ASO^- treated group demonstrates prolonged SM rescue (post hoc Tukey, p=0.0007). Ctrl^ASO^-treated WT mice: n=9, Ctrl^ASO^-treated *Df(16)A⁺/⁻* mice: n=7, Emc10^ASO^-treated WT mice: n=8 and Emc10^ASO^-treated *Df(16)A⁺/⁻* mice: n=8. (**C**) Y-maze task displayed rescue of memory impairments in adult *Df(16)A⁺/⁻* male mice 4- and 9 weeks after Emc10^ASO^ injection. Deficits in short-term spatial memory shown by the reduced number of delayed alternations (%) in Ctrl^ASO^-treated adult *Df(16)A⁺/⁻* mice can be rescued in Emc10^ASO^-treated *Df(16)A⁺/⁻* mice after 4 weeks of ASO injection (left panel) one-way ANOVA, F (3, 29)=7.585, p=0.0007, post hoc Tukey, p=0.0193. Deficits in short-term spatial memory in Ctrl^ASO^-treated adult *Df(16)A⁺/⁻* mice can be rescued in Emc10^ASO^-treated *Df(16)A⁺/⁻* mice also after 9 weeks of ASO injection (right panel) one-way ANOVA, F (3, 29)=6.547, p=0.0016, post hoc Tukey, p=0.0029. Ctrl^ASO^-treated WT mice: n=9, Ctrl^ASO^-treated *Df(16)A⁺/⁻* mice: n=8, Emc10^ASO^-treated WT mice: n=8 and Emc10^ASO^-treated *Df(16)A⁺/⁻* mice: n=8. (**D–F**) Sustained Emc10 de-repression after 2 months of Emc10^ASO^ injection. ASO-mediated inhibition of *Emc10* in different brain regions of *Df(16)A⁺/⁻* mice after 10 weeks of injection. (**D**) qRT-PCR analysis shows significant upregulation of *Emc10* mRNA expression levels in the HPC of Ctrl^ASO^-treated-*Df(16)A⁺/⁻* compared to WT mice one-way ANOVA, F (3, 29)=13.97, p<0.0001; post hoc

*Figure 6 continued on next page*

*Figure 6 continued*

Tukey, p=0.0331. Following ASO treatment, *Emc10* expression is normalized to WT levels in Emc10[ASO]-treated-*Df(16)A[+/−]*compared to Ctrl[ASO]-treated-*Df(16)A[+/−]*mice (post hoc Tukey, p=<0.0002). Ctrl[ASO]-treated WT mice: n=9, Ctrl[ASO]-treated *Df(16)A[+/]* mice: n=8, Emc10[ASO]-treated WT mice: n=8 and Emc10[ASO]-treated *Df(16)A[+/]* mice: n=8. (**E**) qRT-PCR analysis shows a significant upregulation of *Emc10* mRNA expression levels in the PFC of Ctrl[ASO]-treated-*Df(16)A[+/−]*compared to WT mice [one-way ANOVA, F (3, 28)=32.47, p<0.0001; post hoc Tukey, p<0.0001]. Following ASO treatment, *Emc10* expression is normalized to WT levels in the Emc10[ASO] treated-*Df(16)A[+/−]*compared to Ctrl[ASO] treated-*Df(16)A[+/−]*mice (post hoc Tukey, P<0.0001). Ctrl[ASO]-treated WT mice: n=9, Ctrl[ASO]-treated *Df(16)A[+/]* mice: n=7, Emc10[ASO]-treated WT mice: n=8 and Emc10[ASO]-treated *Df(16)A[+/]* mice: n=8. (**F**) qRT-PCR analysis shows a significant upregulation of *Emc10* mRNA expression levels in Somatosensory Cortex (SSC) of Ctrl[ASO]-treated-*Df(16)A[+/−]*compared to WT mice one-way ANOVA, F (3, 28)=23.18, p<0.0001; post hoc Tukey, p=0.0001. Following ASO treatment, *Emc10* expression is normalized to WT levels in Emc10[ASO] treated-*Df(16)A[+/−]*compared to Ctrl[ASO] treated-*Df(16)A[+/−]*mice (post hoc Tukey, p<0.0001). Ctrl[ASO]-treated WT mice: n=9, Ctrl[ASO]-treated *Df(16)A[+/]* mice: n=8, Emc10[ASO]-treated WT mice: n=8 and Emc10[ASO]-treated *Df(16)A[+/]* male mice: n=7. One- or two-way ANOVA as indicated. Data are presented as mean ± SEM, *p<0.05, **p<0.01, ***p<0.01, ****p<0.0001.

The online version of this article includes the following figure supplement(s) for figure 6:

**Figure supplement 1.** Y-maze task in adult male mice after Emc10[ASO2] post-injection.

our previous analysis of constitutive heterozygous Emc10 knockout mice (*Diamantopoulou et al., 2017*) that showed no evidence of motor deficits or anxiety-like behavior, these observations strongly suggest that partial reductions in Emc10 levels of approximately 50% are well tolerated at the organismal level and do not result in behavioral abnormalities. The highest dose for both Emc10-specific ASOs used in mutant *Df(16)A[+/−]*mice was limited to ~300 µg and reduced *Emc10* mRNA to either normal or below normal (30–50% of WT) expression while attaining full behavioral rescue. While higher dose may be required to ameliorate other behavioral deficits (*Shao et al., 2021b*; *Shao et al., 2021a*), it should be noted that an acute injection of 700 µg Emc10[ASO] in WT mice that resulted in ~50% reduction of Emc10 levels (*Figure 5—figure supplement 1*) did not cause secondary cellular and behavioral toxicity. Thus, available evidence indicates that therapeutically effective ASO-mediated normalization of *EMC10* levels can be up- or down-titrated within a well-tolerated range. However, further work is needed to establish a comprehensive safety profile, including the evaluation of non-cognitive phenotypes, to fully validate the therapeutic potential of Emc10-targeting approaches. Additionally, the long-term effects of Emc10 reduction beyond 2 months post-injection require further investigation to determine the full extent of its therapeutic benefits. Future studies will also explore whether ASO-mediated normalization of adult Emc10 levels can restore additional 22q11.2 behavioral and cognitive alterations and whether earlier postnatal ASO treatment could prevent the onset of behavioral deficits or mitigate those resistant to adult interventions.

In addition to reduction of *Emc10* expression, our findings have implications for therapeutic interventions aiming to manipulate its downstream targets. In that respect, it is attractive to speculate that different *EMC10* upregulation-linked phenotypes and their developmental requirements may be driven by dysregulation of distinct, individually or in combinations, downstream EMC targets. Such targets are typically multi-transmembrane domain (TMD) proteins (*Shurtleff et al., 2018*; *Tian et al., 2019*; *Wideman, 2015*) that contain low-hydrophobicity TMDs which are hard to insert into ER membrane and thus require the aid of EMC as a membrane insertase (*Guna et al., 2018*; *Chitwood et al., 2018*; *Shurtleff et al., 2018*). Identification of neurotransmitter receptors, channels, and transporters whose biogenesis, trafficking and membrane insertion are affected by EMC10 upregulation could help establish a link between such targets and 22q11.2-related behavioral dysfunction and guide efforts to develop treatments for specific 22q11.2 deletion symptoms.

Overall, by highlighting the manipulation of EMC10 expression and activity, as well as its downstream targets, as a promising alternative or augmentation to currently available treatments, and emphasizing a broad temporal window for therapeutic and preventive intervention in 22q11.2 deletion-associated cognitive and behavioral symptoms, our results lay the foundation for developing mechanism-based therapeutic strategies. These strategies aim to leverage insights from both human and animal models to improve clinical outcomes in precision medicine for neuropsychiatric disorders. Furthermore, our findings may have broader implications for understanding the role of miRNA dysregulation and ER-mediated membrane protein translocation in other neurodevelopmental and neuropsychiatric disorders with overlapping genetic or phenotypic features.

## Materials and methods

### Mice

We used *Emc10* conditional knockout (see below) and *Df(16)A⁺ᐟ⁻* mice (**Stark et al., 2008**) in C57BL/6 J background. *Df(16)A⁺ᐟ⁻* male mice were crossed to C57BL/6 J female mice to obtain either *Df(16)A⁺ᐟ⁻* or WT littermates.

### Generation of *Emc10* conditional knockout compound mouse lines

To manipulate the expression of the *Emc10* gene in *Df(16)A⁺ᐟ⁻* mice, we used a *Emc10* conditional 'knockout-first' mouse design by conducting Cre- and Flip- dependent genetic switch strategies as described earlier (**Mukai et al., 2019**). This approach enables postnatal manipulation of *Emc10* expression at its endogenous locus keeping Emc10 expression within its physiological levels. $Emc10^{tm1a+/}$ mice (2310044h10rik-Tm1a, MRC Harwell Institute, Oxfordshire, UK) were crossed to a germline Flp mouse line (B6.129S4-Gt(ROSA)26Sortm1(FLP1)Dym/RainJ_JAX:009086) that activates global *Flp* function and leads to the deletion of the frt-flanked sequence(s) in the offspring. The $Emc10^{tm1c}$ offspring from these cross carries a *loxP* flanked WT *Emc10* allele. We used UBC-Cre/ERT2 mice (B6. Cg-Ndor1 Tg(UBC-cre/ERT2)1Ejb /2 J_JAX:008085) in crosses to generate compound $Emc10^{tm1c+/-}$; UBC-cre/ERT2; *Df(16)A⁺ᐟ⁻* mice.

Mice of both sexes and genotypes (mutant and WT littermates) were used for behavioral testing. Separate cohorts of mice were used for Social Memory and Fear Conditioning assays. In general, mice were group housed under a 12 hr light/12 hr dark cycle with controlled room temperature and humidity. Food and water were provided ad libitum. All behavioral experiments were performed on adult male and female mice during the light cycle.

### Cell line donors

#### Q6 and Q5 lines

The Q6 line donor is a 20-year-old female patient with a history of developmental delay and an overall Full-Scale IQ in the low 80 s. She was clinically diagnosed with 22q11.2DS by FISH testing. Her psychotic symptoms, including disorganized behavior and command auditory hallucinations, started when she was 17 years old. During the first break episode, due to the severity of her psychotic symptoms, the patient was hospitalized and was diagnosed with schizophrenia. The patient also developed depressive symptoms, including frequent suicidal ideation. One year after her schizophrenia diagnosis, in addition to her severe psychotic symptoms, the patient was also diagnosed to be in a catatonic state. The patient has remained severely psychotic since the onset of these symptoms at age 17 and has been on multiple antipsychotics without experiencing any clinically meaningful benefit. Regarding her treatment history includes various first-line antipsychotics (including olanzapine, stelazine, aripiprazole, haloperidol, risperidone and clozapine); several antidepressants (sertraline and fluoxetine); a mood stabilizer (lithium) and benzodiazepines (e.g. lorazepam). None of these medications reportedly led to any clinically significant improvement in either the psychotic or the depressive symptoms. The patient has also undergone two rounds of electroconvulsive treatment (ECT), but with only short-lived improvement. The Q5 line donor is the probands dizygotic twin sister who does not carry a 22q11.2 deletion and her psychiatric evaluation ruled out any history of psychiatric symptoms (**Supplementary file 1**). Sibling are of Caucasian Western European descent.

#### Q1 and Q2 lines

The Q1 and Q2 line were previously described (as DEL3 and WT3) (**Li et al., 2021**). The Q1 line donor is a 32-year-old male patient with a history of developmental and speech delay. He was clinically diagnosed with 22q11.2DS by FISH testing at age 4. His psychotic symptoms, started when he was 12 years old. The patient was hospitalized once at age 10 before he was formally diagnosed with schizophrenia. During that time, he also experienced one seizure. The patient also developed mood lability and has OCD-like symptoms although does not meet full criteria for DSM-IV/V OCD. Regarding his treatment history, it includes various first-line antipsychotics as well as metyrosine (started when he was 15). The Q2 line donor is the proband's brother, who does not carry a 22q11.2 deletion and his psychiatric evaluation ruled out any history of psychiatric symptoms (**Supplementary file 1**). Siblings are of Caucasian Western European descent.

### QR20 and QR27 lines

QR20 (MH0159020) and QR27 (MH0159027) lines were obtained from the NIMH Repository and Genomics Resource (http://www.nimhstemcells.org/) (*Lin et al., 2016*). The donor of the QR27 line, 31-year-old male was diagnosed with schizoaffective disorder and 22q11.2DS (*Lin et al., 2016*) while the donor of the QR20 line (58 year old male) was free from any psychiatric symptoms (*Supplementary file 1*). Both are of Caucasian descent.

## hiPSC generation and characterization

Q5 and Q6 hiPSC lines were generated at the Columbia Stem Cell Core via non-integrating Sendai virus-based reprogramming (*Fusaki et al., 2009*) of monocytes from a donor with 22q11.2DS and SCZ and a healthy sibling control. The Q1 and Q2 lines were generated at the Columbia Stem Cell Core and characterized as described earlier (*Li et al., 2021*). QR20 and QR27 lines were obtained from the NIMH Repository and Genomics Resource (http://www.nimhstemcells.org/) (*Lin et al., 2016*). Karyotyping was performed on twenty G-banded metaphase cells at 450–500 band resolution as previously described (*Riera et al., 2019*) to ensure the absence of chromosomal abnormalities in all patient and control derived cell lines (*Figure 1—figure supplement 1A*, *Figure 3—figure supplement 1B*). We confirmed the genotypes of Q6 patient- and Q5 control-derived hiPSCs using a Multiplex Ligation-dependent Probe Amplification (MLPA) assay to detect copy number changes (*Figure 1—figure supplement 1B, C*). To confirm stemness of hiPSC lines, we performed qRT-PCR for markers *NANOG* and *OCT4/POU5F1* (*Figure 1—figure supplement 1D*).

## Genome editing of Q6(22q11.2) hiPCS line

We generated derivatives of the Q6 (22q11.2) hiPSC line carrying either heterozygous (Q6/EMC10[HET]) or homozygous (Q6/EMC10[HOM]) *EMC10* LoF mutations using standard CRISPR/Cas9 genome editing approaches. The genomic gRNA target sequences were EMC10-g1: ACAGTGCCAACTTCCGGAAG (PAM suffix: CGG) and EMC10-g2: GGGACAAGGTACCATCCTGC (PAM suffix: TGG). Mutations in *EMC10* were confirmed by NGS and no off-target candidates were predicted in both lines using COSMID tool (https://crispr.bme.gatech.edu) (*Cradick et al., 2014*). Karyotyping confirmed normal chromosome complement in both modified lines (*Figure 3—figure supplement 1B*). qRT-PCR and WB assays were performed in both lines for the hiPSC markers *NANOG* and *OCT4/POU5F1* to confirm stemness as well as the *RANBP1* gene located within 22q11.2 locus to confirm the deletion (*Figure 3—figure supplement 1C–E*). qRT-PCR and western blot assays were performed to confirm reduction or elimination of EMC10 levels in the *EMC10* LoF mutant lines (*Figure 3—figure supplement 1F–G*).

## Culture and neuronal induction of hiPSC lines

hiPSC lines were maintained in mTeSR Plus medium (catalog#05825, Stemcell Technologies, Vancouver, Canada) on Matrigel (catalog#354277, Corning, Corning, NY, USA) coated tissue culture plate. Cells were fed on any other day and passaged weekly using ReLeSR (catalog#05872, Stemcell Technologies, Vancouver, Canada) dissociation reagent in accordance to their manual. Dissociated cells were pre-plated as reported earlier (*Qi et al., 2017*) at a density of 200,000 cells/cm$^2$ supplemented with 10 µM Y-27632 (catalog# 1254, Tocris Bioscience, Bristol, United Kingdom) on Matrigel-coated plates and differentiation started when confluent. Differentiation of hiPSC was performed as indicated below:

(*i*) hiPSC differentiation into cortical neurons, via a combination of small molecule inhibitors, was performed as described with few modifications (*Qi et al., 2017*). This protocol has been shown to robustly generate cortical neurons while actively suppressing glial differentiation, as evidenced by the lack of upregulation of glial markers such as GFAP, AQP4, or OLIG2. In brief, inhibitors used in LSB +X/P/S/D induction included LDN193189 (250 nM; catalog#04–0074, Stemgent, REPROCELL USA Inc, Beltsville, MD, USA), SB431542 (10 µM; catalog#1614, Tocris Bioscience, Bristol, United Kingdom), XAV939 (5 µM; catalog#3748, Tocris Bioscience, Bristol, United Kingdom), PD0325901 (1 µM; catalog#4192, Tocris Bioscience, Bristol, United Kingdom), SU5402 (5 µM; catalog#1645–05, BioVision Inc, Milpitas, CA, USA), DAPT (10 µM; catalog#2634, Tocris Bioscience, Bristol, United Kingdom). Until day 4 of differentiation, TeSR E6 medium (catalog#05946, Stemcell Technologies, Vancouver, Canada) was added in 1/3 increments every other day. Then, neurobasal (NB) plus medium supplemented with N2 (catalog#17502–048, Gibco, Life Technologies, Grand Island, NY, USA) and

B27 plus supplement (catalog#A35828-01, Gibco, Life Technologies, Grand Island, NY, USA) was added in 1/3 increments every other day from day 4, until reaching 100% neurobasal plus/B27 plus L-glutamine (catalog#35050–061, Gibco, Life Technologies, Grand Island, NY, USA) containing medium supplemented with BDNF (20 ng/ml; catalog#248-BDB, R&D Systems, Minneapolis, MN, USA), cAMP (0.5 mM; catalog#A6885, Sigma-Aldrich, St. Louis, MO, USA) and ascorbic acid (0.2 mM; catalog#A92902, Sigma-Aldrich, St. Louis, MO, USA) (BCA) at day 8 as described (*Qi et al., 2017*). For long-term culture, cells were passaged on day 8 of differentiation by Accutase (catalog#AT-104, Innovative Cell Technologies Inc, San Diego, CA, USA) dissociation for 6 min at 37 °C. Cells were replated at 200,000 cells/cm$^2$ onto Matrigel coated culture dishes. NB plus/B27 plus and BCA medium were used for passaging and long-term culture. Culture medium was changed every 3–4 days.

(*ii*) hiPSC differentiation into neurons via NGN2 overexpression was performed as described previously with few modifications (*Zhang et al., 2013*; *Yi et al., 2016*; *Ho et al., 2016*). In brief, differentiation of hiPSC into neurons (iNs) was conducted by using the lentiviral infection of NGN2 and the reverse tetracycline transactivator rtTa into hiPSCs, followed by selection on puromycin. The lentiviral particles were commercially produced by VectorBuilder (Chicago, IL, USA) using the established and published plasmids pLenti-FUW-M2rtTA (FUW-M2rtTA deposited by Rudolf Jaenisch, Addgene plasmid #20342; http://n2t.net/addgene:20342; RRID:Addgene_20342, *Hockemeyer et al., 2008*), pLenti-TetO-hNGN2-eGFP-puro (pLV-TetO-hNGN2-eGFP-Puro deposited by Kristen Brennand, Addgene, plasmid#79823; http://n2t.net/addgene:79823; RRID:Addgene_79823, *Ho et al., 2016*) and for calcium imaging, the pLenti-TetO-hNGN2-puro (pLV-TetO-hNGN2-Neo deposited by Kristen Brennand, Addgene plasmid # 99378; http://n2t.net/addgene:99378; RRID:Addgene_99378, *Ho et al., 2017*). Around 200 k cells per well (24-well format) were plated on Matrigel coated wells/coverslips in mTeSR plus media supplemented with 10 µM Y-27632. On day 0, lentiviruses were added in fresh basic media, containing always DMEM/F12 (catalog#11330032, Thermo Fisher Scientific, Waltham, MA, USA), human BDNF (10 ng/ml), human NT-3 (10 ng/ml, catalog#450–03, PeproTech, East Windsor, NJ, USA), mouse laminin (0.1 µg/ml, catalog#354232, Corning, NY, USA), N2 supplement, B27 plus supplement, non-essential amino acids (NEAA, catalog#SH30238.01, Cytiva, Marlborough, MA, USA) supplement and doxycycline (1 mg/ml, catalog#D9891-1G, Sigma-Aldrich, St. Louis, MO, USA) to induce TetO gene. On day 1, the culture medium was completely replaced with fresh basic media and a 48 hr puromycin selection (1 mg/l, catalog#P8833-10MG, Sigma-Aldrich, St. Louis, MO, USA) period was started. On day 3, for calcium imaging and morphology analysis, ca. 25% mouse glia cells (prepared as previously reported in *Ho et al., 2016*) were added to the basic media plus Ara-C (2 µM, catalog#C1768-100M, Sigma-Aldrich, St. Louis, MO, USA) and 10% mouse astrocyte conditioned media (catalog#M1811-57, ScienCell, Carlsbad, CA, USA) to promote neuronal health and maturation. On day 5, total medium was changed with basic media plus Ara-C and 10% mouse astrocyte conditioned media. On day 7, total media was removed and replaced with BrainPhys Neuronal media with SM1 supplement (catalog#05792, STEMCELL Technologies, Vancouver, Canada) containing always human BDNF (10 ng/ml), human NT-3 (10 ng/ml), mouse laminin (0.1 µg/ml), N2 supplement, doxycycline (1 mg/ml) and 10% mouse astrocyte conditioned media. From day 9 on, half of the media were removed and replaced with supplemented BrainPhys media every other day. 2.5% FBS (catalog#16141079, Thermo Fisher Scientific, Waltham, MA, USA) was added to the culture medium on day 11 to those cells which were co-cultured with astrocytes to support astrocyte viability. iN cells were assayed for experiments as indicated.

## Cell culture transfection

Transfection of cortical neurons at day 8 of differentiation were performed with the transfection reagent Lipofectamine 2000 (#11668–030, Life Technologies, Carlsbad, CA, USA) as described earlier (*Lackinger et al., 2019*). Cells were transfected for 48 hr with 25 pmol per well (24-well format) of miRNA mimic Pre-miR miRNA precursors (Ambion, Thermo Fisher Scientific, Waltham, MA, USA) as indicated: pre-miR Negative Control #1 (catalog#17110), hsa-miR-185–5 p (catalog#17100, PM12486) and hsa-miR-485–5 p (catalog#17100, PM10837) or with 50 pmol per well (24-well format) of miRNA miRVana inhibitors (Ambion, Thermo Fisher Scientific, Waltham, MA, USA) as indicated: miRNA inhibitor Neg. Ctrl #1 (catalog#4464076), hsa-miR-185–5 p inhibitor (catalog#4464084, MH12485), hsa-miR-485–5 p inhibitor (catalog#4464084, MH10837).

## TAM preparation and feeding

We used oral gavage for TAM delivery during postnatal days 56–70. A TAM feeding protocol was used as previously described (*Mukai et al., 2019*). In brief, TAM (catalog#T5648, Sigma-Aldrich, St. Louis, MO, USA) was dissolved in corn oil (catalog#C8267, Sigma-Aldrich, St. Louis, MO, USA) at 20 mg/ml by vortexing. To avoid toxicity, the following dosages were used for adult animals (8–10 weeks): mice at 17–21 g body weight were fed 5 mg/day; mice at 22–25 g body weight were fed 6 mg/day; mice at 26–29 g body weight were fed 7 mg/day; mice at 30–35 g body weight were fed 8 mg/day. Adult animals were fed for 5 consecutive days followed by 2 days of rest. Animals were then fed for 5 more consecutive days followed by 1 week of rest before RNA/protein or SM assays were performed. Corn oil was used as vehicle control treatment.

## ASOs

Mouse *Emc10*-targeting ASOs used in these studies were 20 bases in length, chimeric 2′ -O- (2-methoxyethyl) (MOE/DNA) oligonucleotides with phosphodiester and phosphorothioate linkages. The central gap of 10 deoxynucleotides is flanked on its 5′ and 3′ sides by five MOE modified nucleotides. Oligonucleotides were synthesized at Ionis Pharmaceuticals (Carlsbad, CA, USA) as described previously (*Cheruvallath et al., 2003*; *McKay et al., 1999*). ASOs were solubilized in 0.9% sterile saline or PBS. Mouse *Emc10*-targeting ASOs used in these studies were 20 bases in length, chimeric 2′ -O- (2-methoxyethyl) (MOE/DNA) oligonucleotides with phosphodiester and phosphorothioate linkages. The central gap of 10 deoxynucleotides is flanked on its 5′ and 3′ sides by five MOE modified nucleotides. Oligonucleotides were synthesized at Ionis Pharmaceuticals (Carlsbad, CA, USA) as described previously (*Cheruvallath et al., 2003*; *McKay et al., 1999*). ASOs were solubilized in 0.9% sterile saline or PBS.

## In vitro screening of ASOs

4T1 cells were trypsinized, counted and diluted to 200,000 cells per ml in room temperature growth medium before adding 100 µL of the cell suspension to the wells of a 2 mm electroporation plate (Harvard Apparatus, Holliston, MA, USA) which contained 11 µL of 10 X ASO in water. Cells were pulsed once at 130 V for 6 mS with the ECM 830 instrument (Harvard Apparatus). After electroporation, the cells were transferred to a Corning Primeria 96-well culture plate (catalog #353872, Corning, NY, USA) containing 50 µL of growth medium. The cells were then incubated at 37 ° C and 5% $CO_2$. After 24 hr, the cells were washed 1 x with PBS before lysing for RNA isolation and analysis. For each treatment condition duplicate wells were tested. ASO1081815 (TTGTTCCTACAGATCTAGGG, referred to in the manuscript as Emc10[ASO1]) was used in behavioral and immunocytochemical assays.

## In vivo screening of ASOs

Candidate *Emc10*-targeting ASOs (700 µg) were stereotactically injected into the right lateral ventricle of C57Bl/6 mice (0.3 mm anterior, 1.0 mm dextrolateral, 3.0 mm ventral from bregma). Reduction of *Emc10* mRNA in the retrosplenial cortex and thoracic spinal cord was evaluate by qRT-PCR at 2 weeks following a single bolus dose. Three ASOs were selected from the screen based on their pharmacological efficacy: 1466167, 1466171, and 1466182. Animals were injected with these ASOs in the right lateral ventricle as described above (n=4) and euthanized 8 weeks post-injection. Animals were evaluated with an observational functional battery test at 3 hr after dosing and then every 2 weeks until euthanasia. The retrosplenial cortex and thoracic spinal cord were harvested for qRT-PCR analysis of *Emc10*, *Aif1* (microglia marker), *Cd68* (phagocytic microglia marker), and *Gfap* (reactive astrocyte marker) mRNA. Brain and spinal cord were also harvested and fixed in formalin solution for histological evaluation. Tissues were stained for H&E and IBA1 (microglia marker), CD68 (phagocytic microglia marker) GFAP (reactive astrocyte marker) and Calbindin (Purkinje Cell marker). Bolus injections of all three candidate ASOs resulted in similar reductions of the *Emc10* mRNA at both 2 and 8 weeks. Of the three candidate ASOs, ASO1466182 (GCCATATCTTTATTAATTAC, referred to in the manuscript as Emc10[ASO2]) showed no signs of in vivo toxicity and the tolerability marker gene expression was similar between ASO- and vehicle-treated animals in the tissues evaluated. There was no positive IBA1 IHC staining in the CNS of any of the treated animals. Therefore, it was ranked as the best candidate for further behavioral analysis in mutant mice.

## Stereotactic intracerebroventricular (ICV) injections of ASOs

ASOs were delivered to 8-week-old mice via ICV injections using a Hamilton syringe (Hamilton Company, Reno, NV, USA) connected to a motorized Stoelting Quintessential Stereotaxic Injector QSI/53311 (Stoelting Co., Wood Dale, IL, USA). The syringe was attached to a glass pipette with a long-tapered end made using a Sutter pipette puller model P-87. Anesthesia was delivered using Kent Scientific VetFlo Traditional Vaporizer VetFlo-1205S (Kent Scientific Corporation, Torrington, CT, USA). Mice were initially put in the isoflurane chamber using 3% isoflurane mixture for 5 min, which was lowered to 2–2.5% when fixed to the stereotactic station. We used KOPF Small Animal Stereotaxic Instruments (Model 940). Carprofen (5 mg/kg, Zoetis Inc, Kalamazoo, MI, USA), and Bupivacaine (2 mg/kg, Hospira, Inc, Lake Forest, Il, USA) were delivered subcutaneously before the incision was made. Additionally, Dexamethasone (2 mg/kg, Bimeda-MTC Animal Healt Inc, Cambridge, ON, Canada) was delivered intramuscularly. The surgical site was shaved and sterilized with betadine and 70% ethanol three times. A small midline incision was made and a hole was drilled in the skull. Stereotactic bregma coordinates used for the right ventricle were - 0.5 mm posterior, −1.1 lateral, and −2.8 mm dorsoventral. Mice were injected with 4 ul of either the Ctrl$^{ASO1/ASO2}$ or *Emc10*$^{ASO1/ASO2}$ (Emc10$^{ASO1}$: 292 μg, Emc10$^{ASO2}$: 280 μg) at a rate of 0.5 μl/min. The needle was left in the injection site for 10 min to allow diffusion and avoid back flow of the ASO upon retraction of the glass pipette. Mice were maintained at a temperature of 37 °C for the duration of the surgery using a water regulated heating pad (T/Pump TP 700, Stryker Corporation, Kalamazoo, MI, USA). Mice were placed on heating pads for in cage recovery and Carprofen was administered subcutaneously for 3 days post-surgery. Mice were then subjected to behavioral experiments/immunohistochemistry 3 weeks post-surgery. qRT-PCR assays were performed 1 week post behavioral assays.

## Quantitative real-time PCR (qRT-PCR)

Total RNA was extracted from HPC, PFC and Somatosensory Cortex (SSC) using the RNeasy Mini Kit (catalog#1038703, QIAGEN, Hilden, Germany) or using the miRVana miRNA isolation kit (#AM1560, Ambion, Thermo Fisher Scientific, Waltham, MA, USA) for RNA extraction of hiPSC and derived neurons samples in accordance to their manuals. cDNA was synthesized using High-Capacity RNA-to-cDNA Kit from Applied Biosystems (cat#4387406, Thermo Fisher Scientific Baltics, Vilnius, Lithuania). qRT-PCR was performed using the Bio-Rad CFX-384 qPCR instrument (Bio-Rad, Hercules, CA, USA) using TaqMan Universal Master Mix II, with UNG (catalog#4440038, Thermo Fisher Scientific Baltics, Vilnius, Lithuania). Mouse *Gapdh* Endogenous Control (cat# 4352339E, Life Technologies, Warrington, United Kingdom) served as housekeeping gene and TaqMan Mm01197551_m1 (catalog#4351372) probe for mouse *Emc10* as well as Mm01208065_m1 for *Emc10-1* (cat# 4351372) and Mm01197555_ m1 for *Emc10-2* (cat# 4351372) mRNA detection were used for the qRT-PCR assay. For mouse qRT-PCR assay of Ctrl ASO$^{ASO1}$/Emc10$^{ASO1}$, threshold cycle of each sample was picked from the linear range to calculate the values for Starting Quantity (SQ) for all samples extrapolated using the Standard Curve. All samples were run together in triplicates on the same plate including the standard curve ran in duplicates. The SQ values were averaged over the triplicates. The values of *Emc10* mRNA levels were then normalized to the values from the *Gapdh* gene expression levels. For mouse qRT-PCR analysis of Ctrl ASO$^{ASO2}$/Emc10$^{ASO2}$, the average of triplicate CT values from each sample was used to calculate the relative RNA levels ($2^{-\Delta CT}$) as described earlier (*Brenes et al., 2016*) and all values were then normalized to Ctrl$^{ASO2}$-treated WT group. qRT-PCR for hiPSC and derived neurons was performed using TaqMan or SYBR Green System (catalog#1725121, iTaq Universal SybrGreen Supermix with ROX; BIO-RAD, Hercules, CA, USA) for mRNA and/or pre-miRNA detection according to manufacturer's instructions. *U6* snRNA was used as housekeeping gene to normalize pre-miRNAs targets. For detection of *RANBP1* (catalog# 4331182, Hs01597912), *OCT4/POU5F1* (catalog#4331182, Hs04260367), *NANOG* (catalog#4331182, Hs02387400), *TBR1* (catalog#4331182, Hs00232429), *GFAP* (catalog#4331182, Hs00909233), *NEUROD1* (catalog#4331182, Hs00159598), *vGLUT1* (catalog#4331182, Hs00220404), *DLX1* (catalog#4331182, Hs00698288), BRN2 (catalog#4331182, Hs00271595) and *EMC10* (catalog#4331182, Hs00382250) mRNA TaqMan probes (Thermo Fisher Scientific, Waltham, MA, USA) were used, as indicated, with *GAPDH* as housekeeping gene control (human *GAPDH* endogenous control, catalog#4325792, Life Technologies, Warrington, United Kingdom). The average of triplicate CT values from each sample was used to calculate the relative RNA levels ($2^{-\Delta CT}$). Primer sequences for pre-miRNA are provided in *Supplementary file 9*. Primers

were purchased from Integrated DNA Technologies (Coralville, IA, USA) and were diluted to a stock concentration of 100 μM.

## Immunohistochemistry

Animals were euthanized using $CO_2$ and then perfused with 4% Paraformaldehyde. The brains were stored at 4 °C in 4% PFA overnight and were then switched to 1 x Phosphate Buffered Saline (PBS). 2.5% low melting agarose in 1xPBS buffer was added to the brains placed inside the plastic molds, which were then moved to 4 °C to create a solid block for slicing. This block was then glued to the vibratome stage (Leica Vibratome VT10005, Wetzlar, Germany) and sectioned at 40 μm thickness. Sections were rinsed in 1xPBS for 5 min and then blocked for 1 hr at room temperature in 2% Normal Goat Serum (NGS) and 0.3%Triton X-100 in 1xPBS solution. Sections were then incubated with either anti-ASO and NeuN or anti-ASO and GFAP primary antibodies in blocking solution at 4 °C and left on a shaker overnight. The following day, sections were washed three times with 1xPBS for 10 min and were then stained with Goat anti-rabbit along with either Goat anti-mouse or Goat anti-chicken secondary antibodies in 2% NGS and 0.4% Triton-X100 solution made in 1xPBS for 2 hr in the dark at room temperature (RT). Following two 10 min washes with 1xPBS, sections were additionally stained with Hoechst nuclear stain diluted in 1xPBS for 15 min in the dark. Lastly, sections were washed three times with 1xPBS for 10 min. Sections from PBS solution were mounted on glass slides, air-dried and cover slipped in Prolong Diamond Antifade Mountant (catalog#P36970, Life Technologies Corporation, Eugene, OR, USA). Slides were left overnight in the dark and then stored at 4 °C. Human neuronal cultures were fixed on coverslips in 4%PFA (catalog#22023, Biotium, Fremont, CA, USA) for 1 hr at RT and then blocked for 1.5 hr at RT in 0.1% Triton-X and 10% horse serum (catalog#H0146, Sigma-Aldrich, St. Louis, MO, USA) solution. After fixation, coverslips were stained with the primary antibody in a 0.1% Triton-X and 2% horse serum solution overnight at 4 °C. Coverslips were then washed 3x15 min with DPBS (catalog#D8537, Sigma-Aldrich, St. Louis, MO, USA) and cells were incubated for 1 hr with the secondary antibody at RT followed by 3x15 min DPBS washing. Tissue sections and cultured cells were imaged on W1-Yokogawa Spinning Disk Confocal (Nikon Instruments, Tokyo, Japan).

## Antibodies for immunohistochemistry

The following primary antibodies were used: Rabbit polyclonal anti-ASO antibody diluted 1:10,000 (Ionis Pharmaceuticals, Carlsbad, CA, USA); Anti-GFAP antibody diluted 1:1000 (Aves Labs Inc, catalog#GFAP), Mouse monoclonal Anti-NeuN antibody diluted 1:200 (Millipore, catalog#MAB377), anti-TUJ1 1:500 (mouse monoclonal, catalog#T8660, Sigma-Aldrich, St. Louis, MO, USA), anti-TBR1 1:100 (rabbit monoclonal, #Ab183032, Abcam, Cambridge, MA, USA), anti-GFP 1:1000 (goat polyclonal, catalog#600-101-215, Rockland Immunochemicals, Pottstown, PA, USA), anti-MAP2 1:2000 (chicken polyclonal, catalog#5392, Abcam, Cambridge, MA, USA). The following secondary antibodies were used for mouse at a dilution of 1:500: Goat anti-Rabbit (Alexa Fluor 488: catalog#AA1008, Invitrogen, Waltham, MA, USA) against ASO; Goat anti-mouse (Alexa Fluor 568: catalog#AA1004, Invitrogen, Waltham, MA, USA) against NeuN, and Goat anti-chicken (Alexa Fluor 568: catalog#A11041, Invitrogen, Waltham, MA, USA) against GFAP. Hoechst 33258 solution diluted 1:1000 (Catalog#94403, Sigma Aldrich, Saint Louis, MO, USA) was used for nuclear staining in brain slices and DAPI Fluoromount-G (catalog#0100–20, Southern Biotech, Birmingham, AL, USA) was used for cultured cells.

## Analysis of dendritic complexity

iNs were prepared on coverslips, fixed at DIV21 and immunostained for TBR1 and MAP2 to identify dendritic branches. Images of dendrites were acquired on a Nikon Spinning Disk Confocal Microscope and captured using the Nikon NIS Elements AR (v.5.21.03 64-bit) software. Images were acquired and analyzed as previously described (*Xu et al., 2013*; *Mukai et al., 2008*). Image analysis of dendritic complexity was conducted blind to genotype. Primary dendrites were defined as any branch emerging from the soma and a secondary dendrite as any branch emerging from a primary dendrite. Dendrite branches were semi-automatically traced using NeuronStudio software (v.0.9.92 64-bit; *Wearne et al., 2005*). The output.swc files were then processed in VAA3D (v.3.1.00; *Peng et al., 2010*; *Peng et al., 2014a*; *Peng et al., 2014b*) and binary images were generated and analyzed using ImageJ (http://rsbweb.nih.gov/ij/, NIH, Bethesda, MD, USA).

## Calcium imaging

Calcium imaging was performed as previously described (*Barreto-Chang and Dolmetsch, 2009*) with a few modifications. Neuronal cells used in calcium imaging experiments were prepared on glass bottom dishes (14 mm, catalog#P35G-1.5–14 C, MatTek, Ashland, MA, USA). Briefly, cells were incubated at DIV37-38 with 0.3 μM Fluo-4, AM (Invitrogen, catalog#F14201) for 30 min at 37 °C in incubation buffer medium containing 170 mM NaCl, 3.5 mM KCl, 0.4 mM $KH_2PO_4$, 20 mM TES (N-tris[hydroxyl-methyl-2-aminoethane-sulfonic acid], 5 mM $NaHCO_3$, 5 mM glucose, 1.2 mM $Na_2SO_4$, 1.2 mM $MgCl_2$, 1.3 mM CaCl2, pH 7.4) and washed once with incubation buffer medium before imaging. After imaging for 2'20" (baseline), 25 mM KCl solution was added to the cells, followed by another 25 mM KCl addition at 3'20" and 4'20" min. Live imaging was performed at RT (~25 °C) on a W1-Yokogawa Spinning Disk Confocal (Nikon Instruments, Tokyo, Japan). ImageJ software with plugin for motion correction (MuliStackReg) and Excel were used to collect, manage and quantify time-lapse excitation ratio images by selecting cell body as ROI.

## Protein extraction for western blot

To extract proteins from mouse HPC, the tissue was dissected and homogenized in QIAzol lysis reagent (catalog#79306, QIAGEN, Hilden, Germany). Chloroform was added to homogenate and the solution was then incubated at room temperature for 3 min. Tissue was spun at 12,000 x *g* at 4 °C and the organic phase was collected for protein extraction. We followed an optimized protocol for protein extraction from Trizol solutions and used 5% SDS +20 mM EDTA +140 mM NaCl buffer solution for protein pellet suspension (*Kopec et al., 2017*). To extract proteins from mouse PFC, the tissue was dissected and homogenized using a modified Pierce RIPA lysis and extraction buffer (RIPA+) (catalog#89900, Thermo Scientific, Rockford, IL, USA) that contains a Halt Protease Inhibitor Cocktail (catalog#1861281, Thermo Scientific, Rockford, IL, USA). Cultured neuronal cells were lysed at day 8 of differentiation. The cultured cells were once washed with cold DPBS (catalog#D8537, Sigma-Aldrich, St. Louis, MO, USA). Cells were then lysed by adding a modified Pierce RIPA lysis and extraction buffer (RIPA+). The plate was shaking for 20 min at 4 °C on an orbital shaker (catalog#980173, Talboys, Troemner, Thorofare, NJ, USA). To remove cell debris, the lysates were centrifuged at maximum speed for 10 min at 4 °C. The protein concentration of the supernatant for all protein samples was determined by Pierce BCA Protein Assay Kit (#23227, Thermo Fisher Scientific, Rockford, IL, USA).

## Western blots

For each lane,~20 μg protein were run on a 4–12% Bis-Tris Criterion XT Precast Gel #3450123, Bio-Rad (Bio-Rad, Hercules, CA, USA) next to the Precision Plus Protein Dual Color Standard (catalog#161–0374, Bio-Rad, Hercules, CA, USA) in SDS-PAGE running buffer (catalog#1610788, Bio-Rad, Hercules, CA, USA) and afterwards transferred to a methanol-activated Immobilon-P PVDF (poly-vinylidene difluoride) membrane (catalog#IPCVH00010, Merck Millipore Ltd., Carrigtwohill, Ireland) by tank blotting at 250mA for 90 min in a cold room (4 °C) in blotting buffer (catalog#1610734, Bio-Rad, Hercules, CA, USA). The membrane was blocked for 2 hr in TBS-T (Tris-buffered saline supplemented with 0.1% Tween) containing 5% milk powder. Antibody dilutions anti-*EMC10/Emc10* 1:1000 (rabbit polyclonal; catalog#Ab181209, Abcam, Cambridge, MA, USA), anti-DGCR8 1:1000 (rabbit monoclonal, catalog#Ab191875, Abcam, Cambridge, MA, USA), anti-RANBP1 1:500 (rabbit polyclonal, catalog#Ab97659, Abcam, Cambridge, MA, USA) and anti-alpha-Tubulin1:1000 (poly-clonal rabbit; catalog#2144 S, Cell Signal, Danvers, MA, USA) as loading control were prepared in TBS-T/milk and the membrane were incubated overnight at 4 °C under slight shaking on an orbital shaker (catalog#980173, Talboys, Troemner, Thorofare, NJ, USA). After three washes with TBS-T, the membrane was incubated with LI-COR goat anti-Rabbit antibody IRDye 800CW (catalog#925–32211, LI-COR Bioscience, Lincoln, NE, USA) for 1.5 hr at RT. After three washes with TBS-T, the membrane was developed using the LI-COR Odyssey CLx system (LI-COR Bioscience, Lincoln, NE, USA) using LI-COR Image Studio software (Ver.5.2) and quantification of band intensity was performed by ImageJ (NIH, Bethesda, MD, USA).

## Bulk RNAseq and bioinformatic analysis of mouse hippocampal samples

Total RNA was isolated from 4 WT Ctrl[ASO1], 4 WT Emc10[ASO1], 5 *Df(16)A*[+/-] Ctrl[ASO1] and 4 *Df(16)A*[+/-]Emc10[ASO1] treated male hippocampi. Stranded polyA +enriched RNA sequencing libraries were prepared

at the Columbia Genome Center (Columbia University, New York, USA) to generate 40 million paired-end reads on Illumina Novaseq 6000 instrument using STRYPOLYA library prep kit. Sequence reads were aligned to the mouse genome (Ensembl, GRCm38) using the STAR sequence alignment tool (version 2.7) (*Dobin et al., 2013*) and gene count matrices were generated. Differential gene expression was analyzed using the DESeq2 pipeline (*Love et al., 2014*) in R and volcano plots were generated using the open-source Enhanced Volcano package in R (https://github.com/kevinblighe/EnhancedVolcano; *Blighe et al., 2023*).

## Bulk RNAseq and small RNA/miRNAseq of hiPSC-derived cortical neurons at DIV8

Total RNA was extracted from hiPSC-derived cortical neurons using the miRVana miRNA isolation kit (#AM1560, Ambion, Thermo Fisher Scientific, Waltham, MA, USA) before bulk RNAseq (paired-ended sequencing; Illumina NovaSeq 6000) or miRNAseq (single-end sequencing; Illumina HiSeq 2500) were performed. For bulk RNAseq, Poly(A) RNA sequencing library was prepared following Illumina's TruSeq-stranded-mRNA (Illumina, San Diego, CA, USA) sample preparation protocol. RNA integrity was checked with Bioanalyzer 2100 (Agilent, CA, USA). Poly(A) tail-containing mRNAs were purified using oligo-(dT) magnetic beads with two rounds of purification. After purification, poly(A) RNA was fragmented using divalent cation buffer in elevated temperature. Quality control analysis and quantification of the sequencing library were performed using Agilent Technologies 2100 Bioanalyzer High Sensitivity DNA Chip (Agilent, CA, USA). Paired-ended sequencing was performed on Illumina's NovaSeq 6000 (LC Sciences, Houston, TX, USA) sequencing system. For miRNAseq, total RNA quality and quantity was analyzed with Bioanalyzer 2100 (Agilent, CA, USA), with RIN number >7.0. Approximately 1 μg of total RNA was then used to prepare small RNA library according to the protocol of TruSeq Small RNA Sample Prep Kits (Illumina, San Diego, CA, USA). Then, a single-end sequencing 50 bp on an Illumina Hiseq 2500 following the vendor's recommended protocol was conducted.

## Bioinformatics analysis of human neuron bulk RNAseq at DIV8

Cutadapt (*Martin, 2011*) and in house perl scripts were used to remove the reads that contained adaptor contamination, low-quality bases and undetermined bases. Sequence quality was subsequently verified using FastQC (http://www.bioinformatics.babraham.ac.uk/projects/fastqc/). HISAT2 (*Kim et al., 2015*) was used to map reads to the genome of ftp://ftp.ensembl.org/pub/release-101/fasta/homo_sapiens/dna/. The mapped reads of each sample were assembled using StringTie (*Pertea et al., 2015*). Then, all transcriptomes were merged to reconstruct a comprehensive transcriptome using perl scripts and GffCompare. After the final transcriptome was generated, StringTie and edgeR (*Robinson et al., 2010*) were used to estimate the expression levels of all transcripts. StringTie was used to assess expression levels for mRNAs by calculating FPKM. The differentially expressed mRNAs were selected with log2 (fold change)>1 or log2 (fold change) <-1 and with statistical significance (p-value <0.05) by R package edgeR. For the VolcanoPlot visualization, the web-based R package Shiny application 'VolcanoPlot' (https://paolo.shinyapps.io/ShinyVolcanoPlot/) was used. DEGs (adj. p value <0.05) were plotted by selecting the axes (Log2(FC) range = –2.5/2.5 and -Log10(pvalue)=15) and setting the cutoff selection (p-value threshold = 1.3 (0.05), Log2(FC) threshold 0.4 and 3). The GO-Term enrichment analysis was performed for the up- and downregulated protein-coding genes (1937/2094, DEG) by using the standard setting (g:SCS threshold) of the gProfiler webtool (https://biit.cs.ut.ee/gprofiler/gost) (*Raudvere et al., 2019*). TargetScan (v8.0, http://www.targetscan.org/vert_80/) was used for miRNA binding site prediction (*Agarwal et al., 2015*). Intersection of genes and predicted targets were conducted by using the VIB / UGent Bioinformatics & Evolutionary Genomics (Gent, Belgium) webtool 'Venn' (https://bioinformatics.psb.ugent.be/webtools/Venn/).

## Bioinformatics analysis of human neuron miRNA-seq at DIV8

Raw reads were subjected to an in-house program, ACGT101-miR (LC Sciences, Houston, TX, USA) to remove adapter dimers, junk, low complexity, common RNA families (rRNA, tRNA, snRNA, snoRNA) and repeats. Subsequently, unique sequences with length in 18~26 nucleotide were mapped to specific species precursors in miRBase (v22.0) by BLAST search to identify known miRNAs and novel 3p- and 5p- derived miRNAs. Length variation at both 3' and 5' ends and one mismatch inside of the sequence were allowed in the alignment. The unique sequences mapping to specific species mature miRNAs

in hairpin arms were identified as known miRNAs. The unique sequences mapping to the other arm of known specific species precursor hairpin opposite to the annotated mature miRNA-containing arm were considered to be novel 5p- or 3 p derived miRNA candidates. The remaining sequences were mapped to other selected species precursors (with the exclusion of specific species) in miRBase 22.0 by BLAST search, and the mapped pre-miRNAs were further BLASTed against the specific species genomes to determine their genomic locations. The above two mentioned mapped 5 p and 3 p sequences were defined as known miRNAs. The unmapped sequences were BLASTed against the specific genomes, and the hairpin RNA structures containing sequences were predicted from the flank 80 nt sequences using RNAfold (http://rna.tbi.univie.ac.at/cgi-bin/RNAWebSuite/RNAfold.cgi) software. The criteria for secondary structure prediction were: (1.) number of nucleotides in one bulge in stem (≤12) (2.) number of base pairs in the stem region of the predicted hairpin (≥16) (3.) cutoff of free energy (kCal/mol ≤−15) (4.) length of hairpin (up and down stems + terminal loop≥50) (5.) length of hairpin loop (≤20). (6.) number of nucleotides in one bulge in mature region (≤8) (7.) number of biased errors in one bulge in mature region (≤4) (8.) number of biased bulges in mature region (≤2) (9.) number of errors in mature region (≤7) (10.) number of base pairs in the mature region of the predicted hairpin (≥12) (11.) percent of mature in stem (≥80). For the VolcanoPlot visualization, the web-based R package Shiny application 'VolcanoPlot' (https://paolo.shinyapps.io/ShinyVolcanoPlot/) was used. DEmiRs (p-value <0.05) of known miRNAs were plotted by selecting the axes (Log2(FC) range=−5/5 and -Log10(pvalue)=10) and setting the cutoff selection (p-value threshold = 1.3 (0.05), Log2(FC) threshold 0.4). For the GO-term enrichment analysis of the up- and down-regulated miRNAs (153/133, DEmiRs) the webtool miRNet 2.0 (https://www.mirnet.ca/) with standard settings (tissue: nervous, targets: genes [miRTarBase 8.0]) was used (*Chang et al., 2020*). TargetScan (v8.0, http://www.targetscan.org/vert_80/) was used for all miRNA binding site prediction (*Agarwal et al., 2015*) for miR-185–5 p, miR1306-5p and miR-1286 and using the biochemical predicted occupancy model (*McGeary et al., 2019*) for table sorting.

## Bulk RNAseq and bioinformatic analysis of NGN2-induced neurons at DIV21

We used poly-A pull-down to enrich mRNAs from total RNA samples, then proceed with library construction using Illumina TruSeq chemistry (Illumina, San Diego, CA, USA). Libraries were then sequenced using Illumina NovaSeq 6000 at Columbia Genome Center. We multiplexed samples in each lane, which yields targeted number of paired-end 100 bp reads for each sample. We used RTA (Illumina) for base calling and bcl2fastq2 (version 2.19) for converting BCL to fastq format, coupled with adaptor trimming. We performed a pseudoalignment to a kallisto index created from transcriptomes (Ensembl v96, Human:GRCh38.p12) using kallisto (0.44.0) (*Bray et al., 2016*). We tested for differentially expressed genes using DESeq2 (adj. p value <0.05), R packages designed to test differential expression between two experimental groups from RNA-seq counts data. Intersection of DEGs were conducted by using the VIB / UGent Bioinformatics & Evolutionary Genomics (Gent, Belgium) webtool 'Venn' (https://bioinformatics.psb.ugent.be/webtools/Venn/). GO-Term enrichment analysis was performed for the up- and downregulated DEGs Q5(Ctrl)/Q6(22q11.2) and normalized up- and downregulated DEGs in Q6/EMC10$^{HET}$ and Q6/EMC10$^{HOM}$ by using the standard setting (g:SCS threshold) of the gProfiler webtool (https://biit.cs.ut.ee/gprofiler/gost) (*Raudvere et al., 2019*). Heatmaps of DEGs that were normalized in Q6/EMC10$^{HET}$ and Q6/EMC10$^{HOM}$ were generated with the R based web tool Heatmapper (http://www.heatmapper.ca) (*Babicki et al., 2016*). PPI network analysis of the rescued 103 DEGs in Q6/EMC10$^{HET}$ and Q6/EMC10$^{HOM}$ conditions were performed by using the web-application konnect2prot (*Kumar et al., 2023*). Hereby, GO-Term enrichment analysis of the identified 30 matched DEGs were performed by using g-Profiler webtool as indicated above.

## Behavioral assays

Mice were 11–17 weeks old at the time of behavioral testing except of the social memory assay in juveniles which was performed in 3-week-old animals. Behavior was assayed 3–9 weeks following surgical delivery of ASOs or 1 week following TAM/corn oil treatment. The following behavioral assays were performed in this study: Open field assay, social memory assay, contextual fear conditioning and Y-maze delayed alternation task. The experimenter was blind to mouse genotype and treatments

while performing behavioral assays and data analysis. Animals were given at least 1-week intervals between behavioral tests.

## Open field assay

The open field activity assay was performed as described earlier (*Diamantopoulou et al., 2017*). In brief, mouse activity was monitored in a clear illuminated acrylic chamber (25 cm x 25 cm) equipped with infrared sensors to automatically record horizontal and vertical activity (Coulbourn Instruments, Whitehall, PA, USA). Each mouse was initially placed in the center of the chamber and its activity was recorded and collected in 1 min bins for 1 hr using TruScan (v1.012–00) software (Coulbourn Instruments, Whitehall, PA, USA). The floors and walls of the open field were cleaned with 70% ethanol between trials.

## Social memory assay

Assays were performed as described earlier in juvenile (postnatal day 22–24) (*Lackinger et al., 2019*) and adult (*Piskorowski et al., 2016*; *Diamantopoulou et al., 2017*) mice. All experimental mice were single housed, transferred to the testing room 1 hr prior to testing and returned to their home cages after the completion of the experiment. Both male and female were tested in the juvenile SM assays, in the TAM/corn oil treatment assays as well as in the ASO1 assays. Only males were tested in the ASO2 assays. For the adult SM assays, stimulus mice (C57 BL/6 J) were obtained from Jackson Laboratory (Bar Harbor, ME, USA). All stimulus mice were between the ages of 3–4 weeks. Test and stimulus mice were sex-matched in the experimental trials. All trials were recorded using a video camera (Webcam Pro 9000, Logitech, Lausanne, Switzerland) and recorder software (Logitech video recording software). Stimulus mice were color marked on the tails to distinguish the stimulus mice from the test mice, during video analysis. The videos were manually scored for total interaction time over the course of the trials for interactions initiated by the test animal including anal sniffing, nose-to-nose touch and close following. For the novel/familiar paradigm, test and stimulus (novel) mice were placed together in a neutral cage and the interaction was recorded (trial 1). One hour after trial 1, the same stimulus (familiar) mouse was placed together with the test mouse and the interaction was recorded again (trial 2). A similar procedure was followed for the control novel/novel paradigm, except that we used different stimulus mice for trial 1 and trial 2 such that at trial 2 the stimulus mice were also novel for the experimental mice. Trials with experimental mice showing highly aggressive behavior towards the stimulus mice or mice that interacted for less than 24 s in trial 1 were excluded from the analysis.

## Contextual Fear Conditioning assay

Contextual Fear Conditioning assays were performed as described earlier (*Stark et al., 2008*; *Diamantopoulou et al., 2017*) using a Coulbourn animal shocker (Model H13-15 110 V, Coulbourn Instruments, Whitehall, PA, USA). Sound levels were checked with a Digital Sound Level Meter (Model: 407730, Extech Instruments, MA, USA) before beginning the trials. Using a cotton swab, pure lemon extract (McCormick & Co, Hunt Valley, MD, USA) was introduced into the testing chamber adding nine different but equally distributed spots on a napkin. Test mice were placed in the test chamber and received two pairs of a tone (30 s, 82 db) and a co-terminating shock (2 s, 0.7mA). Mice were then carefully picked with forceps and returned to their home cage. After 24 hr, mice were placed in the FC box again with same environment and lemon scent for 6 min in the absence of tone and shock to test for contextual memory. The box and grid were cleaned with 70% EtOH before and between every test run on both days. Videos were recorded and analyzed by using FreezeFrame 3 software (Harvard Apparatus, Holliston, MA, USA).

## Y-Maze assay

The Y-maze apparatus was made from white acrylic that consists of three equal-sized arms (38 cm long, 13 cm high, and 8 cm wide) of which each arm of the Y-maze was positioned at an equal angle and was purchased from SD Instruments (cat#7001–0419, San Diego, CA, USA). The Y-maze assay was performed as described previously in more detail (*Tripathi et al., 2020*; *Kraeuter et al., 2019*). In brief, adult male mice were tested on delayed alternations. Exploration in all three arms of the Y-maze was performed after a 1 hr delay from an initial training phase of 10 min, where one arm of the maze

was blocked (delayed alternation). Delayed alternation (%) was calculated as the number of entries in all three arms divided by the total number of entries in the first 5 min of the 10 min test phase, whereas the number of entries per arm was used as a measurement of activity and locomotion. The movement of mice was recorded with a camera mounted above the apparatus and the number of arm entries was counted manually.

## Statistical analysis

Data were analyzed using GraphPad Prism (GraphPad Software, Inc, San Diego, CA, USA). Data were evaluated as indicated, using either unpaired two-tailed t-test, Kolmogorov–Smirnov test, one-way, two-way, or three-way analysis of variance (ANOVA) tests followed by post hoc Tukey's multiple comparison test for comparisons across all groups. Data are presented as mean ± SEM. p Values for each comparison are described in the figure legends.

## Acknowledgements

We thank Dr. Huynh-Hao Bui and Andrew Watt for design and identification of ASOs, Mark Andrade and the Ionis synthesis group for ASO synthesis, and Dr. Mark Graham for ASO study design and feedback. We thank Naoko Haremaki for genotyping, maintaining the mouse colony and for assisting in the SM and Y-maze assays. We thank Panagiota Apostolou for her help with the calcium imaging setup. We thank Yan Sun and Vivian Zhu for the hiPSC line validation and maintenance. We thank Barbara Corneo and the Columbia Stem Cell Core Facility for the Q1, Q2, Q5 and Q6 hiPSC lines generation. Bio-samples of Q20 and Q27 hiPSCs were obtained from NIMH Repository & Genomics Resource. We would like to thank Zuckerman Institute's Cellular Imaging platform for instrument use and technical advice. We thank the Columbia Genome Center for genome sequencing and analysis support. We thank Linda Brzustowicz and Bill Manley from the Rutgers University and the staff members at RUCDR. Data and biomaterials generated in Study 125/Site 393 were funded by a NIMH grant to Dr. Herb Lachman (MH087840: Analysis of Glutamatergic Neurons Derived from Patient-Specific iPS Cells). The co-investigators on this grant included Dr. Deyou Zheng and Dr. Reed Carroll from the Albert Einstein College of Medicine. Patients and controls were recruited at the Albert Einstein College of Medicine and at the Child Psychiatry Branch, NIMH, directed by Dr. Judith L Rapoport. We thank all participating subjects and their families for their contributions. The overview *Figure 6A* was created with BioRender.com (agreement number:CM271GHNWZ). This work was supported by National Institute of Mental Health Grant (2R01MH097879), a Columbia University Translational Therapeutics Pilot Award to JAG and the Stavros Niarchos Foundation (SNF). This research used the service of the Columbia Genome Center (Genomics and High Throughput Screening Shared Resource), that was funded in part through the NIH/NCI Cancer Center Support Grant P30CA013696.

## Additional information

### Competing interests

Annie Ferng, Curt Mazur, Holly Kordasiewicz: Affiliated with Ionis Pharmaceuticals, Inc; no other competing interests to declare. Robert J Shprintzen: Affiliated with The Virtual Center for Velo-Cardio-Facial-Syndrome; no other competing interests to declare. The other authors declare that no competing interests exist.

### Funding

| Funder | Grant reference number | Author |
| --- | --- | --- |
| National Institute of Mental Health | 2R01MH097879 | Joseph A Gogos |
| Columbia University | | Joseph A Gogos |
| Stavros Niarchos Foundation | | Joseph A Gogos |

| Funder | Grant reference number | Author |
|--------|------------------------|--------|

The funders had no role in study design, data collection and interpretation, or the decision to submit the work for publication.

## Author contributions

Pratibha Thakur, Data curation, Formal analysis, Validation, Investigation, Visualization, Methodology, Writing – review and editing, Designed the in vivo ASO screening strategy; Performed ASO injections/surgeries; Conducted mouse-related RNA expression studies, immunohistochemistry, and behavioral assays/analysis in ASO-treated animals; Martin Lackinger, Conceptualization, Data curation, Formal analysis, Validation, Investigation, Visualization, Methodology, Writing – original draft, Writing – review and editing, Designed, performed, and analyzed all human iPSC and human iPSC-derived-neuronal experiments (including RNAseq/miRNAseq); Designed in vivo ASO screening strategy and conducted behavioral assays in (longitudinal) ASO-treated and Emc10 conditional knockout mice; Contributed to mouse-related RNA/protein expression assays, coordinated all behavioral and molecular experiments; Anastasia Diamantopoulou, Conceptualization, Validation, Investigation, Writing – review and editing, Contributed to the design/implementation of in vivo ASO screening strategy; Sneha Rao, Formal analysis, Visualization, Writing – review and editing, Analyzed mouse RNA-sequencing data; Yijing Chen, Validation, Investigation, Writing – review and editing, Contributed to ASO-related immunohistochemistry assays; Khakima Khalizova, Investigation, Writing – review and editing, Contributed to ASO injections/surgeries and qRT-PCR assays; Annie Ferng, Resources, Validation, Visualization, Methodology, Writing – review and editing, Contributed to the identification and characterization of lead ASOs; Curt Mazur, Resources, Validation, Visualization, Methodology, Writing – review and editing, Contributed to the identification and characterization of lead ASOs; Holly Kordasiewicz, Resources, Validation, Visualization, Methodology, Writing – review and editing, Contributed to the identification and characterization of lead ASOs; Robert J Shprintzen, Resources, Writing – review and editing, Provided patient referrals; Sander Markx, Resources, Validation, Investigation, Writing – original draft, Writing – review and editing, Contributed to the generation and initial characterization of human iPSC lines; Bin Xu, Conceptualization, Resources, Data curation, Formal analysis, Validation, Investigation, Visualization, Writing – original draft, Writing – review and editing, Contributed to the generation and initial characterization of human iPSC lines and the design of human neuron-related assays; Joseph A Gogos, Conceptualization, Data curation, Supervision, Funding acquisition, Writing – original draft, Project administration, Writing – review and editing, Designed the study

## Author ORCIDs

Martin Lackinger ⓘ https://orcid.org/0000-0002-7946-5366
Robert J Shprintzen ⓘ https://orcid.org/0000-0001-8940-4973
Joseph A Gogos ⓘ https://orcid.org/0000-0002-7491-4476

## Ethics

The generation and use of human iPS cell lines was approved by New York State Psychiatric Institute (NYSPI) Internal Review Board (NYSPI IRB protocol #7500) and Columbia University's Human Embryo and Embryonic Stem Cell Research Committee. All animal procedures were carried out in accordance with and approved by the Columbia University Institutional Animal Care and Use Committee (IACUC protocol #AC-AABG4571, #AC-AABS9602 and #AC-AABG8556).

All animal procedures were carried out in accordance with and approved by the Columbia University Institutional Animal Care and Use Committee (IACUC protocol #AC-AABG4571, #AC-AABS9602 and #AC-AABG8556).

Reviewer #1 (Public review): https://doi.org/10.7554/eLife.103328.3.sa1
Author response https://doi.org/10.7554/eLife.103328.3.sa2

---

# Additional files

## Supplementary files

Supplementary file 1. Clinical and demographic characteristics of 22q11.2DS/SCZ patients and

healthy controls.

Supplementary file 2. Differentially expressed microRNAs in cortical neurons (DIV8) from 22q11.2DS/SCZ (Q6) and control line (Q5) derived using small-molecule inhibitors of SMAD and WNT signaling pathways.

Supplementary file 3. DEGs in cortical neurons (DIV8) from 22q11.2DS/SCZ (Q6) and control line (Q5) derived using small-molecule inhibitors of SMAD and WNT signaling pathways.

Supplementary file 4. List of targets of downregulated miRNAs identified from the intersection of predicted targets of downregulated miRNAs and upregulated DEGs in hiPSC-derived cortical neurons at DIV8.

Supplementary file 5. Predicted targets of miR-185–5 p, miR-1286 and miR-1306–5 p in *EMC10* 3′UTR.

Supplementary file 6. DEGs in NGN2-iNs (DIV21) from 22q11.2DS/SCZ (Q6) and control line (Q5) that were normalized ('rescued') in Q6 EM10 HET and Q6 EMC10 HOM lines.

Supplementary file 7. List of 103 DEGs normalized in both Q6 EMC10 HET and EMC10 HOM lines, used for PPI konnect2prot network analysis.

Supplementary file 8. List of genes upregulated in Ctrl$^{ASO1}$ treated-*Df(16)A$^{+/-}$*mice compared to Ctrl$^{ASO1}$ treated-WT mice but not in the Emc10$^{ASO1}$ treated-*Df(16)A$^{+/-}$*compared to Emc10$^{-ASO1}$ treated-WT mice.

Supplementary file 9. List of primer sequences used for qRT-PCRs.

MDAR checklist

### Data availability

The sequencing data described in this manuscript were deposited into the Gene Expression Omnibus database under accession number GSE236596 and are available at the following URL: https://www.ncbi.nlm.nih.gov/geo/query/acc.cgi?acc=GSE236596.

The following dataset was generated:

| Author(s) | Year | Dataset title | Dataset URL | Database and Identifier |
|-----------|------|---------------|-------------|------------------------|
| Joseph AG, Martin L | 2025 | An antisense oligonucleotide-based strategy to ameliorate cognitive dysfunction in the 22q11.2 Deletion Syndrome | https://www.ncbi.nlm.nih.gov/geo/query/acc.cgi?acc=GSE236596 | NCBI Gene Expression Omnibus, GSE236596 |

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

# Appendix 1

## Appendix 1—key resources table

| Reagent type (species) or resource | Designation | Source or reference | Identifiers | Additional information |
|---|---|---|---|---|
| Strain, strain background (*Mus musculus*) | Df(16)A$^{+/-}$ | Ref (*Stark et al., 2008*) | N/A | C57BL/6 J background; 1.3 Mb deficiency syntenic to 1.5 Mb human 22q11.2 deletion |
| Strain, strain background (*Mus musculus*) | Emc10$^{tm1a+/-}$ | MRC Harwell Institute | 2310044h10rik-Tm1a | 'Knockout-first' allele; crossed with Flp line |
| Strain, strain background (*Mus musculus*) | Emc10$^{tm1c+/-}$ | This paper | N/A | Conditional KO via Flp recombination; loxP-flanked WT allele |
| Strain, strain background (*Mus musculus*) | UBC-Cre/ERT2 | Jackson Laboratory | RRID:IMSR_JAX:008085 | B6.Cg-Ndor1 Tg(UBC-cre/ERT2)1Ejb/2 J; tamoxifen-inducible Cre |
| Strain, strain background (*Mus musculus*) | Flp mouse line | Jackson Laboratory | RRID:IMSR_JAX:009086 | B6.129S4-Gt(ROSA)26Sortm1(FLP1)Dym/RainJ; germline Flp recombinase |
| Strain, strain background (*Mus musculus*) | C57BL/6 J | Jackson Laboratory | RRID:IMSR_JAX:000664 | 3–4 weeks old stimulus mice for SM assays |
| Cell line (*Homo sapiens*) | Q5 hiPSC | This paper | Columbia Stem Cell Core Facility | Control line from healthy twin |
| Cell line (*Homo sapiens*) | Q6 hiPSC | This paper | Columbia Stem Cell Core Facility | 22q11.2DS/SCZ patient line |
| Cell line (*Homo sapiens*) | Q1 hiPSC | This paper, Ref (*Li et al., 2021*) | Columbia Stem Cell Core Facility | 22q11.2DS/SCZ patient line; previously DEL3; Ref (*Li et al., 2021*) |
| Cell line (*Homo sapiens*) | Q2 hiPSC | This paper, Ref (*Li et al., 2021*) | Columbia Stem Cell Core Facility | Control line from healthy sibling; previously WT3; Ref (*Li et al., 2021*) |
| Cell line (*Homo sapiens*) | QR27 hiPSC | NIMH Repository, Ref (*Lin et al., 2016*) | MH0159027 | 22q11.2DS/SCZ patient line; Ref (*Lin et al., 2016*) |
| Cell line (*Homo sapiens*) | QR20 hiPSC | NIMH Repository, Ref (*Lin et al., 2016*) | MH0159020 | Control line from healthy donor, Ref (*Lin et al., 2016*) |
| Cell line (*Homo sapiens*) | Q6/EMC10$^{HET}$ | This paper | N/A | Heterozygous EMC10 LoF mutation in Q6 via CRISPR/Cas9 |
| Cell line (*Homo sapiens*) | Q6/EMC10$^{HOM}$ | This paper | N/A | Homozygous EMC10 LoF mutation in Q6 via CRISPR/Cas9 |
| Cell line (*Mus musculus*) | 4T1 | Ionis Pharmaceuticals | N/A | Used for in vitro ASO screening |
| Cell line (*Mus musculus*) | Mouse glial cells | Prepared in accordance to Ref (*Pak et al., 2018*) | N/A | 25% in NGN2-iN co-culture |
| Antibody | Anti-EMC10 (rabbit polyclonal) | Abcam | cat#Ab181209 | 1:1000 for Western blot |
| Antibody | Anti-DGCR8 (rabbit monoclonal) | Abcam | cat#Ab191875 | 1:1000 for Western blot |
| Antibody | Anti-RANBP1 (rabbit polyclonal) | Abcam | cat#Ab97659 | 1:500 for Western blot |
| Antibody | Anti-alpha-Tubulin (rabbit polyclonal) | Cell Signaling | cat#2144 S | 1:1000 for Western blot |

*Appendix 1 Continued on next page*

*Appendix 1 Continued*

| Reagent type (species) or resource | Designation | Source or reference | Identifiers | Additional information |
|---|---|---|---|---|
| Antibody | Anti-ASO (rabbit polyclonal) | Ionis Pharmaceuticals | N/A | 1:10000 for IHC; detects phosphorothioate backbone of the ASO |
| Antibody | Anti-GFAP (chicken polyclonal) | Aves Labs Inc. | cat#GFAP | 1:1000 for IHC |
| Antibody | Anti-NeuN (mouse monoclonal) | Millipore | cat#MAB377 | 1:200 for IHC |
| Antibody | Anti-TUJ1 (mouse monoclonal) | Sigma-Aldrich | cat#T8660 | 1:500 for IHC |
| Antibody | Anti-TBR1 (rabbit monoclonal) | Abcam | cat#Ab183032 | 1:100 for IHC |
| Antibody | Anti-GFP (goat polyclonal) | Rockland Immunochemicals | cat#600-101-215 | 1:1000 for IHC |
| Antibody | Anti-MAP2 (chicken polyclonal) | Abcam | cat#5392 | 1:2000 for IHC |
| Recombinant DNA reagent | pLenti-FUW-M2rtTA | Ref (*Hockemeyer et al., 2008*); Addgene | plasmid#20342 | Lentivirus; for NGN2 neuronal induction |
| Recombinant DNA reagent | pLenti-TetO-hNGN2-eGFP-puro | Ref (*Ho et al., 2016*); Addgene | plasmid#79823 | Lentivirus; for NGN2 neuronal induction |
| Recombinant DNA reagent | pLenti-TetO-hNGN2-puro | Ref (*Ho et al., 2017*); Addgene | plasmid#99378 | Lentivirus; for NGN2 neuronal induction |
| Sequence-based reagent | hsa-miR-185–5 p mimic | Ambion (Thermo Fisher) | cat#17100, PM12486 | Transfection |
| Sequence-based reagent | hsa-miR-485–5 p mimic | Ambion (Thermo Fisher) | cat#17100, PM10837 | Transfection |
| Sequence-based reagent | pre-miR Negative Control #1 | Ambion (Thermo Fisher) | cat#17110 | Transfection |
| Sequence-based reagent | hsa-miR-185–5 p inhibitor | Ambion (Thermo Fisher) | cat#MH12485 | Transfection |
| Sequence-based reagent | hsa-miR-485–5 p inhibitor | Ambion (Thermo Fisher) | cat#MH10837 | Transfection |
| Sequence-based reagent | miRNA inhibitor Neg. Ctrl #1 | Ambion (Thermo Fisher) | cat#4464076 | Transfection |
| Sequence-based reagent | EMC10-g1 gRNA (ACAGTGCC AACTTCCG GAAG) | This paper | N/A | CRISPR/Cas9 targeting EMC10; PAM: CGG |
| Sequence-based reagent | EMC10-g2 gRNA (GGGACAAG GTACCATC CTGC) | This paper | N/A | CRISPR/Cas9 targeting EMC10; PAM: TGG |
| Sequence-based reagent | qRT-PCR primers | Integrated DNA Technologies | N/A | See *Supplementary file 9* for sequences |
| Sequence-based reagent | TaqMan Mm01197551_m1 (*Emc10*) | Thermo Fisher | cat#4351372 | Mouse *Emc10* qRT-PCR |
| Sequence-based reagent | TaqMan Mm01208065_m1 (*Emc10-1*) | Thermo Fisher | cat#4351372 | Mouse *Emc10* qRT-PCR |

*Appendix 1 Continued on next page*

*Appendix 1 Continued*

| Reagent type (species) or resource | Designation | Source or reference | Identifiers | Additional information |
|---|---|---|---|---|
| Sequence-based reagent | TaqMan Mm01197555_m1 (*Emc10-2*) | Thermo Fisher | cat#4351372 | Mouse *Emc10* qRT-PCR |
| Sequence-based reagent | Mouse *Gapdh* Endogenous Control | Life Technologies | cat#4352339E | Housekeeping gene for mouse qRT-PCR |
| Sequence-based reagent | TaqMan Hs00382250 (*EMC10*) | Thermo Fisher | cat#4331182 | Human *EMC10* qRT-PCR |
| Sequence-based reagent | TaqMan Hs01597912 (*RANBP1*) | Thermo Fisher | cat#4331182 | Human *RANBP1* qRT-PCR |
| Sequence-based reagent | TaqMan Hs04260367 (*OCT4/POU5F1*) | Thermo Fisher | cat#4331182 | Human *OCT4* qRT-PCR |
| Sequence-based reagent | TaqMan Hs02387400 (*NANOG*) | Thermo Fisher | cat#4331182 | Human *NANOG* qRT-PCR |
| Sequence-based reagent | TaqMan Hs00232429 (*TBR1*) | Thermo Fisher | cat#4331182 | Human *TBR1* qRT-PCR |
| Sequence-based reagent | TaqMan Hs00909233 (*GFAP*) | Thermo Fisher | cat#4331182 | Human *GFAP* qRT-PCR |
| Sequence-based reagent | TaqMan Hs00159598 (*NEUROD1*) | Thermo Fisher | cat#4331182 | Human *NEUROD1* qRT-PCR |
| Sequence-based reagent | TaqMan Hs00220404 (*vGLUT1*) | Thermo Fisher | cat#4331182 | Human *vGLUT1* qRT-PCR |
| Sequence-based reagent | TaqMan Hs00698288 (*DLX1*) | Thermo Fisher | cat#4331182 | Human *DLX1* qRT-PCR |
| Sequence-based reagent | TaqMan Hs00271595 (*BRN2*) | Thermo Fisher | cat#4331182 | Human *BRN2* qRT-PCR |
| Sequence-based reagent | Human *GAPDH* Endogenous Control | Life Technologies | cat#4325792 | Housekeeping gene for human qRT-PCR |
| Sequence-based reagent | Emc10[ASO1] (ASO1081815; TTGTTCCTACAGATCTAGGG) | Ionis Pharmaceuticals | N/A | ICV injection; targets *Emc10* intron 2 |
| Sequence-based reagent | Emc10[ASO2] (ASO1466182; GCCATATCTTTATTAATTAC) | Ionis Pharmaceuticals | N/A | ICV injection; targets *Emc10* intron 1 |
| Chemical compound, drug | Tamoxifen (TAM) | Sigma-Aldrich | cat#T5648 | Solved in corn oil |
| Chemical compound, drug | Y-27632 | Tocris Bioscience | cat#1254 | Differentiation |
| Chemical compound, drug | LDN193189 | Stemgent | cat#04–0074 | Differentiation |
| Chemical compound, drug | SB431542 | Tocris Bioscience | cat#1614 | Differentiation |

*Appendix 1 Continued on next page*

*Appendix 1 Continued*

| Reagent type (species) or resource | Designation | Source or reference | Identifiers | Additional information |
|---|---|---|---|---|
| Chemical compound, drug | XAV939 | Tocris Bioscience | cat#3748 | Differentiation |
| Chemical compound, drug | PD0325901 | Tocris Bioscience | cat#4192 | Differentiation |
| Chemical compound, drug | SU5402 | BioVision Inc. | cat#1645–05 | Differentiation |
| Chemical compound, drug | DAPT | Tocris Bioscience | cat#2634 | Differentiation |
| Chemical compound, drug | BDNF (human) | R&D Systems | cat#248-BDB | Differentiation |
| Chemical compound, drug | NT-3 (human) | PeproTech | cat#450–03 | Differentiation |
| Chemical compound, drug | cAMP | Sigma-Aldrich | cat#A6885 | Differentiation |
| chemical compound, drug | Ascorbic acid | Sigma-Aldrich | cat#A92902 | Differentiation |
| Chemical compound, drug | Ara-C | Sigma-Aldrich | cat#C1768 | Differentiation |
| Chemical compound, drug | Puromycin | Sigma-Aldrich | cat#P8833 | 1 mg/l for NGN2-iN selection |
| Chemical compound, drug | Doxycycline | Sigma-Aldrich | cat#D9891 | For NGN2 induction |
| Chemical compound, drug | Fluo-4 AM | Invitrogen | cat#F14201 | For calcium imaging |
| Chemical compound, drug | Mouse laminin | Corning | cat#354232 | NGN2-iN culture |
| Chemical compound, drug | Lipofectamine 2000 | Life Technologies | cat#11668–030 | Transfection reagent |
| Chemical compound, drug | QIAzol lysis reagent | Qiagen | cat#79306 | Protein extraction from HPC |
| Commercial assay or kit | RNeasy Mini Kit | Qiagen | cat#1038703 | RNA extraction from mouse tissues |
| Commercial assay or kit | miRVana miRNA isolation kit | Ambion (Thermo Fisher) | cat#AM1560 | RNA/miRNA extraction from hiPSC/neurons |
| Commercial assay or kit | High-Capacity RNA-to-cDNA Kit | Applied Biosystems | cat#4387406 | cDNA synthesis |
| Commercial assay or kit | TaqMan Universal Master Mix II | Thermo Fisher Scientific | cat#4440038 | qRT-PCR |
| Commercial assay or kit | iTaq Universal SYBR Green Supermix | Bio-Rad | cat#1725121 | qRT-PCR with SYBR Green |
| Commercial assay or kit | Pierce BCA Protein Assay Kit | Thermo Scientific | cat#23227 | Protein quantification |
| Software, algorithm | ImageJ | NIH | RRID:SCR_003070 | Image and Western blot analysis |
| Software, algorithm | NeuronStudio (v0.9.92) | Ref (80) | RRID:SCR_013798 | Dendritic complexity analysis |
| Software, algorithm | VAA3D (v3.1.00) | Ref (*Peng et al., 2010*; *Peng et al., 2014a*; *Peng et al., 2014b*) | RRID:SCR_002609 | 3D dendritic analysis |
| Software, algorithm | FreezeFrame 3 | Harvard Apparatus | N/A | For fear conditioning analysis |

*Appendix 1 Continued on next page*

*Appendix 1 Continued*

| Reagent type (species) or resource | Designation | Source or reference | Identifiers | Additional information |
|---|---|---|---|---|
| Software, algorithm | TruScan (v1.012–00) | Coulbourn Instruments | N/A | Open field recordings |
| Software, algorithm | gProfiler | Ref (*Raudvere et al., 2019*) | RRID:SCR_006809 | GO term enrichment analysis |
| Software, algorithm | TargetScan (v8.0) | Ref (*Agarwal et al., 2015*) | RRID:SCR_010845 | miRNA target prediction |
| Software, algorithm | konnect2prot | Ref (*Kumar et al., 2023*) | N/A | PPI network analysis |
| Software, algorithm | COSMID | Ref (*Cradick et al., 2014*) | N/A | CRISPR off-target prediction |
| Software, algorithm | miRNet 2.0 | Ref (*Chang et al., 2020*) | N/A | miRNA target network analysis |
| Software, algorithm | Nikon NIS Elements AR (v5.21.03) | Nikon Instruments | RRID:SCR_014329 | Confocal image capture |
| Software, algorithm | GraphPad Prism | GraphPad Software | RRID:SCR_002798 | Data visualization and analysis |
| Other | Mouse astrocyte conditioned media | ScienCell | cat#M1811-57 | 10% in NGN2-iN culture |
| Other | Glass bottom dishes (14 mm) | MatTek | cat#P35G-1.5–14 C | Calcium imaging |

