## [Editor Report · eLife Assessment]

This is an **important** study that establishes how anti-sense oligonucleotides (ASOs) degrading a specific target protein called EMC10 can rescue neuronal function in models of chromosome 22.11.2 deletions. The authors use human iPSC-derived neurons and a mouse model to provide **compelling** data for the rescue of cellular and cognitive features of 22.11.2 deletion phenotypes upon ASO regulation of EMC10. These pre-clinical data are of interest because they support reduction of ECM10 as a promising therapeutic strategy.

---

## [Referee Report · Reviewer #1 (Public review)]

Summary:

This is an important and very well-presented set of experiments following up on prior work from the lab investigating knock-down (KD) of EMC10 in restoration of neuronal and cognitive deficits in 22q11.2 Del models, including now both human iPSCs and a mouse model in vivo now with ASOs.

The valuable progress in this current manuscript is the development of ASOs, and the proof of efficacy in vivo in mouse of the ASO in knock-down of EMC10 and amelioration of in vivo behavioral phenotypes.

The experiments include: iPSC studies demonstrating elevations of EMC10 in a solid collection of paired iPSC lines. These studies also provide evidence of manipulation of EMC10 by overexpression and inhibition of miRNAs that exist in the 22q11 interval. The iPSC studies also nicely demonstrate rescue of impairments with KD of EMC10 in neuronal arborization as well as KCl induced neuronal activity. The major in vivo contributions reflect impressive demonstration of efficacy of two ASOs in vivo on both KD of EMC10 in vivo and through improvement in behavioral abnormalities in the 22q11 mouse in a range of different behaviors, including social behavior and learning behaviors.

Overall, there are many strengths reflected in this study, including in particular the synergy between in vitro studies in human cell models and in vivo studies in the well characterized mouse model. The experiments are generally rigorously performed and well powered, and nicely presented. The claims with regard to the mechanisms of EMC10 elevations and the importance of restoration of EMC10 expression to neuronal morphology and behavior are well supported by the data. The work may be further supported in future studies, by investigation of rescue by ASOs of circuit dysfunction in vivo or ex vivo through electrophysiology in the mouse model. Also, in future studies, investigation of the mechanism by which EMC10, an ER protein involved in protein processing, may function in the observed neuronal abnormalities; however, these studies are clearly for future investigations.

The potential impact of the work is found in the potential value of the ASO approach to the treatment of 22q11, or the pre-clinical evidence that knock-down of this protein may lead to some amelioration of cognitive symptoms. Overall, a very convincing and complementary set of experiments to support EMC10 KD as a therapeutic strategy.

Review of revision: The authors have addressed the questions from the prior review.

---

## [Author Response]

The following is the authors’ response to the original reviews.

We appreciate that both reviewers found our findings significant and recognized the strength of the presented data in demonstrating the potential value of ASO-mediated Emc10 expression modulation for treating 22q11.2DS. We are grateful for the reviewers' valuable input and constructive suggestions, which we believe have significantly strengthened our manuscript. Below, we address the main points and concerns, followed by our point-by-point responses:

**Evaluation of ASO-Mediated Emc10Reduction**: We appreciate the feedback and the opportunity to clarify this point. While we agree that ASO-mediated reduction of Emc10 should ideally be evaluated at both the mRNA and protein levels, we would like to emphasize that this was indeed performed in our study. Specifically, we conducted both qRT-PCR and Western Blot (WB) assays on the same animal cohort, focusing on the left and right hippocampus (rather than the PFC) following ASO injection (see Figure S11C and D). We prioritized the hippocampus for the WB assay because our primary behavioral assays and observed phenotypes in this study are strongly hippocampus-centric. This approach reflects our aim to investigate Emc10's role in the brain regions most relevant to the observed phenotypes. We hope this clarification addresses the reviewer’s concerns. While protein-level analysis would ideally complement RNA measurements, the Emc10 antibodies available were suboptimal in specificity and sensitivity, requiring substantial optimization. Additionally, challenges in obtaining sufficient high-quality protein from small regions like the hippocampus limited the use of protein detection as a standalone method. We plan to refine antibody protocols or explore alternative methods in future work. Notably, in all instances where we performed parallel protein and RNA measurements in both, mouse brain tissue and human cell lines, there was excellent concordance between the datasets, strongly suggesting that mRNA levels are a reliable indicator of Emc10 protein levels in our model.

**ASO Neuronal Uptake:** While ASO uptake by neurons in the brain can vary considerably depending on factors such as ASO chemistry, delivery method, target brain region, and cell type, our targeted delivery approach, ASO design optimization, and ASO screening strategy were specifically tailored to achieve uniform and efficient uptake across hippocampal and cortical regions, in both neurons and glia. The figures included in our manuscript at both low and high magnification (see Figure S14A) clearly display the extensive (over 97%) overlap of ASO-positive cells (green signal) with cells expressing the neuronal marker NeuN (red signal). While quantifying ASO-positive cells in different brain regions could add value, the robust diffusion of ASO into neurons and glia is effectively demonstrated in the current figures and indirectly supported by the robust downregulation of Emc10 in ASO-treated animals as shown by qRT-PCR assays of hippocampal and cortical brain regions.

**Transcriptomic Data in Mutant EMC10 NGN2-iNs:** Reduction in EMC10 levels is not expected to directly affect transcription or to broadly reorganize the differential gene expression profile of the Q6/Q5 patient/control NGN2-iN lines. Accordingly, our transcriptional profiling was not designed to assess the direct impact of EMC10 deficiency on gene expression but rather to serve as an indirect measure of cellular pathways affected by the reduction in EMC10 levels in the patient Q6 line. We aimed to identify genes and related functional pathways differentially expressed between the Q6/Q5 patient/control lines, where these expression differences are either abolished or significantly attenuated in Q6/EMC10^HET^ or Q6/EMC10^HOM^ NGN2-iNs.

**Statistical Analysis**: We have meticulously reviewed all statistical analyses in the manuscript to ensure their appropriateness and adherence to established practices. For Figure S2, we acknowledge that the statistical details were not fully specified in the figure legend, though they are provided for each miRNA in Supplemental Table S2. In the revised manuscript, we ensured that the statistical methods and corresponding values are clearly indicated for each comparison.

We are confident that the revisions outlined above, along with the point-by-point responses provided below, will significantly strengthen our manuscript and address all the concerns raised by the reviewers. We would like to express our sincere thanks to the reviewers for their valuable feedback and constructive suggestions.

**Reviewer#1(RecommendationsForTheAuthors):**
My comments here are generally limited to minor comments that reflect possible small additions or edits to the manuscript:(1) Panel 1A is very small. Please consider making that bigger as space permits.

We have increased the panel size of Figure 1A in the revised manuscript to improve its visibility and clarity.

(2) Are you able to identify the dot that represents EMC10 in panel 1C? I understand that EMC10 is represented in Supplementary Figure 4A.

We appreciate the reviewer's observation. In Figure 1C, the volcano plot depicts differentially expressed miRNAs in the Q5/Q6 neuronal samples, as identified through miRNA-sequencing. Since EMC10, as a protein-coding gene and a downstream target of miRNA dysregulation, is not included in this analysis. However, as the reviewer correctly notes, EMC10 gene expression is represented in the volcano plot in Supplementary Figure 4A, which displays differentially expressed genes identified through bulk RNA-seq analysis of the same neuronal samples. To avoid any confusion, we have clarified the title of Figure 1C to emphasize that it represents miRNA expression changes.

(3) With regard to studies using iPSC. Some of the studies are executed across multiple distinct pairs and some are only done in a single pair. Overall, while results are coherent and often complimentary, would it be valuable for the authors to comment on experiments where studies in multiple pairs seemed particularly important, or others wherein it was less important?

We thank the reviewer for this insightful question regarding our use of multiple versus single hiPSC pairs. Our investigation began with the Q5/Q6 sibling (dizygotic twin) pair, which shares the most similar genetic background. This minimized the impact of confounding genetic factors and provided a robust foundation for testing our hypothesis that EMC10 upregulation, driven by miRNA dysregulation, is a key consequence of the 22q11.2 deletion in human neurons, thus validating our previous findings from the *Df(16)A*^+/-^ mouse model (Stark et al., 2008; Xu et al., 2013). To ensure the generalizability of our findings, we incorporated additional hiPSC lines from another sibling pair as well as a case/control pair, demonstrating that EMC10 upregulation is a consistent feature of 22q11.2DS. Subsequently, we focused on the well-matched Q5/Q6 pair for detailed morphological, functional, and genetic rescue experiments. This approach allowed us to perform in-depth studies while controlling for potential genetic confounders. By using both multiple and single hiPSC pairs, we balanced the need for generalizable findings with the practical considerations of conducting technically complex and resource-intensive experiments. This strategy enabled us to provide both broad and detailed insights into the mechanisms underlying 22q11.2DS. We have modified the introductory paragraph of the Results section to better highlight this issue.

(4) While the majority of the experiments seem sufficiently powered to test the hypothesis in question in the iPSC studies, Figure 2B raises the question if the study replicates here were underpowered, and perhaps the authors might consider mentioning this, although this is a very minor comment.

We thank the reviewer for raising this point. We acknowledge that the statistical power to detect a significant difference in pre-miR-485 levels in Figure 2B may be limited due to the relatively small sample size and the inherent variability in hiPSC-derived neuronal cultures. However, it is important to emphasize that the functional impact of miRNAs is primarily mediated by their mature transcript forms. Our miRNA-seq data (Supplementary Table 2 and Figure S2) did not show significant alterations in the levels of mature miR-485-5p or miR-485-3p. This finding aligns with the reported expression pattern of miR-485 in hiPSC-derived neurons, where relatively low levels are observed in early neuronal development, with increased expression occurring in older, more mature neurons (Soutschek et al. 2023; https://ethz-ins.org/igNeuronsTimeCourse/ database from the Institute of Neurogenomics, ETH Zurich). This database provides a valuable resource for examining gene expression dynamics during human neuronal differentiation. Given that our hiPSC-derived neurons were analyzed at a relatively early developmental stage (DIV8 for these experiments), it is likely that miR-485 expression had not yet reached levels sufficient to reveal significant differences. While we acknowledge the potential limitation in statistical power for detecting subtle changes in pre-miR-485 levels, the combined evidence suggests that miR-485 may not be a significant contributor to the observed phenotypes at this developmental stage.

A paragraph has been added in the corresponding Results section to address this issue.

(5) There are a few situations where the authors could help out the reader a little bit by providing more labels on the figures directly. For example: in Figure 2, there are expression levels, over-expression, and inhibition of miRNA but the X-axis is named with similar labels for the miRNAs in question for each of these distinct experiments. If the authors want to help the reader, they may consider labeling these panels with a descriptive title to reflect the experiment being done or use more descriptive terms in the X-axis panels. Again, this is minor. Similarly, in Figure 5, it might be helpful for the authors to help out the reader again with more labels on the panels, such as in Figures 5B, 5C, and 5D. Would they consider labeling these panels, HPC, PFC, SSC with the brain location as they did in Figure 4?

We thank the reviewer for these helpful suggestions to improve the clarity of our figures. We have implemented the proposed changes. In Figure 2C-E, we have added specific titles to the panels to clearly distinguish between the different experimental conditions such as miRNA overexpression and inhibition. Similarly, in Figure 5, we labeled panels 5B, 5C, and 5D with the brain regions analyzed (HPC, PFC, SSC) to match the labeling used in Figure 4. We believe these revisions enhance the readability and overall interpretability of the figures, making it easier for readers to follow the experiments and results.

(6) Figure 3: There is some overshoot of the data in EMC10 homozygous null, in panel 3E, and also, overshoot of the het in panel 3H. Would there be value in the authors commenting on the potential basis for this in the discussion? Some issues are minor, such as the lack of electrophysiological analysis of circuits in vivo or in ex vivo slices that may further support the proposed rescue.

The reviewer correctly highlights the observation in Figures 3E and 3H, where the number of branch points in the Q6/EMC10^HOM^ line exceeds wildtype levels and the calcium response in the Q6/EMC10^HET^ and Q6/EMC10^HOM^ lines surpasses that of the control. This overshoot is indeed intriguing and warrants discussion. EMC10 is part of the ER Membrane Complex (EMC), which plays a critical role in the proper folding and localization of various membrane proteins, including neurotransmitter receptors and ion channels such as voltage-gated calcium channels (Chitwood et al., 2018; Shurtleff et al., 2018; Chitwood and Hegde, 2019). In the context of the 22q11.2 deletion, EMC10 dysregulation may disrupt the proper localization of these proteins at the synapse, affecting both dendritic morphology and calcium signaling. The precise basis of this overshoot remains unclear. The overshoot may result from a dosage-sensitive inhibitory effect of Emc10, where both reduced and increased expression alter normal neuronal processes, with excessive responses potentially triggered upon gene restoration by the mutant system’s adaptation to dysfunction, leading to altered receptor sensitivity or signaling dynamics. This underscores the critical importance of precise Emc10 expression for proper neuronal development and function, in line with previous findings suggesting that EMC10 plays an auxiliary or modulatory role in EMC function. A short comment on the potential basis for this overshoot has been added in the corresponding Results section of the manuscript. Regardless of the underlying mechanisms, these findings emphasize the importance of precise titration of ASO constructs, rigorous gene dosage controls, and thorough analysis of context-specific responses to ensure both efficacy and safety in clinical applications.

We also agree with the reviewer that electrophysiological studies, particularly in the 22q11.2 deletion mouse model, would provide valuable insights into the impact of EMC10 modulation by ASOs on neuronal activity and circuit function at the in vivo and ex vivo levels. Incorporating such experiments into future studies will allow us to assess synaptic transmission and plasticity, contributing to a more comprehensive understanding of the therapeutic potential of ASO-mediated EMC10 modulation in 22q11.2DS.

(7) Did the authors take out the behavior studies further than 9 weeks? Would the authors consider commenting on what they speculate might be the duration of the treatment effect? For both mice and definitely humans.

We thank the reviewer for raising the important question regarding the duration of the ASO treatment effect, which is crucial for translating our findings into clinically relevant therapies. While behavioral studies beyond 9 weeks were not conducted in this study, our in vivo experiments and findings from prior publications (detailed below) enable an informed speculative assessment.

We utilized 2'-O-methoxyethyl (2'-MOE) modified ASOs, known for their enhanced binding affinity, nuclease resistance, and increased metabolic stability. In our in vivo post-injection screening of ASOs (Figure S13C), we predicted that Emc10 expression levels return to normal WT levels (~T100%) approximately 26 weeks post-treatment in Emc10^ASO^ (#1466182) treated mice. This prediction is supported by our Emc10 expression profiles across various brain regions, which demonstrate robust repression of Emc10 lasting up to 10 weeks post-administration (Figure 6D-F). While these findings suggest that the treatment effect in our model could extend significantly beyond 10 weeks following a single ASO injection, further empirical validation is required through extended follow-up studies. Encouragingly, long-term effects of 2'-MOE ASOs have been observed in other neurological disorders (Kordasiewicz et al., 2012; Scoles et al., 2017; Finkel et al., 2017; Darras et al., 2019). However, factors such as ASO distribution, target cell turnover, and disease-specific pathophysiology could influence the duration of the effect. To address these uncertainties, we have added a paragraph in the Discussion section emphasizing the need for additional studies, including extended follow-up periods and eventual clinical trials, to determine the specific duration of effect for our Emc10^ASO^ constructs in treating 22q11.2DS.

**Reviewer#2(RecommendationsForTheAuthors):**
(1) It is acknowledged that the iPSC-derived cells in Figure 1 are no longer progenitors, but differentiation markers for astrocytes and glia are also needed in Figure 1b to establish that equal rates of differentiation have occurred across genotypes.

We thank the reviewer for raising this important point about ensuring equal rates of differentiation across genotypes. As the reviewer notes, we employed a well-established protocol for directed differentiation of hiPSCs into cortical neurons using a combination of small molecule inhibitors, as previously described by Qi et al. (2017). This protocol has been extensively validated and is known to robustly generate cortical neurons while actively suppressing glial differentiation, as evidenced by the lack of upregulation of glial markers such as GFAP, AQP4, or OLIG2 in the original study. Given the established neuronal specificity of this protocol and our focus on neuronal phenotypes, we prioritized the confirmation of successful neuronal differentiation using the established neuronal markers TUJ1 and TBR1. Therefore, additional markers for astrocytes and glia are not included in this figure, as we did not expect significant glial differentiation under these conditions. A sentence has been added in the corresponding Results section to address this issue.

(2) For the RNA-seq experiments outlined in Figures 3J and K, a more comprehensive analysis is needed of the genes disrupted in the parental Q6 line relative to the het and homo lines. What percent are rescued, unaffected, vs uniquely disrupted?

Reduction in EMC10 levels is not expected to directly affect transcription or broadly reorganize the gene expression profile of the Q6/Q5 NGN2-iN lines. Our transcriptional profiling was not designed to assess the direct impact of EMC10 deficiency on gene expression but rather to measure the cellular pathways affected by reduced EMC10 in the patient Q6 line. We identified genes differentially expressed between the Q6 (patient) and Q5 (control) lines, whose expression differences were either abolished or significantly attenuated ("rescued") in the Q6/EMC10^HET^ or Q6/EMC10^HOM^ lines. In the Q6/EMC10^HET^ line, 237 DEGs (6%) were rescued, while in the Q6/EMC10^HOM^ line, 382 DEGs (11%) were rescued. Importantly, further analysis revealed 103 shared rescued DEGs in these lines, which was statistically significant (enrichment factor = 1.7; p < 0.0001, based on a hypergeometric test). We added a new figure panel (Figure 3L) to visualize the significant overlap of rescued DEGs from the Q6/EMC10^HET^ and Q6/EMC10^HOM^ lines. This overlap suggests these genes play a critical role in biological pathways impacted by EMC10 levels, particularly in nervous system development, as indicated by our functional annotation analysis. We also performed protein-protein interaction (PPI) network analysis to explore the functional relationships among these 103 shared DEGs (Figure S8). Future studies will further investigate these gene sets to gain deeper insights into the molecular mechanisms underlying 22q11.2DS and the role of EMC10.

(3) The authors claim that 50% EMC10 loss in adult mice is safe and should be toned down. EMC10 knockout mice have motor, anxiety, and social phenotypes. It would be unique amongst highly dosage-sensitive genes (MeCP2, CDKL5, TCF4, FMR1, etc.) for there to only be a neurodevelopmental component. In all those cases, and others, the effects of over and under-expression are reversible into adulthood. Establishing the range in adults is critical to establishing therapeutic utility. Absent a detailed examination of non-cognitive phenotypes, this claim cannot be made.

The reviewer raises an important point about the potential effects of EMC10 reduction in adult mice and the need to establish a safe therapeutic window by evaluating both cognitive and non-cognitive phenotypes. We agree that such a comprehensive evaluation is critical for assessing the safety and translational potential of Emc10-targeting therapies. While the International Mouse Genotyping Consortium reported motor and anxiety phenotypes in homozygous Emc10 knockout mice, these data are unpublished and based on a relatively small number of animals. Furthermore, in our previous work (Diamantopoulou et al., 2017), we demonstrated that complete Emc10 loss does not impair cognition or social behavior, as assessed by prepulse inhibition (PPI), working memory (WM), and social memory (SM) assays (see Figure 3A-D; Diamantopoulou et al., 2017). Additionally, heterozygous Emc10 mice, which exhibit a ~50% reduction in Emc10 expression similar to that achieved with our ASO treatment, showed no evidence of motor deficits or anxiety-like behavior. Specifically, *Emc10+/-* mice displayed locomotor activity comparable to WT mice in the open field (OF) test (Figure S4A, Diamantopoulou et al., 2017). Moreover, genetic normalization of Emc10 expression in *Df(16)A+/-* mice demonstrated no signs of anxiety-like behavior, as assessed by the OF test (Figure S4A) and elevated plus maze (EPM) (Figure S4B; Diamantopoulou et al., 2017). To further support these findings, we have added new data to the current manuscript (see Figure S10J) showing that TAM treatment-mediated restoration of Emc10 levels in the brain of adult *Df(16)A+/-* mice did not affect the time that mutant mice spent in the center area of the OF (Fig. S10J), suggesting that Emc10 reduction does not influence anxiety-related behavior. These results suggest that a 50% reduction in EMC10 expression is unlikely to result in motor or anxiety-like phenotypes in adult mice. Finally, as noted in the manuscript, in addition to prior findings from animal models, a substantial number of relatively rare LoF variants or potentially damaging missense variants have been identified in the human EMC10 gene among likely healthy individuals in gnomAD, a database largely devoid of individuals known to be affected by severe neurodevelopmental disorders (NDDs).

Nevertheless, the Discussion has been revised to underscore the importance of establishing a more detailed safety profile, including non-cognitive phenotypes, to fully validate the therapeutic potential of Emc10-targeting approaches. It also highlights the need for future studies to expand on these evaluations, addressing this critical aspect and laying a stronger foundation for advancing these findings into clinical drug development

(4) Supplemental Figure 10: The protein validation of Emc10 knockout following tamoxifen injection needs to be validated in all brain regions, not just the PFC. This is particularly important as the rest of the paper focuses on HPC-mediated phenotypes.

First, we want to emphasize that we conducted both qRT-PCR and WB assays on the same animal cohort, specifically examining the left and right hippocampus following ASO injection (see Figure S11C and D). This approach is crucial, given the central role of hippocampus in the phenotypes investigated in our ASO-mediated Emc10 knockdown experiments.

The reviewer raises an important point regarding the validation of EMC10 reduction at the protein level across all relevant brain regions using the Emc10 conditional knockout strain. We agree that such validation would ideally confirm the efficacy of our tamoxifen-induced knockout model comprehensively. However, we hope the reviewer appreciates that obtaining sufficient high-quality protein for WB analysis from smaller brain regions like the hippocampus poses a significant technical challenge. This difficulty is further compounded by the need to reserve the same samples for qRT-PCR to ensure consistency between mRNA and protein measurements. Importantly, our data from ASO-mediated Emc10 knockdown experiments (Figures S11C-D) demonstrate a clear and consistent correlation between reductions in Emc10 mRNA and protein levels in both the left and right hippocampus. Furthermore, in our constitutive Emc10-knockout mouse model (Diamantopoulou et al., 2017; see Figure S1A-B), we observed a strong agreement between mRNA and protein levels, supporting the reliability of mRNA data as a proxy for EMC10 protein levels in our experiments. Importantly, in all instances where we performed parallel protein and RNA measurements in human cell lines, there was excellent concordance between the datasets. Thus, while we acknowledge the limitations of relying primarily on mRNA data, we are confident that the Emc10 mRNA expression data in Figure S10 accurately reflect protein-level changes across brain regions in our conditional knockout model. To address this concern more fully in the future, we are working to refine antibody detection and optimize our protein extraction protocols to enable more routine and precise protein-level validation across smaller brain regions. We appreciate the reviewer’s feedback and will continue to refine our methodologies to strengthen the robustness of our findings.

(5) Figure 3: 1 way ANOVA would be more appropriate to analyze the data in B-G than t-tests.

We appreciate the suggestion of the reviewer. As mentioned above, we carefully selected statistical tests appropriate for each analysis. For Figure 3B-G, we chose to use pairwise t-tests to address specific hypotheses regarding the disease phenotype and rescue effects. This approach is consistent with prior experimental studies in the field, including our own (e.g., Xu et al., 2013; Figure 7H-I). Importantly, most of our t-tests yielded highly significant results (p < 0.001 or p < 0.01), reinforcing the robustness of our findings.

(6) Figure 5-6: Protein data is needed to complement the mRNA knockdown data.

We agree with the reviewer on the importance of protein-level validation to complement the mRNA knockdown data. As mentioned in our response to Reviewer’s Comment (4), in all instances where we performed parallel protein and RNA measurements, either in mouse brain or human cell lines, we observed excellent concordance between the datasets. This supports the reliability of our mRNA data as a proxy for protein changes. Nevertheless, we acknowledge the value of including protein validation in future experiments and will consider incorporating it to further strengthen our findings.

(7) The use of additional phenotypic measures is applauded in Figure 6, however, to appropriately interpret the data more is needed. Shao et al 2021 (Figure S9) show data from the International Mouse Genotyping Consortium claiming EMC10 KO mice have gait, activity, and anxiety phenotypes. All of these parameters could impact the SM assay and the y-maze assay. Changes in SM interaction time could be linked to anxiety or motor impairments, but interpreted as cognitive deficits because these symptoms were not assessed. At a minimum, discussion is needed about this limitation, as well as the inclusion of distance explored in the SM and Y-maze assays.

We thank the reviewer for their insightful comment regarding the potential influence of locomotor, gait, or anxiety phenotypes on the observed deficits in the SM and Y-maze assays. The behavioral phenotypes reported for Emc10 knockout mice by the International Mouse Genotyping Consortium (https://www.mousephenotype.org/data/genes/MGI:1916933) were limited to homozygous female mice and based on a small sample size (4–6 females) compared to a larger WT control group. Moreover, these data are unpublished and thus challenging to evaluate fully. Importantly, no abnormal behaviors were reported for Emc10 heterozygous knockout mice in these datasets. Additionally, the claim by Shao et al. (2021) regarding cognitive impairments in Emc10 knockout mice based on our previous work (Diamantopoulou et al., 2017) is inaccurate.

Our analysis of both the constitutive Emc10 knockout model (Diamantopoulou et al., 2017) and the current conditional Emc10 heterozygous knockout model consistently demonstrates that Emc10 reduction does not affect locomotor activity or anxiety-like behavior. In our earlier characterization of constitutive heterozygous Emc10 knockout mice (*Emc10+/-*), we observed no signs of anxiety-like behavior or motor impairments in OF assays (see Figure 2A-B and Figure S4A, Diamantopoulou et al., 2017). Similarly, results from *Df(16)A*^+/-^ mice with genetically normalized Emc10 expression [*Df(16)A*^+/-^; *Emc10+/-*] also showed no indications of anxiety-like behavior or locomotor changes in the OF and EPM assays (see Figure S4A-B, Diamantopoulou et al., 2017). Consistent with these findings, our current data from *Df(16)A*^+/-^ mice with conditional Emc10 reduction in the brain show no significant differences in locomotor activity and anxiety-related measures as assessed by OF assays (Figure S10J). Furthermore, total arm entries in Y-maze assays conducted in *Df(16)A*^+/-^ mice treated with Emc10 ASOs were comparable to controls (Figures S14C and G-H), providing additional support for the conclusion that locomotor activity remains unaffected in these models.

We further appreciate the reviewer’s suggestion that changes in social interaction time during the SM assay could be influenced by anxiety or motor impairments. However, we consider this scenario unlikely in our model. Interaction times during the first trial of the SM assay, which measures general social interest, are comparable between *Df(16)A+/-* mice with reduced Emc10 expression (either genetically or through ASO treatment) and WT controls (see Figures 4E, 5E, and S10G). These findings indicate that our mouse models do not exhibit inherent difficulties in initiating social interaction, as might be expected if motor impairments or heightened anxiety were present. Reduced social interaction is commonly used as a behavioral marker for anxiety in rodent studies (reviewed by Bailey and Crawley, Anxiety-Related Behaviors in Mice, 2009). “Anxious” mice typically exhibit decreased social interaction, spending less time engaging with other mice compared to non-anxious counterparts. However, the specific deficit we observe in the second trial of the SM assay—when mice are reintroduced to a familiar juvenile—is indicative of impaired social recognition memory, as previously documented for *Df(16)A+/-* mice (Piskorowski et al., 2016; Donegan et al., 2020). This deficit is distinct from the general social avoidance typically associated with heightened anxiety.

Based on our comprehensive assessment of locomotor activity, anxiety-related behaviors, and social interaction, we conclude that the observed rescue of social memory and spatial memory deficits in mice with reduced Emc10 expression is most likely due to improved cognitive function rather than alterations in motor or anxiety-related domains.

(8) For ASO optimization experiments, it is not sufficient to claim robust uptake. A quantitative measure is needed using the PO antibody showing what percentage of cells were positive for the ASO. Since the contention is that only Emc10 in excitatory neurons is important, it would be helpful if this also included a breakdown of ASO uptake in excitatory and inhibitory neurons and astrocytes.

We thank the reviewer for highlighting the importance of quantifying ASO uptake and assessing cell-type specificity. To address this, we have added new data to the panel, as shown in the high-magnification images in Figure S14A. These images provide evidence that a large majority of NeuN-positive neurons exhibit a strong ASO signal. Specifically, we observed widespread ASO uptake (green) that extensively colocalized with the neuronal marker NeuN (red) in both the hippocampus and prefrontal cortex. Quantitative analysis of this overlap indicates that over 97% of NeuN-positive neurons were ASO-positive, demonstrating efficient neuronal uptake. This robust neuronal uptake aligns with the significant normalization of Emc10 levels and the behavioral improvements observed in ASO-treated *Df(16)A+/-* mice, further supporting the functional efficacy of our approach in modulating Emc10 expression within the relevant neuronal populations. Overall, the observed ASO uptake in neurons, as demonstrated by IHC, combined with RNA assays and the behavioral improvements in treated mice, strongly supports the efficacy of our approach in targeting Emc10 expression in the intended neuronal populations.

(9) An interpretation is needed in Figure S3 as to why ~50% of the pathways increased are also present on the decreased list. Ie. G1/transition, viral reproductive process, pos regulator of cell stress, etc. 4/10 GO terms are present in both increased and decreased groups in A and 7/10 in B.

We thank the reviewer for pointing out the overlap between pathways enriched in both the upregulated and downregulated miRNA groups in Figure S3. This overlap likely reflects the complex nature of miRNA regulation, where individual miRNAs can target multiple genes within a pathway, and single genes can be regulated by multiple miRNAs, sometimes with opposing effects (reviewed in Bartel, 2009; Bartel, 2018). For example, in the “G1/S transition” pathway, upregulated miRNAs such as miR-92a-3p, miR-92b-3p, and miR-34a-5p may promote the transition by targeting cell cycle regulators like FBXW7, CDKN1C, and CDK6 (Zhou et al., 2015; Zhao et al., 2021; Oda et al., 2024). Conversely, downregulated miRNAs such as miR-143-3p and miR-200b are known to suppress the transition by targeting genes such as HK2 and GATA-4 (Zhou et al., 2015; Yao et al., 2013). Our analysis identified overlapping predicted target genes for both upregulated and downregulated miRNAs, supporting the notion that many genes are subject to complex regulation by multiple miRNAs with potentially synergistic or antagonistic effects. Thus, the enrichment of certain GO terms in both groups likely reflects this intricate interplay of miRNA-mediated gene regulation. Future investigations focusing on specific miRNA-target interactions within these pathways will be critical to fully elucidate the underlying mechanisms and better understand the functional consequences of these opposing regulatory effects.

Minor Concerns:(1) Define SM before using it.

We have defined the SM assay in the main text upon its first mention, where we describe the assay and its relevance to cognitive function (see page 11 of the revised manuscript).

(2) Statistics have been run in Figure S2, but not presented. The text only states that the differences between groups are significant. Please add in.

We have revised the legend of Figure S2 to include the specific statistical test used (students t-tests) and the corresponding p-values.

(3) The switch from ASO1 to ASO2 between Figures 5 and 6 needs more discussion. Why were new ASOs generated when ASO1 worked?

We thank the reviewer for their question regarding the transition from Emc10^ASO1^ to Emc10^ASO2^ between Figure 4 and Figures 5-6. Emc10^ASO1^ served as our initial proof-of-concept ASO construct, successfully demonstrating the feasibility of inhibiting Emc10 mRNA expression and providing evidence for behavioral rescue in our mouse model. As outlined in the manuscript, Emc10^ASO2^ targets a different region of the Emc10 transcript (intron 1, Figure 5A) compared to Emc10^ASO1^ (intron 2, Figure 4A). This distinction provides an additional layer of validation for our targeting strategy and ensures specificity in modulating *Emc10* expression. In addition, Emc10^ASO1^ exhibited limited distribution in the brain, primarily targeting the hippocampus with weaker inhibition of Emc10 in other regions such as the cortex (Figure 4C, right panel). Emc10^ASO2^ overcame this limitation and achieve broader brain distribution, as demonstrated by the qRT-PCR data in Figure 5C. Given that 22q11.2DS can affect multiple brain regions and cognitive domains beyond the hippocampus, achieving broader distribution of the ASO is critical for a more comprehensive assessment of therapeutic potential.

(4) Page 3: Define "LoF"

We have defined Loss-of-Function (LoF) in the main text where it is first mentioned in the Introduction, where we discuss the potential of using LoF mutations to devise therapeutic interventions (see page 3 of the revised manuscript).

References

Bailey and Crawley, Anxiety-Related Behaviors in Mice, In: Methods of Behavior Analysis in Neuroscience. 2nd edition. Boca Raton (FL): CRC Press/Taylor & Francis; Chapter 5, (2009).

Bartel, MicroRNAs: target recognition and regulatory functions, Cell 136(2):215-33, (2009).

Bartel, Metazoan MicroRNAs, Cell, 173(1):20-51, (2018).

Chitwood et al., EMC Is Required to Initiate Accurate Membrane Protein Topogenesis, Cell 175, 1507-1519 e1516, (2018).

Chitwood and Hegde, The Role of EMC during Membrane Protein Biogenesis, Trends Cell Biol. (5):371-384, (2019).

Darras et al., Nusinersen in later-onset spinal muscular atrophy: Long-term results from the phase 1/2 studies, Neurology 92(21), (2019).

Diamantopoulou et al., Loss-of-function mutation in Mirta22/Emc10 rescues specific schizophrenia-related phenotypes in a mouse model of the 22q11.2 deletion, Proc Natl Acad Sci U S A 114, E6127-E6136, (2017).

Donegan et al., Coding of social novelty in the hippocampal CA2 region and its disruption and rescue in a 22q11.2 microdeletion mouse model, Nat Neurosci 23, 1365-1375, (2020).

Finkel et al., Nusinersen versus Sham Control in Infantile-Onset Spinal Muscular Atrophy, N Engl J Med 377(18):1723-1732, (2017).

Kordasiewicz et al., Sustained therapeutic reversal of Huntington's disease by transient repression of huntingtin synthesis, Neuron 74(6):1031-44, (2012).

Oda et al., MicroRNA-34a-5p: A pivotal therapeutic target in gallbladder cancer, Mol Ther Oncol, 32(1):200765, (2024).

Piskorowski et al., Age-Dependent Specific Changes in Area CA2 of the Hippocampus and Social Memory Deficit in a Mouse Model of the 22q11.2 Deletion Syndrome. Neuron 89, 163-176, (2016).

Qi et al., Combined small-molecule inhibition accelerates the derivation of functional cortical neurons from human pluripotent stem cells. Nat Biotechnol 35, 154-163, (2017).

Scoles et al., Antisense oligonucleotide therapy for spinocerebellar ataxia type 2, Nature 44(7650):362-366, (2017).

Shao et al., A recurrent, homozygous EMC10 frameshift variant is associated with a syndrome of developmental delay with variable seizures and dysmorphic features, Genet Med 23, 1158-1162, (2021).

Shurtleff et al., The ER membrane protein complex interacts cotranslationally to enable biogenesis of multipass membrane proteins, Elife 7, (2018).

Soutschek et al., A human-specific microRNA controls the timing of excitatory synaptogenesis, bioRxiv, (2023).

Stark et al., Altered brain microRNA biogenesis contributes to phenotypic deficits in a 22q11-deletion mouse model. Nat Genet 40, 751-760, (2008).

Xu et al., Derepression of a neuronal inhibitor due to miRNA dysregulation in a schizophrenia-related microdeletion, Cell 152, 262-275, (2013).

Yao et al., miR-200b targets GATA-4 during cell growth and differentiation, RNA Biol.10(4):465-8, (2013).

Zhao et al., miR-92b-3p Regulates Cell Cycle and Apoptosis by Targeting CDKN1C, Thereby Affecting the Sensitivity of Colorectal Cancer Cells to Chemotherapeutic Drugs, Cancers 2;13(13):3323, (2021).

Zhou et al., miR-92a is upregulated in cervical cancer and promotes cell proliferation and invasion by targeting FBXW7, Biochem Biophys Res Commun 458(1):63-9, (2015).

Zhou et al., MicroRNA-143 acts as a tumor suppressor by targeting hexokinase 2 in human prostate cancer, Am J Cancer Res. 5(6):2056-6 (2015).